# Adversarial Latent Embedding Repair for LLM Continual Learning

Xilin Xia [1]   Xialiang Tong [2]   Jie Wang [1⊠]   Chi Ma [1]   Shengxue Li [1]   Yinqi Bai [1]   Yuhang Jiang [1]   Xing Li [2]
Jianye Hao [2 3]   Mingxuan Yuan [2]   Feng Wu [1]

## Abstract

Research on continual learning for LLMs seeks to acquire new skills without catastrophic forgetting of established prior knowledge. However, domain-specific fine-tuning still triggers severe, long-tailed forgetting issues even under narrow updates, particularly when the pre-training data is inaccessible. To tackle this challenge, we propose ALER, a data-free continual learning framework that adversarially searches for a small set of latent prompt embeddings to maximize logit divergence from a frozen reference model, proactively exposing high-risk forgetting modes at each step. It then performs online distillation from the frozen reference using the discovered embeddings to retain prior behaviors while preserving target-domain adaptation. We provide theoretical guarantees on the efficiency of our targeted repair, and extensive experiments demonstrate consistent improvements in the retention–adaptation frontier over representative baselines across 2 domain-specific fine-tuning datasets and 6 general-purpose benchmarks, suggesting a more proactive approach for LLM continual learning.

## 1. Introduction

Continual learning confronts a central tension in modern AI: adapting to new data while preserving hard-earned prior knowledge. (Grossberg, 1980; McCloskey & Cohen, 1989; Kirkpatrick et al., 2017) In the context of large language models (LLMs), this paradigm manifests as **continual domain post-training**, which iteratively adapts a pre-trained checkpoint to new tasks, preferences, or environments while retaining broad, general-purpose capabilities acquired during pre-training. (Gururangan et al., 2020; Ke et al., 2023) Such capability is increasingly necessary in practice, since deployed models must evolve with shifting domains and constraints, yet pre-training corpora are often inaccessible due to privacy, licensing, or sheer scale. (Carlini et al., 2021; Huang et al., 2024) However, even narrow updates can induce severe, long-tailed degradation in pre-trained performance; furthermore, the unavailability of pre-training data renders standard rehearsal-based mitigation impractical. (Luo et al., 2025; Li et al., 2024) This exposes a critical research gap: achieving reliable knowledge retention during continual post-training, particularly where forgetting is concentrated yet high-impact.

Recent research has actively explored this **pre-training-data-inaccessible** regime. Existing approaches can be broadly grouped into three directions: (i) constraining parameter drift via regularization or trust-region objectives (Kirkpatrick et al., 2017; Zhu et al., 2025); (ii) approximating missing pre-training data with synthetic or reconstructed rehearsal (Huang et al., 2024; Bansal & Sanghavi, 2025); and (iii) preserving prior representations through forgetting-aware objectives and architectural or parameter-efficient constraints (Li et al., 2024; Lin et al., 2025). Despite this progress, most methods remain largely *reactive*: they accept the forgetting pattern induced by incoming data and then blunt it through global constraints or generic replay. Crucially, however, forgetting is often *concentrated* and localized—regressions tend to cluster on a small subset of conceptual regions (Toneva et al., 2018; Jin & Ren, 2024b; Chen et al., 2025b). Therefore, without first pinpointing where the prior breaks, limited retention effort can be diluted and fail to protect the most vulnerable behaviors.

Departing from the reactive paradigm, we propose ALER (**A**dversarial **L**atent **E**mbedding **R**epair), a proactive framework for pre-training-data-inaccessible domain-specific fine-tuning. The core idea is a find-before-fix protocol: first identify where the current model is most likely to forget, then spend the limited retention budget on those high-risk regions. Guided by a learning-dynamics analysis and controlled motivation experiments on lightweight Transformers, we corroborate that forgetting is markedly non-uniform and long-tailed (Toneva et al., 2018; Jin & Ren, 2024b). Motivated by this structure, ALER adversarially

[1]MoE Key Laboratory of Brain-inspired Intelligent Perception and Cognition, University of Science and Technology of China [2]Noah's Ark Lab, Huawei Technologies [3]College of Intelligence and Computing, Tianjin University. Correspondence to: Jie Wang <jiewangx@ustc.edu.cn>.

*Proceedings of the 43rd International Conference on Machine Learning*, Seoul, South Korea. PMLR 306, 2026. Copyright 2026 by the author(s).

searches for a small set of latent prompt embeddings that maximizes a forgetting-risk functional, which is instantiated as the predictive divergence between the current model and a frozen reference, and thereby localizes the most compromised regions of the prior. **ALER** then performs lightweight online distillation from the frozen reference on the discovered embeddings, selectively repairing the fragile tail while preserving target-domain adaptation.

To demonstrate the effectiveness of **ALER**, we conduct domain-specific fine-tuning on two target-domain datasets and assess retention on a suite of six general-purpose benchmarks. Across extensive comparisons, **ALER** consistently improves the retention–adaptation frontier, achieving strong target-domain gains while substantially reducing regressions in general capabilities. Notably, it preserves the averaged general-benchmark score essentially intact, within a $0.02\%$ drop, and even yields a slight $0.12\%$ improvement in some settings while maintaining comparable target accuracy.

We summarize our three major contributions as follows. **(1) Characterization:** We study domain-specific fine-tuning in a pre-training-data-inaccessible regime and, via learning-dynamics analysis and controlled toy experiments, substantiate that forgetting is highly non-uniform and long-tailed rather than diffuse. **(2) Method:** We introduce **ALER**, a proactive find-before-fix framework that adversarially searches for a small set of latent prompt embeddings maximizing a forgetting-risk functional, and performs lightweight online distillation on these embeddings to selectively repair the fragile tail of prior behaviors. **(3) Evaluation:** On two target-domain datasets with retention evaluated over six general-purpose benchmarks, **ALER** consistently improves the retention–adaptation frontier, preserving averaged general performance essentially intact while maintaining comparable target accuracy.

## 2. Preliminary

**Language modeling** We consider an auto-regressive language model parameterized by $\theta \in \mathbb{R}^P$ over a vocabulary $\mathcal{V}$ with $|\mathcal{V}| = V$. Given a context $\mathbf{c}$, the model outputs logits $z_\theta(\mathbf{c}) \in \mathbb{R}^V$ and a conditional distribution $\pi_\theta(\cdot \mid \mathbf{c}) = \text{softmax}\left(z_\theta(\mathbf{c})\right) \in \Delta^{V-1}$. For a next token $y \in \mathcal{V}$, the per-token negative log-likelihood (NLL) is $\ell(\mathbf{c}, y; \theta) = -\log \pi_\theta(y \mid \mathbf{c})$.

**Domain-specific fine-tuning** We fine-tune a released checkpoint $\theta_{\text{ref}}$ on a target-domain dataset $\mathcal{D}_f = \left\{\left(\mathbf{x}^{(u)}, \mathbf{y}^{(u)}\right)\right\}_{u=1}^M$, where $\mathbf{x}^{(u)}$ is a prompt sequence and $\mathbf{y}^{(u)} = \left(y_1^{(u)}, \ldots, y_{L_u}^{(u)}\right)$ is its completion. The SFT objec-

tive is the standard token-level NLL,

$$\mathcal{L}_{\text{sft}}(\theta) = \sum_{(\mathbf{x}, \mathbf{y}) \in \mathcal{D}_f} \sum_{l=1}^{|\mathbf{y}|} -\log \pi_\theta(y_l \mid \mathbf{x}, \mathbf{y}_{<l}). \quad (1)$$

We denote the pre-training dataset by $\mathcal{D}_p = \left\{\mathbf{x}^{(n)}\right\}_{n=1}^N$, where each $\mathbf{x}^{(n)} = \left(x_1^{(n)}, \ldots, x_{T_n}^{(n)}\right)$ and $x_t^{(n)} \in \mathcal{V}$. Crucially, $\mathcal{D}_p$ (or its underlying distribution) is *inaccessible* in our setting and cannot be replayed.

**Sample-wise forgetting** Following previous work (Jin & Ren, 2024a), we measure forgetting as the degradation in the model's compression capability (i.e., NLL increase) over prior pre-trained knowledge. Specifically, for a pre-training sample $\mathbf{x} \in \mathcal{D}_p$, the forgetting is defined as

$$\text{forget}(\mathbf{x}) \triangleq \ell_p(\mathbf{x}; \theta') - \ell_p(\mathbf{x}; \theta), \quad (2)$$

where $\ell_p(\mathbf{x}; \theta) = -\sum_{t=1}^T \log \pi_\theta(x_t \mid \mathbf{x}_{<t})$ and $\mathbf{x}_{<t} = (x_1, \ldots, x_{t-1})$.

## 3. Motivation

In this section, we motivate **ALER** from both a learning-dynamics perspective and controlled experiments, establishing that forgetting is inherently *non-uniform* and *concentrated*, which calls for a *find-before-fix* strategy.

### 3.1. Learning Dynamics of Forgetting

Consider one SFT update on a fine-tuning example $\chi_u = (\mathbf{x}^{(u)}, \mathbf{y}^{(u)})$: $\theta' = \theta - \eta \nabla_\theta \mathcal{L}_{\text{sft}}(\theta; \chi_u)$. A first-order learning-dynamics expansion (Ren & Sutherland, 2024) gives, for any pre-training sample $\mathbf{x}^{(n)}$,

$$\ell_p^{(n)}(\theta') - \ell_p^{(n)}(\theta) = -\eta \sum_{t=1}^{T_n} \sum_{l=1}^{L_u} \delta_{t,l} + O(\eta^2), \quad (3)$$

where $\mathbf{c}_{n,t} \triangleq \mathbf{x}_{<t}^{(n)}$ and $\mathbf{c}_{u,l} \triangleq (\mathbf{x}^{(u)}, \mathbf{y}_{<l}^{(u)})$, and

$$\delta_{t,l} = g_\theta(\mathbf{c}_{n,t}, x_t^{(n)})^\top \mathcal{K}_\theta(\mathbf{c}_{n,t}, \mathbf{c}_{u,l}) \, g_\theta(\mathbf{c}_{u,l}, y_l^{(u)}). \quad (4)$$

Here $g_\theta(\mathbf{c}, y) \triangleq \nabla_z \ell(\mathbf{c}, y; \theta)$ denotes the logit-space gradient with $z = z_\theta(\mathbf{c})$, and $\mathcal{K}_\theta(\mathbf{c}, \mathbf{c}')$ is the neural tangent kernel (NTK) induced by the model; its formal definition and derivation are deferred to Appendix E.1.

Intuitively, this kernel measures how much a gradient update induced by a fine-tuning context changes the prediction at a prior context; large kernel alignment means the new example can strongly interfere with that part of the old behavior. Eq. (4) implies two immediate consequences: **(i) Non-uniform impact.** Forgetting decomposes into interaction terms $\delta_{t,l}$, hence different prior samples incur markedly

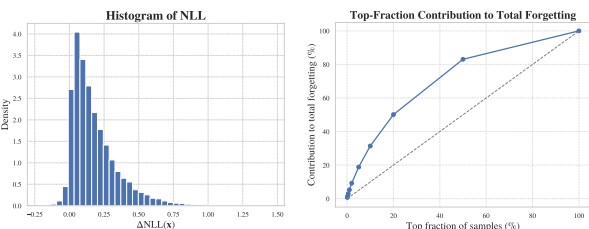

*Figure 1.* **Forgetting is long-tailed and concentrated.** *Left:* histogram of sample-wise forgetting $\Delta\text{NLL}(\mathbf{x})$ on the pre-training corpus after fine-tuning. *Right:* cumulative share of total forgetting contributed by the top fraction of samples (sorted by $\Delta\text{NLL}$); over $50\%$ of total forgetting comes from the top $20\%$ samples, far above the uniform baseline (dashed).

different damage depending on their coupling to the incoming update. **(ii) Model-mediated coupling.** The NTK $\mathcal{K}_\theta$ quantifies how updates propagate through the model, inducing structured dependencies between fine-tuning contexts and prior contexts. These observations motivate allocating retention effort via *model-guided localization* rather than uniform constraints.

### 3.2. Motivation Experiment

We empirically probe the *shape* of forgetting in a controlled synthetic setting. We use a Hidden Markov Model (HMM) (Baum & Petrie, 1966; Rabiner, 1989) to generate a pre-training dataset $\mathcal{D}_p$ and pre-train a lightweight Transformer; we then fine-tune the model on a *narrow* distribution obtained from another HMM with the same sub-structure. We measure sample-wise forgetting on $\mathcal{D}_p$ as $\Delta\text{NLL}(\mathbf{x})$ defined in (2); full details are deferred to Appendix B.1.

Fig. 1 reveals two salient phenomena. **(i) Long-tailed forgetting.** The distribution of $\Delta\text{NLL}(\mathbf{x})$ concentrates near zero for most samples, yet exhibits a pronounced right tail: a small subset suffers substantially larger degradation. A tiny fraction attains negative $\Delta\text{NLL}$, indicating incidental improvements on a few sequences. **(ii) Strong concentration.** When ranking samples by $\Delta\text{NLL}$, the cumulative contribution curve bends sharply upward: more than half of the total forgetting is attributable to only the top $20\%$ most-forgotten samples.

**Implication.** Forgetting is not a diffuse, uniform drift over the prior; it is dominated by a small, high-impact subset. This concentration suggests that effective retention should prioritize *localizing* the most vulnerable modes and allocating repair budget to them—precisely the design principle we instantiate later via risk-maximizing prompt search and targeted distillation.

## 4. Methodology

We introduce **ALER**, a two-stage framework for continual domain post-training when pre-training data are unavail-

---

**Algorithm 1** Fine-tuning with **ALER**

**Input:** ref-model $\pi_{\text{ref}}$, model $\pi_\theta$, fine-tuning dataset $\mathcal{D}_f$.
▷ Standard Fine-tuning Loop
**Function** FINETUNE$(\pi_\theta, \pi_{\text{ref}}, \mathcal{D}_f)$
    **for** $t = 1$ **to** $T$ **do**
        $\mathcal{L}_{\text{target}} \leftarrow \text{DownStreamTask}(\pi_\theta, \pi_{\text{ref}}, \mathcal{D}_f)$
        $\mathcal{L} \leftarrow \mathcal{L}_{\text{target}} + \lambda \cdot \textbf{ALER}\,(\pi_\theta, \pi_{\text{ref}})$
        $\theta \leftarrow \theta - \eta_\theta \nabla_\theta \mathcal{L}$
    **end for**
**End Function**

---

**Function ALER** $(\pi_\theta, \pi_{\text{ref}})$
    Initialize latent embedding $\mathbf{e} \in \mathbb{R}^{l \times d}$
    ▷ Phase I: Adversarial Search with Risk Function
    **for** $k = 1$ **to** $K$ **do**
        $\mathbf{e} \leftarrow \mathbf{e} + \eta_{\mathbf{e}} \nabla_{\mathbf{e}} \text{KL}\Big(\pi_{\text{ref}}(\cdot \mid \mathbf{e}) \,\big\|\, \pi_\theta(\cdot \mid \mathbf{e})\Big)$
    **end for**
    ▷ Phase II: Distillation Repair
    $\mathcal{L}_{\text{repair}} \leftarrow \text{DistillationAlgorithm}(\pi_\theta, \pi_{\text{ref}}, \mathbf{e})$
    **return** $\mathcal{L}_{\text{repair}}$
**End Function**

---

able. Rather than relying on global regularization, **ALER** proactively localizes and repairs the semantic regions where fine-tuning most degrades prior knowledge.

As shown in Figure 2, **ALER** follows a find–fix protocol. We first adversarially search for latent embeddings that expose high-risk forgetting modes on the data manifold. We then distill from a frozen reference model on these embeddings to selectively restore vulnerable behaviors. The approach complements existing methods and can be integrated as a plug-in retention module.

We next describe each component and provide a theoretical analysis of the efficiency of targeted repair.

### 4.1. Adversarial Search Phase

Our method is built on a simple principle: in data-free continual learning, forgetting should be *found* before it is *fixed*. Domain post-training typically causes *localized* regressions—only a small, long-tailed subset of prior behaviors deteriorates severely—so uniformly regularizing the entire model is both inefficient and misaligned with where damage actually concentrates. We therefore design an explicit *adversarial search* procedure that proactively locates the most vulnerable regions of the prior manifold and compresses them into a small set of latent embeddings, which can be repaired efficiently via online self-distillation in the next phase.

**From forgetting to a computable risk.** Let $\pi_{\theta_t}$ denote the model after $t$ steps of domain post-training, and let

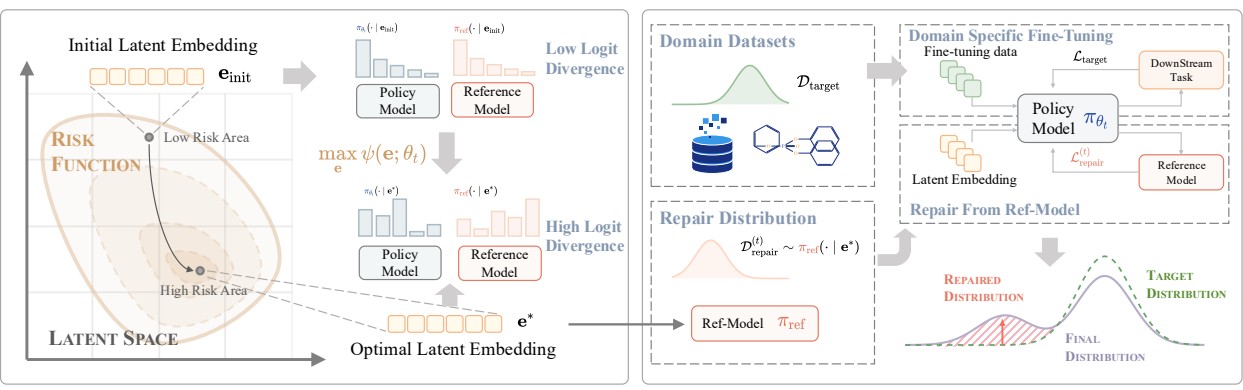

**(1) Adversarial Search Phase**                    **(2) Distillation Repair Phase**

*Figure 2.* Overview of our proposed **ALER** framework. **ALER** follows a find–fix protocol. (Phase I: Find) it optimizes a small set of latent prompt embeddings to maximize the proxy forgetting risk, instantiated as the forward KL divergence between the current policy and a frozen reference. (Phase II: Fix) it performs lightweight online distillation from the reference on the discovered embeddings, selectively repairing the most vulnerable modes while preserving target adaptation.

$\pi_{\mathrm{ref}}$ be a frozen reference initialized from the same pre-trained weights. In Appendix E.2 we show that the expected increase in NLL on pre-training data can be rewritten as an *excess cross-entropy* over pre-training contexts. When the true pre-training conditional is inaccessible, it is natural to approximate it with $\pi_{\mathrm{ref}}(\cdot \mid \mathbf{c})$, since pre-training can be viewed as an optimal compression of the pre-training corpus. Let $\mu_p$ denote the distribution over pre-training contexts induced by drawing a pre-training sequence and token position, i.e., $\mathbf{c} = \mathbf{x}_{<t}$ for $(\mathbf{x}, t)$ from the inaccessible pre-training corpus. This yields the *reference cross-entropy* surrogate

$$\mathcal{R}_{\mathrm{ref}}(\theta) \;=\; \mathbb{E}_{\mathbf{c}\sim\mu_p}\, \mathbb{E}_{y\sim\pi_{\mathrm{ref}}(\cdot|\mathbf{c})}\Big[ -\log\pi_\theta(y \mid \mathbf{c})\Big], \quad (5)$$

whose change $\mathcal{R}_{\mathrm{ref}}(\theta') - \mathcal{R}_{\mathrm{ref}}(\theta)$ serves as a consistent proxy of expected sample-wise forgetting. Moreover, for each context $\mathbf{c}$, $\mathcal{R}_{\mathrm{ref}}$ admits the standard decomposition into an entropy term plus a forward KL:

$$\mathbb{E}_{y\sim\pi_{\mathrm{ref}}(\cdot|\mathbf{c})}\Big[ -\log\pi_\theta(y \mid \mathbf{c})\Big]$$
$$= H\big(\pi_{\mathrm{ref}}(\cdot \mid \mathbf{c})\big) + \mathrm{KL}\Big(\pi_{\mathrm{ref}}(\cdot \mid \mathbf{c}) \,\big\|\, \pi_\theta(\cdot \mid \mathbf{c})\Big), \quad (6)$$

where the entropy term is independent of $\theta$. Consequently, up to a $\theta$-independent constant, minimizing (5) is equivalent to minimizing the forward KL. This motivates using the KL divergence as our *proxy forgetting risk*.

**Definition 4.1** (Proxy forgetting risk)**.** For any conditioning input $\mathbf{z}$ (e.g., a text context or a latent prompt embedding), the *proxy forgetting risk* of $\pi_{\theta_t}$ is

$$\psi(\mathbf{z};\theta_t) \;=\; \mathrm{KL}\Big(\pi_{\mathrm{ref}}(\cdot \mid \mathbf{z}) \,\big\|\, \pi_{\theta_t}(\cdot \mid \mathbf{z})\Big). \quad (7)$$

Definition 4.1 yields an *actionable* and *localized* signal: it assigns a per-input vulnerability score that aligns with the

long-tailed structure of forgetting and compares full predictive distributions rather than a single decoded trajectory. This surrogate is not ad hoc: it is exactly the $\theta$-dependent component of the reference cross-entropy (5) via (6), and thus inherits a clear connection to sample-wise forgetting (Appendix E.2). In addition, the resulting objective admits closed-form gradients with respect to model logits, which makes it suitable for gradient-based adversarial probing.

**Latent embedding adversarial search: compressing vulnerable priors.** Directly searching over discrete text contexts $z = \mathbf{c}$ to maximize (7) is intractable. We therefore introduce a latent prompt embedding $\mathbf{e} \in \mathbb{R}^{\ell \times d}$, treated as a soft prefix that conditions both $\pi_{\mathrm{ref}}$ and $\pi_{\theta_t}$. Instantiating Definition 4.1 with $z = \mathbf{e}$ yields

$$\psi(\mathbf{e};\theta_t) \;=\; \mathrm{KL}\Big(\pi_{\mathrm{ref}}(\cdot \mid \mathbf{e}) \,\big\|\, \pi_{\theta_t}(\cdot \mid \mathbf{e})\Big), \quad (8)$$

and we adversarially search for the most vulnerable latent prompt by

$$\mathbf{e}^\star \;=\; \arg\max_{\mathbf{e}}\ \psi(\mathbf{e};\theta_t). \quad (9)$$

Operationally, at step $t$ we freeze $\theta_t$ and perform a small number of gradient-ascent steps on $\mathbf{e}$ to maximize (8). The resulting $\mathbf{e}^\star$ is a *compact compression* of the model's worst prior drift: it concentrates the most vulnerable pre-trained behaviors into a tiny synthetic carrier, so that the repair budget in the next phase can be spent exactly where forgetting is most severe.

**Two key tricks for effective and stable search.** **(1) Rugged landscape and semantic coverage.** The objective in (9) is rugged, with narrow basins tied to distinct vulnerable semantics, as illustrated in Figure 7 in Appendix C.2; single-prompt optimization typically mode-collapses and misses long-tail failures. We therefore optimize a randomly

initialized *batch* of latent prompts and maximize the *aggregated* batch risk to improve semantic coverage under a fixed compute budget. **(2) Manifold grounding with a semantic constraint.** Unconstrained continuous prompts may drift off the token-embedding manifold and exploit non-linguistic directions, yielding noisy prompts that are difficult to distill. We constrain latent tokens to the vocabulary-embedding manifold (via convex mixtures) and impose a lightweight entropy regularizer to discourage diffuse combinations, producing stable, distillable prompts. Please refer to Appendix B.2 for the full formulation.

In summary, the adversarial search phase converts data-free forgetting diagnosis into an optimization process: it produces a small set of latent prompts $\mathbf{e}^\star$ that maximally exposes the model's worst drift from its pre-trained self, enabling efficient and targeted online repair in the next phase.

### 4.2. Distillation Repair Phase

The adversarial search phase yields $\mathbf{e}^*$, a compact latent probe that concentrates the modes along which the current model $\pi_{\theta_t}$ is most susceptible to forgetting. We now repair these vulnerabilities by distilling the frozen reference $\pi_{\mathrm{ref}}$ back into the student conditioned on $\mathbf{e}^\star$. The key is to recover prior competencies without oversmoothing newly acquired domain-specific modes. Therefore, our default instantiation uses a mode-seeking reverse-KL objective.

**Reverse-KL repair on $\mathbf{e}^\star$.** We minimize the reverse KL on the conditional distribution induced by $\mathbf{e}^\star$:

$$\mathcal{L}_{\mathrm{RKL}} = \mathrm{KL}\Big(\pi_{\theta_t}(\cdot \mid \mathbf{e}^\star) \,\big\|\, \pi_{\mathrm{ref}}(\cdot \mid \mathbf{e}^\star)\Big). \qquad (10)$$

Reverse KL provides a mode-seeking repair objective: it penalizes student mass placed on behaviors that the frozen reference deems implausible, while avoiding the mode-covering pressure of forward KL. This is important during adaptation, where forward KL can over-cover the reference support and dilute sharp probability mass formed on the target domain. We optimize (10) directly on the searched latent prompt, without additional rollout or decoding, since $\mathbf{e}^\star$ is already selected to expose the current student–reference discrepancy and therefore serves as a compact repair signal.

**Compatibility with alternative repair choices.** Our framework separates vulnerability discovery from the choice of repair operator: the adversarial search only produces the carrier $\mathbf{e}^\star$, while the subsequent repair can be instantiated in different ways. Besides the default reverse-KL repair, one can use a forward-KL objective,

$$\mathcal{L}_{\mathrm{FKL}} = \mathrm{KL}\Big(\pi_{\mathrm{ref}}(\cdot \mid \mathbf{e}^\star) \,\big\|\, \pi_{\theta_t}(\cdot \mid \mathbf{e}^\star)\Big), \qquad (11)$$

which encourages the student to cover the reference distribution on the searched carrier. This gives a white-box

alternative when reference likelihoods or logits are available. When only sampling access to the reference is available, we can instead perform black-box repair by decoding samples $\mathbf{x} \sim \pi_{\mathrm{ref}}(\cdot \mid \mathbf{e}^\star)$ and mixing them with the current batch. Thus, the same searched carrier supports both distribution-level and sample-level repair mechanisms.

**Overall training objective.** At each step, we optimize the combined loss

$$\mathcal{L}_t = \mathcal{L}_{\mathrm{target}} + \lambda \, \mathcal{L}_{\mathrm{repair}}, \qquad (12)$$

where $\mathcal{L}_{\mathrm{repair}}$ is instantiated by RKL (default) or by the FKL / black-box variants above, and $\lambda$ controls the retention–adaptation trade-off. Importantly, the repair term is *targeted* by construction: it is evaluated only on $\mathbf{e}^\star$ found at the current step, so it protects the most vulnerable prior behaviors with minimal interference to on-domain optimization.

### 4.3. Theoretical Analysis

Our adversarial search phase explicitly targets conditioning inputs that maximize the proxy forgetting risk $\psi(\mathbf{e}; \theta)$, which is the $\theta$-dependent component of the reference cross-entropy surrogate and hence tracks expected forgetting up to a constant. This yields a principled find-before-fix view: search identifies where the model deviates most from the frozen reference, and repair concentrates distillation exactly on those vulnerable regions rather than spreading effort uniformly or only following the fine-tuning stream. We now summarize the key implication: under a fixed repair budget, allocating repair to a small set of searched high-risk prompts yields a strictly larger reduction in old-knowledge forgetting risk than either (i) **budget-matched random repair** or (ii) **on-batch-only repair**, provided the appropriate structural conditions hold. Operationally, the theory explains *why* targeted repair is efficient: when forgetting is long-tailed, the same budget achieves more by capturing the dominant vulnerable modes; and when the fine-tuning distribution is misaligned with the pre-training distribution, on-batch repair can systematically miss these modes. Full formal statements, constants, and proofs are deferred to Appendix E.3 due to space constraints.

Let $\theta_0$ denote the model parameters at the beginning of a repair step, and let $\theta^+$ denote the parameters after applying that repair update. Let $\theta_*^+$, $\theta_{\mathrm{rand}}^+$, and $\theta_{\mathrm{batch}}^+$ be the post-repair parameters produced by (respectively) targeted repair on searched prompts, random repair, and on-batch-only repair, all under the same repair budget.

**Assumption 4.2** (Finite-budget repair geometry)**.** Consider the population forgetting risk $\mathcal{F}(\theta) = \mathbb{E}_{z \sim \mathcal{D}_p}[\psi(z; \theta)]$ over the (inaccessible) pre-training distribution $\mathcal{D}_p$. We assume: **(1) Local quadratic control** along the short repair paths traversed by the compared methods, $\mathcal{F}$ admits uniform local

quadratic approximation; **(2) Long-tail repairability** the old-risk "repairable energy" is concentrated so that the best budget-$k$ repair directions capture a strictly larger-than-uniform share; **(3) On-batch misalignment** under distribution mismatch, on-batch-induced budget-$k$ repair directions capture only a strict fraction of the best budget-$k$ old-risk energy.

**Proposition 4.3** (Strict dominance under finite budget). *For a suitable stepsize range (explicit in Appendix E.3.5), targeted repair on the searched high-risk prompts satisfies*

$$\mathcal{F}(\theta_*^+) \le \mathbb{E}\big[\mathcal{F}(\theta_{\text{rand}}^+)\big] - c_{\text{rand}}, \qquad (13)$$

$$\mathcal{F}(\theta_*^+) \le \mathcal{F}(\theta_{\text{batch}}^+) - c_{\text{batch}}, \qquad (14)$$

*where $c_{\text{rand}} > 0$ and $c_{\text{batch}} > 0$ are constants that depend on the model, the data distributions, and the repair budget (given explicitly in Appendix E.3).*

**Takeaway.** Proposition 4.3 (proved in Appendix E.3) formalizes the core message of our method: with a fixed repair budget, *where* we spend repair matters. By adversarially discovering a small set of high-risk prompts, ALER concentrates repair on the most vulnerable prior modes, which yields a provable, strict improvement over both uniform random repair and on-batch-only repair whenever forgetting is concentrated and the fine-tuning stream does not faithfully cover the pre-training distribution.

# 5. Experiment

Our experiments consist of five parts: **(1)** experiment setup (§5.1); **(2)** main results (§5.2, adaptation–retention trade-off); **(3)** ablations study (§5.3); **(4)** sensitivity analysis (§5.4); **(5)** training dynamics analysis(§5.5). [1]

## 5.1. Experiment Setup

**Target Datasets** We fine-tune and evaluate our model on two widely used datasets: **(1) SciKnowEval** (Feng et al., 2024), a large-scale benchmark of scientific questions designed to assess scientific knowledge from basic recall to higher-level reasoning. In this work, we use Level L3 questions from the chemistry subset. **(2) FinGPT** (Liu et al., 2023), a collection of finance-related texts annotated with sentiment polarity (and, in some cases, intensity).

**Forgetting Metrics** To quantitatively measure the forgetting of prior knowledge under different fine-tuning methods, we evaluate the fin-tuned models on a suite of general-purpose benchmarks spanning multiple capability dimensions, including **MMLU** (broad knowledge), **MMLU-Pro** (harder knowledge and reasoning), **GSM8K** (math reasoning), **GPQA** (graduate-level QA), **IFEval** (instruction following), and **HumanEval** (code generation) (Hendrycks et al., 2020; Wang et al., 2024c; Cobbe et al., 2021; Rein et al., 2024; Zhou et al., 2023; Chen, 2021). We report the performance drop relative to the pre-fine-tuning baseline on each benchmark as our forgetting score.

**Baseline Setup** We compare our methods against a set of representative and competitive baselines: **(1) TALR** and **(2) PSFT** (Lin et al., 2025; Zhu et al., 2025) represents optimization-centric methods that use forgetting-aware objectives. **(3) SSR** (Huang et al., 2024) uses the reference model to generate self-synthesized rehearsal. **(4) DataMix** directly mixes the fine-tuning data with general-purpose data. **(5) Freeze** (Zheng et al., 2025a) propose a simple yet efficient strategy that freezes the bottom layers of the model. **(6) Orthogonal** (Wang et al., 2023a) mitigates task interference via orthogonal subspace learning.

Moreover, direct generation (referred to as **Naive**), standard supervised fine-tuning **SFT**, and SFT with a standard KL regularization term (referred to as **SFT+KL**) are included as standard reference approaches.

Implementation details (datasets, benchmark protocols, and baseline hyperparameters) are in Appendix B.3.

## 5.2. Main Results

We evaluate continual fine-tuning methods on two target datasets (SciKnowEval and FinGPT) and quantify catastrophic forgetting by measuring post-fine-tuning performance on a suite of general-purpose benchmarks. Fig. 4 summarizes the overall trend, where the $y$-axis is target-domain validation accuracy, and the $x$-axis is the average score on general benchmarks after fine-tuning. Detailed results are reported in Table 1 and Table 4.

Those results highlight four key findings: **(1) Standard fine-tuning induces severe forgetting.** Standard SFT yields large gains on the target task, but causes a substantial drop on the general benchmark suite, confirming that forgetting is a dominant failure mode in domain fine-tuning. Adding a vanilla KL term mitigates the drop, yet cannot eliminate forgetting. **(2) Forgetting is heterogeneous across capability domains.** The degradation is not uniform across benchmarks: knowledge-intensive and harder reasoning evaluations (MMLU/MMLU-Pro/GPQA) typically exhibit larger drops than others, while instruction-following and code-generation benchmarks can degrade in different patterns depending on the target domain. This heterogeneity supports the view that forgetting concentrates on a subset of capabilities, rather than a global performance shift. **(3) Our ALER consistently repairs forgetting while preserving target performance.** Across both target datasets, ALER improves general-capability retention over SFT and representative baselines, while maintaining comparable target-domain

---

[1] Our codes are publicly available at https://github.com/dakfjalka/aler-distill

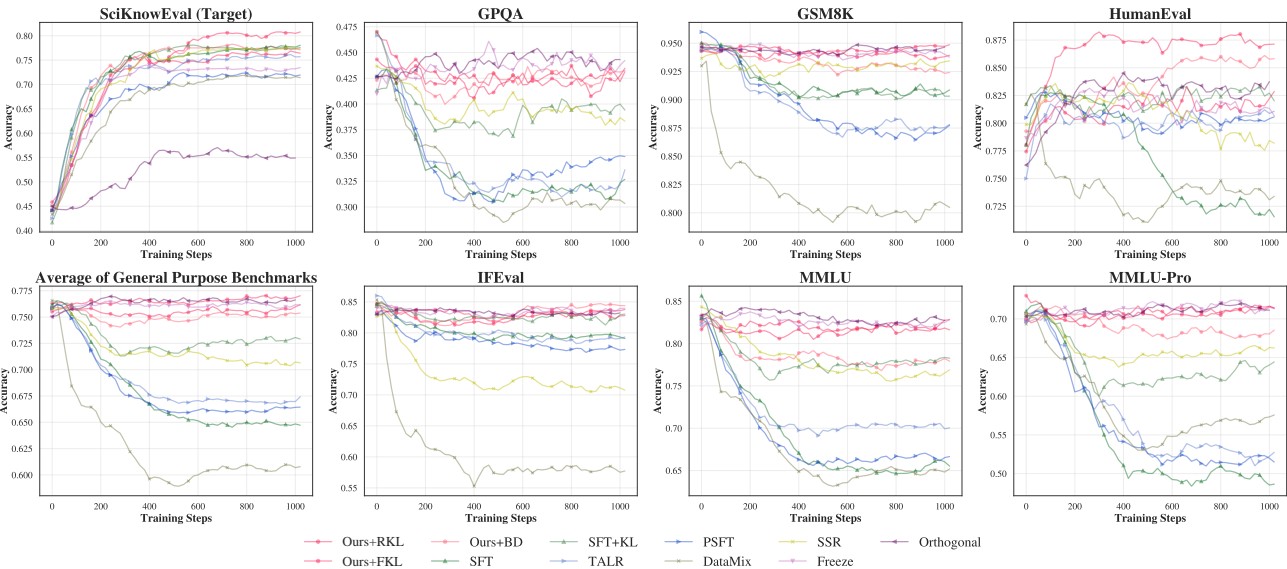

*Figure 3.* Training dynamics on SciKnowEval. We track the accuracy target and on each general-purpose benchmark every 20 steps.

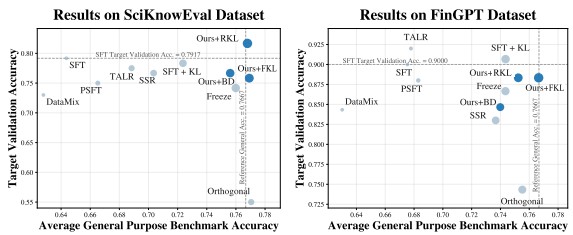

*Figure 4.* Comparison of different methods on SciKnowEval (left) and FinGPT (right). Each point represents a method, where the $x$-axis denotes the average accuracy on general-purpose benchmarks, and the $y$-axis denotes the target-domain validation accuracy. The size of each point is proportional to its hypervolume improvement.

validation accuracy. This indicates that our method performs targeted repair of vulnerable priors rather than merely slowing down optimization. **(4) Generalization tendency via mode-seeking repair.** Beyond the reported benchmarks, **ALER** shows a generalization tendency: by adversarially searching high-risk modes (where the student deviates most from the reference) and by adversarially searching high-risk modes and applying mode-seeking reverse-KL repair, the correction signal is concentrated on compact representatives of long-tail forgotten regions, which can generalize beyond the observed fine-tuning samples.

Beyond the four key findings above, we make several empirical observations that help interpret the design space of anti-forgetting strategies. **(1) Freezing is a simple yet strong baseline,** achieving strong retention with only a modest and acceptable drop in target accuracy (useful when peak target performance is not required). **(2) Naive data mixing is insufficient: DataMix** brings limited gains, likely because uniform mixing poorly targets long-tail forgotten regions and introduces gradient interference under a fixed budget.

**(3) Update-level constraints are stabilizing but blunt: Orthogonal** can preserve some abilities yet often sacrifices target fit, suggesting a mismatch between parameter-space geometry and functional behavior.

### 5.3. Ablation Study

To provide further insights into **ALER**, we implemented three carefully designed ablation experiments by removing the components: **(1) Random Latent Embedding (RE).** We replace the optimized latent embedding $\mathbf{e}^\star$ with a randomly sampled prompt embedding of the same shape and apply the same repair loss. **(2) Random search (RS).** We remove gradient ascent, and instead sample candidate prompts uniformly at random, selecting the one with the largest risk score under the same budget. **(3) Without Manifold Constraint trick (WM).** We optimize $\mathbf{e}$ directly as a free continuous vector. All ablations use the same search budget to ensure fair comparison.

As shown in Table 2, we observe clear performance degradations when removing key components. **RE** leads to a pronounced failure on *both* the target task and general retention, indicating that forgetting mitigation critically depends on discovering informative high-risk prompts rather than distilling from arbitrary latent embeddings. **RS** preserves target accuracy relatively well but exhibits a clear drop in general average, suggesting that budget-matched random search is substantially less effective at locating representative vulnerable modes, and thus yields a weaker repair signal for long-tail priors. **WM** incurs a modest degradation, implying that the manifold constraint improves stability of the search, but is not the primary driver of **ALER** 's gains.

*Table 1.* Main results on SciKnowEval (chemistry L3). We report the target-domain validation accuracy (SciKnowEval) and the post-fine-tuning performance on general benchmarks (GPQA, GSM8K, HumanEval, IFEval, MMLU, MMLU-Pro). Numbers in parentheses denote the performance accuracy scores; **Avg** is the average across the general benchmarks.

| Method | SciKnowEval (Target) | GPQA | GSM8K | HumanEval | IFEval | MMLU | MMLU-Pro | Avg |
|---|---|---|---|---|---|---|---|---|
| Naive | 41.67 (Ref) | 44.33 | 94.67 | 81.71 | 84.67 | 83.33 | 71.33 | 76.67 (Ref) |
| *Standard SFT* | | | | | | | | |
| SFT | 79.17 (+37.50) | 33.67 | 91.67 | 69.51 | 78.33 | 64.00 | 49.00 | 64.36 (-12.31) |
| SFT + KL | 78.33 (+36.66) | 37.67 | 90.33 | 79.88 | 83.33 | 78.00 | 65.00 | 72.37 (-4.30) |
| *Baselines* | | | | | | | | |
| TALR | 77.50 (+35.83) | 38.00 | 88.67 | 81.71 | 80.00 | 70.33 | 54.33 | 68.84 (-7.83) |
| PSFT | 75.00 (+33.33) | 34.67 | 88.67 | 81.10 | 77.67 | 67.00 | 50.00 | 66.52 (-10.15) |
| SSR | 76.67 (+35.00) | 37.33 | 93.67 | 77.44 | 69.67 | 78.00 | 66.00 | 70.35 (-6.32) |
| DataMix | 73.00 (+31.33) | 29.33 | 79.67 | 74.39 | 67.66 | 67.00 | 58.67 | 62.79 (-13.88) |
| Freeze | 74.17 (+32.50) | 45.67 | 93.00 | 78.66 | 84.00 | **84.00** | 70.67 | 75.99 (-0.68) |
| Orthogonal | 55.00 (+13.33) | 44.33 | 95.00 | **85.98** | 83.67 | 82.33 | **71.00** | **77.05** (+0.38) |
| *Ours* | | | | | | | | |
| **Ours+RKL** | **81.67** (+40.00) | 45.67 | 94.33 | 84.76 | **84.33** | 81.00 | 70.67 | 76.79 (+0.12) |
| **Ours+FKL** | 75.83 (+34.16) | **46.00** | **95.33** | 83.53 | 83.00 | 83.00 | 70.67 | 76.92 (+0.25) |
| **Ours+BD** | 76.67 (+35.00) | 43.67 | 92.67 | **85.98** | **84.33** | 77.00 | 70.00 | 75.61 (-1.06) |

*Table 2.* Ablation results on SciKnowEval (chemistry L3). Numbers in parentheses denote changes relative to **Ours+RKL**.

| Method | SciKnowEval (Target) | General Avg |
|---|---|---|
| **Ours+RKL** | **81.67** (Ref) | **76.79** (Ref) |
| RE | 72.50 (-9.17) | 67.11 (-9.68) |
| RS | 79.16 (-2.51) | 73.80 (-2.99) |
| WM | 80.00 (-1.67) | 75.60 (-1.19) |

### 5.4. Sensitivity Analysis

The adversarial search phase of **ALER** is primarily controlled by two hyperparameters: the latent-embedding length $L$, which sets the representational capacity of each latent prompt, and the latent-embedding batch size $B$, which governs the breadth of prior regions explored per update. Given the highly non-convex and irregular search landscape (Figure 7), increasing $B$ increases the likelihood of visiting diverse regions and reaching high-risk local maxima. Together, $L$ and $B$ regulate the repair signal through complementary axes of *capacity* and *coverage*.

To isolate their effects, we perform **one-factor-at-a-time sweeps**. We vary $L \in \{1, \ldots, 64\}$ while keeping all other settings fixed (optimizer, learning-rate schedule, search budget, and model architecture), and separately vary $B \in \{1, \ldots, 64\}$ under an identical configuration. This design

attributes observed performance changes solely to the swept hyperparameter. Due to limited space, we defer the results to the Table 7 and Table 6 in Appendix C.5.

Table 7 and Table 6 show a consistent unimodal trend for both $L$ and $B$: performance improves from small values, peaks at an intermediate regime, and then degrades as either hyperparameter grows further. The best performance is achieved at $L = 8$ and $B = 8$, indicating that effective repair requires balancing expressiveness with exploration. We interpret the initial gains as increased ability to capture informative, high-risk modes of the corrupted prior. Beyond the optimum, larger $L$ or $B$ admits more spurious, high-variance adversarial solutions, causing noisy latent prompts to be distilled into the student; this amplifies optimization instability and ultimately harms prior retention.

### 5.5. Visualization Analysis

To intuitively illustrate training dynamics, we track target validation accuracy and general-benchmark performance every 20 fine-tuning steps. As illustrated in Figure 3, SFT improves the target quickly but incurs an early, steep drop on general benchmarks that later plateaus; **ALER** maintains similar target convergence while substantially flattening the general-performance decay, especially on knowledge/reasoning benchmarks. **ALER** acts as an online stabilization mechanism that prevents early drift of vulnerable priors

rather than relying on late-stage recovery.

## 6. Conclusion and Limitations

In this paper, we propose **ALER**, a plug-and-play self-distillation framework that actively discovers high-risk latent prompts—where the student deviates most from a frozen reference—and performs online repair mode-seeking distillation. Experiments show that **ALER** improves retention while preserving target performance compared with a suite of baselines. However, several limitations remain. Our current experiments focus on domain-specific post-training with point-wise latent prompt search, and do not directly evaluate extremely long-context continual learning or long-range dependency retention. Forgetting is heterogeneous and long-tailed; **ALER** currently lacks capability-aware control over *what* to repair and *how strongly* to protect different skills, suggesting finer-grained, skill-adaptive repair as a direction. The latent-prompt objective is highly non-smooth, and our gradient-ascent search can be inefficient or unstable; developing more principled optimization and exploration strategies is an important next step. Finally, the inner-loop search introduces additional overhead, and black-box decoded repair can add rollout cost; adaptive scheduling and selective repair may improve efficiency.

## Acknowledgement

This work was supported by the National Natural Science Foundation of China, grant 624B1011, U23A20388, and 62021001, and in part by the National Key R&D Program of China under contract 2022ZD0119801. This work was supported in part by Huawei as well. We would like to thank our lab-mate, Hongyu Liu, for his valuable discussions and constructive feedback throughout this project. We would like to thank all the anonymous reviewers for their insightful comments.

## Impact Statement

This paper presents work whose goal is to advance the field of Machine Learning. There are many potential societal consequences of our work, none of which we feel must be specifically highlighted here.

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

# A. Background and Related Work

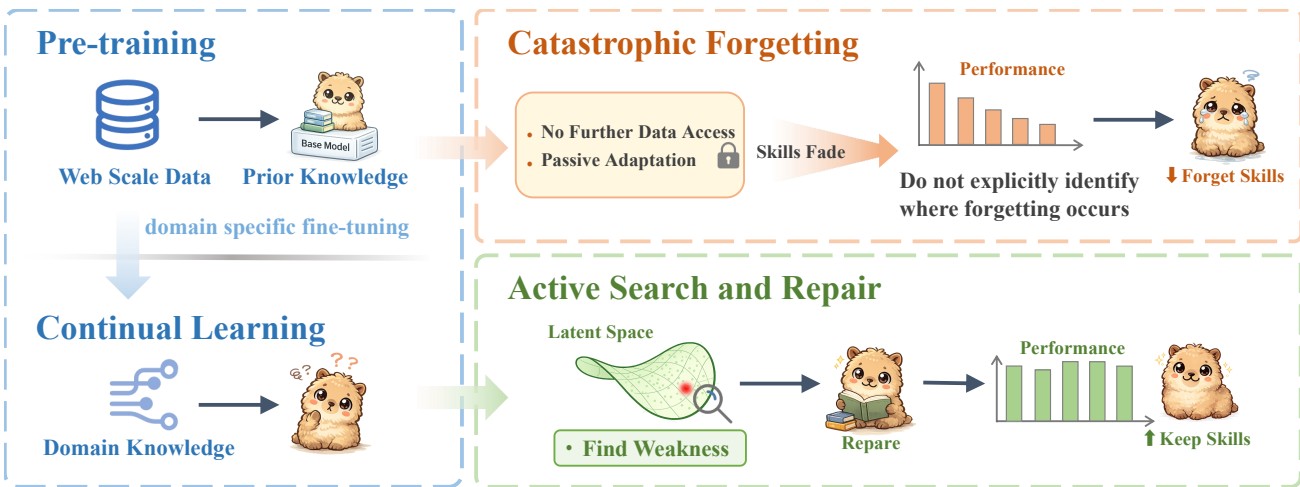

*Figure 5.* **Motivation and high-level idea.** In continual post-training, the model adapts to a new domain without access to the original web-scale data, which can cause catastrophic forgetting of prior skills (top). Our approach performs active search and repair by identifying vulnerable regions in a latent space and applying targeted repair, improving retention while adapting to the domain (bottom).

**Context.** Figure 5 summarizes the continual post-training setting and highlights why forgetting can be localized and hard to diagnose under passive adaptation, motivating active search-and-repair strategies.

**Continual learning for LLMs: overview and trajectory.** As LLMs are deployed in rapidly evolving environments, practitioners increasingly require *continual post-training*—continually incorporating new knowledge, tasks, or preferences without repeatedly pretraining from scratch. Recent surveys propose LLM-specific taxonomies that mirror the modern training pipeline, typically organizing methods into continual pretraining, continual instruction tuning, and continual alignment, while also contrasting continual learning with alternative update paradigms such as retrieval augmentation and model editing (Wu et al., 2024; Shi et al., 2025; Zheng et al., 2025b; Yang et al., 2025). Complementing these overviews, benchmark efforts (e.g., TRACE for aligned LLMs and CITB for continual instruction tuning) expose substantial regressions in general capability and instruction-following under sequential updates (Wang et al., 2023b; Zhang et al., 2023). Mechanistic and empirical analyses further suggest that observed "forgetting" can stem from shifts in task inference/alignment or optimization geometry, motivating diagnostics beyond aggregate accuracy (Kotha et al., 2024; Li et al., 2024; Zheng et al., 2025a). This line of work collectively motivates methods that can *localize* where degradation concentrates and *repair* it with minimal collateral drift.

**Regularization and Constrained Optimization.** Continual learning often stabilizes sequential updates via parameter-importance regularization or explicit constraints on updates, e.g., EWC/SI/MAS and distillation-style retention (Kirkpatrick et al., 2017; Zenke et al., 2017; Aljundi et al., 2018; Li & Hoiem, 2017). Constraint/projection formulations further enforce non-increase of past-task loss, such as GEM/A-GEM (Lopez-Paz & Ranzato, 2017; Chaudhry et al., 2018). In LLM post-training, trust-region style control is common through KL-to-reference constraints (PPO/TRPO and RLHF) and related preference optimization views (Schulman et al., 2017; 2015; Hao et al., 2025). Adversarial objectives can be viewed as worst-case regularization (Madry et al., 2017), and efficient NLP adversarial training often uses continuous (embedding-space) perturbations such as FreeLB/ALUM and recent continuous attacks for LLM robustness/jailbreak (Zhu et al., 2019; Liu et al., 2020; Xhonneux et al., 2024). STAR searches for worst-case parameter perturbations to induce stable loss basins (Eskandar et al., 2025). In contrast, **ALER** searches in the prompt/latent-embedding space to identify functional regions where the current model diverges most from a frozen reference, and repairs those regions by targeted distillation. Thus, ALER differs from STAR in both search space and repair mechanism.

**Data Selection and Token-Level Objectives.** Forgetting can also be mitigated by controlling training data composition and granularity of reweighting, including domain-adaptive/continued pretraining and training-dynamics guided selection (Gururangan et al., 2020; Swayamdipta et al., 2020). Recent LLM work further moves from global mixing heuristics toward predicting which upstream behaviors/examples are at risk of being forgotten, and toward token-level loss shaping during refinement (Jin & Ren, 2024b; Lin et al., 2025).

**Replay and Synthetic (Generative) Rehearsal.** Replay interleaves historical exemplars or pseudo-samples to approximate earlier distributions, including exemplar replay (iCaRL), constraint-based replay (GEM/A-GEM), and generative rehearsal (DGR) (Rebuffi et al., 2017; Lopez-Paz & Ranzato, 2017; Chaudhry et al., 2018; Shin et al., 2017). In LLM continual/instruction tuning, synthetic rehearsal, and curated replay are widely used and increasingly benchmarked under realistic streams (Huang et al., 2024; Wang et al., 2023b; Zhang et al., 2023). DER stores past logits in a replay buffer and distills the current model toward these stored dark targets (Buzzega et al., 2020). It is effective when past data or logits can be retained, but it requires rehearsal memory and does not match our pre-training-data-inaccessible setting. ALER instead uses a frozen reference model and searches latent prompts, requiring no replay buffer of pre-training data or stored logits.

**Parameter Isolation and Parameter-Efficient Continual Adaptation.** Parameter-isolation methods mitigate interference by allocating task-specific subnetworks, masks, or expandable capacity (e.g., progressive columns, dynamic expansion, pruning/masking, and hard task attention) (Rusu et al., 2016; Fernando et al., 2017; Yoon et al., 2017; Mallya & Lazebnik, 2018; Mallya et al., 2018). In the PEFT regime, continual adaptation is localized to lightweight modules such as adapters/LoRA or soft prompts/prefixes on a frozen backbone (Houlsby et al., 2019; Hu et al., 2022; Li & Liang, 2021; Lester et al., 2021), and prompt/module pools further enable rehearsal-free continual learning via selecting or composing small task-conditioned parameters (Wang et al., 2022b;a; Smith et al., 2023; Wang et al., 2024a). Recent LLM/VLM post-training pushes this modularity toward routing and MoE-style expertization for retention–transfer trade-offs, spanning MoE adapters and self-expanding adapter mixtures (Wang et al., 2025b), MoE instruction tuning as a general MoE+alignment context (Shen et al., 2023), and industrial-scale continual instruction tuning with adversarially-regularized LoRA experts (Kang et al., 2025).

**Analyses of Forgetting in LLMs.** A growing body of work studies how and why LLMs forget during sequential post-training. TRACE provides a comprehensive continual learning benchmark for aligned LLMs and reports severe drops in general ability and instruction-following after sequential tuning (Wang et al., 2023b). CITB introduces evaluation protocols and task streams for continual instruction tuning, showing that common CL techniques do not automatically translate to instruction-tuned settings (Zhang et al., 2023). Beyond benchmarks, mechanistic analyses connect forgetting to implicit task inference shifts (Kotha et al., 2024), loss-landscape geometry (flatness/sharpness) (Li et al., 2024), and even "spurious" forgetting caused by task misalignment rather than true knowledge erasure (Zheng et al., 2025a). Recent evidence further clarifies the role of *training data distribution* during post-training: systematic comparisons of SFT and RL across model families and tasks find RL consistently incurs less forgetting, attributing the robustness primarily to the use of (approximately) on-policy data rather than auxiliary design choices (Chen et al., 2025a). Complementarily, a data-perspective study of multimodal post-training reports that reinforcement fine-tuning better preserves prior knowledge on Qwen2.5-VL; the learning-dynamics analysis suggests RFT tends to reinforce samples better aligned with the base model's probability landscape, and that using RFT-generated rollouts as supervised data can substantially reduce forgetting while retaining fast acquisition (Zhang et al., 2025). Together, these analyses motivate fine-grained diagnostics and localized repair targets, especially in data-free or limited-data continual post-training.

**Model Merging and Weight-Space Fusion.** When historical data are unavailable, another complementary direction is *weight-space integration*, which fuses multiple specialized models without additional training data. General model merging approaches such as model soups (weight averaging across fine-tuned checkpoints) (Wortsman et al., 2022) and interference-aware merging (e.g., TIES-Merging) (Yadav et al., 2023) provide practical recipes for consolidating multiple adapted models. Building on representation alignment, RECALL proposes a hierarchical model merging framework that computes inter-model similarity via layer-wise hidden representations over clustered "typical" samples, then performs adaptive parameter fusion to preserve domain-general shallow features while integrating task-specific deeper representations (Wang et al., 2025a). Model merging offers a strong data-free alternative to rehearsal, and it is complementary to input-space localization approaches.

**Adjacent update paradigms: editing, unlearning, and watermarking.** Beyond continual learning, several adjacent paradigms study targeted behavior change with minimal side effects. Knowledge/model editing aims to update specific facts or behaviors without degrading unrelated capabilities; recent surveys systematize formulations, taxonomies, and evaluation protocols for LLM editing (Wang et al., 2024b). Machine unlearning targets the *removal* of specific data influence, motivated by privacy and compliance; recent LLM-specific surveys provide taxonomies and threat models for unlearning evaluation (Qiu et al., 2025; Le-Khac & Truong, 2025). Adversarial assessment-and-repair has also been explored in unlearning robustness: dynamic adversarial suffix attacks can recover supposedly unlearned knowledge, and latent adversarial unlearning frameworks improve robustness via a min–max formulation (Yuan et al., 2025). Finally, watermarking work that injects fictitious knowledge studies how embedded traces persist through continual pretraining and supervised

fine-tuning, providing an orthogonal lens on post-training retention and verification (Cui et al., 2025). These directions are closely related in spirit (targeted behavior control), but differ in objectives (retain vs edit vs erase vs trace).

## B. Implementation Details

### B.1. Implementation Details of Motivation Experiments

**Experiment settings** We first generated an HMM with 5 hidden states and 8 vocabulary size to generate the pre-training dataset $\mathcal{D}_p$ with 40,000 samples. The size of the vocabulary $\mathcal{V}$ is set to 8. We pre-train a lightweight transformer with 2 layers and 4 attention heads, with a model dimension of 128 for 200,000 steps. Then, we use a sub-structure of the pre-training HMM to generate 1,000 fine-tuning data, and fine-tune the pre-trained model $\pi_\theta$ with the standard SFT process for 500 steps. This results in a new model $\pi_{\theta'}$.

### B.2. Implementation Details of ALER

**Probe suffix.** To make the searched latent prompt $\mathbf{e}^\star$ immediately "actionable" by the LM (without requiring a real pre-training sample), we append a fixed, human-readable suffix before the soft prompt when querying either the student or the frozen reference. Throughout our experiments, we use the following probe suffix: *Answer the next question briefly and correctly.*

**Where distillation gradients flow (token-selective vs. sequence-wide).** A key engineering choice is *token scope*: whether the repair loss backpropagates through the entire sequence (full teacher forcing) or only through the generated content. In **ALER**, **the default repair is token-selective and does *not* apply sequence-wide KL**. Concretely, for both Ours+FKL (forward KL) and the analytic form of Ours+RKL (reverse KL), we compute distillation on the **next-token distribution only** under the probing context $\mathbf{c}$:

$$\mathcal{L}_{\text{FKL}} \ = \ \mathrm{KL}\big(\pi_{\text{ref}}(\cdot \mid \mathbf{c}) \,\|\, \pi_\theta(\cdot \mid \mathbf{c})\big), \qquad \mathcal{L}_{\text{RKL}} \ = \ \mathrm{KL}\big(\pi_\theta(\cdot \mid \mathbf{c}) \,\|\, \pi_{\text{ref}}(\cdot \mid \mathbf{c})\big),$$

where both KL terms are evaluated at a **single step** (the last position of $\mathbf{c}$). Thus, repair gradients are induced solely by aligning (or mode-seeking against) the one-step predictive distribution, instead of propagating through a long target sequence. This design keeps the repair lightweight and reduces interference with target-domain optimization.

**Ours+BD (black-box decoded replay) and gradient masking.** For black-box repair, we decode a continuation $\mathbf{y} \sim \pi_{\text{ref}}(\cdot \mid \mathbf{c})$ from the frozen reference and treat it as replay data. Training on replay uses standard teacher forcing *only on the generated continuation*: the probe suffix tokens are **masked out** (ignored) in the loss, while the generated tokens are used as labels. Hence, gradients flow through the model parameters via $\log \pi_\theta(\mathbf{y} \mid \mathbf{c})$, but **only for the continuation token positions**. This makes BD explicitly "generated-content-only" in terms of gradient propagation, while remaining fully black-box w.r.t. the decoding process.

**Two key tricks for effective and stable search.** **(1) Rugged landscape and semantic coverage.** Empirically, the objective in (9) is highly rugged, with many narrow basins corresponding to different vulnerable semantics. Optimizing a single prompt can easily overfit to one basin and miss other long-tail failure modes. To improve coverage with minimal complexity, we optimize a *batch* of latent prompts with *random initialization* and maximize the aggregated risk across the batch. Intuitively, increasing the batch size raises the probability of exploring diverse basins, thereby capturing multiple forgetting modes under a small compute budget. **(2) Manifold grounding with a semantic constraint.** Unconstrained continuous prompts may drift off the token-embedding manifold and exploit adversarial directions that do not correspond to plausible linguistic states, producing $\mathbf{e}^\star$ that behaves like noise and is difficult to distill. To ground the search in realistic semantics, we constrain each latent token to lie in the convex hull spanned by the vocabulary embedding matrix $E \in \mathbb{R}^{V \times d}$. For each latent position $j \in \{1, \ldots, \ell\}$, we parameterize $\mathbf{e}_j = \sum_{i=1}^{V} \alpha_{j,i} E_i$, $\boldsymbol{\alpha}_j = \text{softmax}(\mathbf{w}_j)$, where $\mathbf{w}_j \in \mathbb{R}^V$ are free logits and $\boldsymbol{\alpha}_j$ lies on the simplex. To further prevent adversarial search from degenerating into unstructured mixtures, we add a lightweight *semantic constraint* via the (negative) entropy of $\boldsymbol{\alpha}$: $\mathcal{L}_{\text{reg}} = -\sum_{j=1}^{\ell} \sum_{i=1}^{V} \alpha_{j,i} \log \alpha_{j,i}$. This encourages stable, meaningful mixtures on the token manifold and yields latent prompts that are more robust and more amenable to downstream distillation.

## B.3. Implementation Details of Main Experiments

### B.3.1. TARGET DOMAIN-SPECIFIC FINE-TUNING DATASETS SETUP

We perform domain-specific post-training on two target datasets: **SciKnowEval** (chemistry subset, Level L3) and **FinGPT** (finance sentiment), following the standard formulation of supervised fine-tuning on instruction–response pairs.

**SciKnowEval (Chemistry, L3).** SciKnowEval (Feng et al., 2024) is a multi-level scientific QA benchmark spanning basic recall to higher-order reasoning. We use *Level L3* questions from the *chemistry* subset as our target-domain data, where each example is formatted as a prompt (question) paired with a completion (reference answer) for SFT.

**FinGPT (Finance Sentiment).** FinGPT (Liu et al., 2023) is a collection of finance-related texts annotated with sentiment polarity (and, in some cases, intensity). We cast each instance into an instruction-style prompt describing the input text and the prediction task, and use the annotated label/text as the supervised completion.

**Train/Validation split and isolation.** For each target dataset, we *hold out a fixed subset* as a validation set for the target domain performance evaluation. The validation set is **strictly disjoint** from the training set at the example level (no shared instances, prompts, or duplicated texts), ensuring that all reported target-domain performance reflects genuine generalization rather than leakage. Unless otherwise stated, we report *target-domain validation accuracy* as the adaptation metric throughout the main experiments.

### B.3.2. GENERAL PURPOSE BENCMARKS SETUP

To quantify *retention* of broad capabilities after domain post-training, we evaluate on six general-purpose benchmarks that cover knowledge, reasoning, math, instruction following, and code generation. We follow each benchmark's standard evaluation protocol and report its primary metric (accuracy or pass@1). For the larger benchmarks, we use the *first 300 instances* in the released evaluation split as our test subset for efficiency and consistent comparison; benchmarks with fewer than 300 instances are evaluated in full.

**MMLU.** Massive Multitask Language Understanding (MMLU) (Hendrycks et al., 2020) is a multiple-choice test spanning **57** subjects (STEM, humanities, social sciences, and professional topics), designed to measure broad world knowledge and problem-solving.

**MMLU-Pro.** MMLU-Pro (Wang et al., 2024c) is a more challenging and robust variant of MMLU, featuring more reasoning-focused questions, removing noisy/trivial items, and expanding the multiple-choice set from **4 to 10** options to reduce guessing.

**GPQA.** GPQA (Rein et al., 2024) is a "Google-proof" graduate-level multiple-choice QA benchmark consisting of **448** expert-written questions in **biology, physics, and chemistry**, targeting difficult domain knowledge and scientific reasoning.

**GSM8K.** GSM8K (Cobbe et al., 2021) is a grade-school math word-problem dataset with **8.5K** high-quality problems (with a standard test split), emphasizing multi-step arithmetic reasoning.

**IFEval.** Instruction-Following Eval (IFEval) (Zhou et al., 2023) evaluates whether a model follows *verifiable* natural-language instructions (e.g., length constraints, keyword constraints, formatting requirements). It includes **25** instruction types and roughly **500** prompts, each containing one or more verifiable constraints.

**HumanEval.** HumanEval (Chen, 2021) is an execution-based code generation benchmark with **164** hand-crafted Python programming problems; solutions are judged by functional correctness via unit tests (commonly reported as pass@1).

### B.3.3. BASELINE SETUP

To rigorously evaluate the effectiveness of our approach, we benchmark it against a diverse suite of representative and competitive baselines: (1) **TALR** and (2) **PSFT** (Lin et al., 2025; Zhu et al., 2025) represent optimization-centric approaches that refine the training objective to be forgetting-aware. (3) **SSR** (Huang et al., 2024) leverages the reference model to generate self-synthesized rehearsal data. (4) **DataMix** directly mixes the fine-tuning data with general-purpose data. (5) **Freeze** (Zheng et al., 2025a) proposes a simple yet efficient strategy that freezes the bottom layers of the model. (6) **Orthogonal** (Wang et al., 2023a) mitigates task interference via orthogonal subspace learning.

**Token-Adaptive Loss Reweighting (TALR)** (Lin et al., 2025) addresses the trade-off between domain adaptation and general capability retention. Derived as the *closed-form solution to a constrained optimization problem*, it dynamically reweights the loss at the token level. Specifically, it down-weights the loss for tokens where the fine-tuning model's probability diverges significantly from the pre-trained reference model ($w_i^* \propto p_\theta(x_i)^{1/\tau}$), thereby focusing updates on tokens aligned with prior knowledge. We set $\tau = 1.0$ in our implementation.

**Proximal Supervised Fine-Tuning (PSFT)** (Zhu et al., 2025) incorporates a trust-region constraint—inspired by reinforcement learning—into the supervised fine-tuning process. It prevents the model's policy from deviating excessively from the reference policy by clipping the probability ratio $r_t(\theta) = \pi_\theta(y|x)/\pi_{\text{ref}}(y|x)$:

$$\mathcal{L}_{\text{PSFT}}(\theta) = \mathbb{E}\left[\min\left(r_t(\theta), \text{clip}(r_t(\theta), 1 - \epsilon, 1 + \epsilon)\right)\right] \tag{15}$$

This constraint ensures that the fine-tuned model remains within a safe neighborhood of the original model, preserving general utility while adapting to the downstream task. We set the clip ratio $\epsilon$ to $0.2$ as suggested in the original paper.

**Self-Synthesized Rehearsal (SSR).** SSR (Huang et al., 2024) mitigates forgetting by generating rehearsal examples from a reference model and mixing them with target-domain fine-tuning. Concretely, we synthesize a candidate pool via ICL-style prompting and keep a fixed budgeted subset for training (up to 1,000 rehearsal examples per run), with the candidate pool sized as a small constant multiple of the final selected set ($3\times$ candidates) to encourage diversity. We use a short generation budget for synthesis and refinement (256 new tokens) and standard nucleus sampling (temperature 0.9, top-$p$ 0.9). Following our appendix description, rehearsal and target data are mixed at a 1:4 ratio (rehearsal:target) under the same total step budget as SFT. For reproducibility and efficiency, rehearsal generations are cached under the run directory and reused across ranks when available.

**DataMix** DataMix is a simple rehearsal baseline that interleaves target-domain training data with a general-purpose instruction corpus (we use `tatsu-lab/alpaca` (Taori et al., 2023) in our runs). We keep the mixing mild (general:target = 1:4) and implement it by probabilistic interleaving at the dataset level, so each batch is drawn from either source with fixed probabilities derived from the mixing ratio. To avoid increasing compute and to keep the overall budget comparable, we subsample the general dataset to a small fixed size (570 examples) and stop when the first dataset is exhausted. All other optimization and evaluation settings match the standard SFT configuration.

**Freeze Strategy (Freeze)** (Zheng et al., 2025a) preserves low-level representations by freezing the bottom transformer blocks (and embeddings) during fine-tuning. In our experiments, we freeze the bottom 6 layers and the input embedding matrix, as suggested by freezing 20% of the layers in the original paper. We follow a "paper-style" freezing protocol: frozen parameters remain in the model, and we optionally mask their gradients before gradient clipping to avoid affecting the global norm.

**Orthogonal Subspace Learning (Orthogonal)** We implement an orthogonality baseline (Wang et al., 2023a) as a lightweight regularizer for LoRA fine-tuning, encouraging the trainable LoRA update to be orthogonal to the frozen base weight. For a LoRA-wrapped layer $W = W_0 + \Delta W$ with $\Delta W = s \cdot (BA)$, we add

$$\mathcal{L}_{\text{orth}} = \left(\frac{\langle W_0, \Delta W \rangle_F}{\|W_0\|_F \|\Delta W\|_F + \varepsilon}\right)^2, \tag{16}$$

yielding the overall objective $\mathcal{L} = \mathcal{L}_{\text{SFT}} + \lambda_{\text{orth}}\mathcal{L}_{\text{orth}}$. To avoid materializing $\Delta W$ for large matrices, we estimate the Frobenius inner product by uniformly sampling entries of $W_0$ and the corresponding $\Delta W$ (computed from $A, B$), using a sample size of 2048 in our experiments.

### B.3.4. EXPERIMENT SETUP

We use Qwen3-4B-Instruct-2507 (Team, 2025) as the main model for our experiments. We select this model because it achieves exceptional performance on a wide range of general-purpose benchmarks while maintaining a lightweight parameter footprint. This high baseline capability makes it an ideal candidate for rigorously quantifying the extent of catastrophic forgetting during domain-specific adaptation.

Regarding the optimization configuration, we maintain a consistent setup across all experiments to ensure a fair comparison. We utilize the **AdamW** (Loshchilov & Hutter, 2017) optimizer with a learning rate of $4 \times 10^{-5}$ decayed via a cosine scheduler, with a warmup ratio of 0.03. The global batch size is set to 32, and all models are fine-tuned for a total of $1,000$ steps. Additionally, we apply gradient clipping with a maximum gradient norm of 1.0 to stabilize the training process. Regarding the implementation details, we establish our training pipeline using the **TRL** library (von Werra et al., 2020) to ensure robustness and scalability. For the evaluation phase, we employ the **SGLang** (Zheng et al., 2024) inference engine, leveraging its optimized RadixAttention mechanism to accelerate the generation process. All configuration parameters for SGLang are maintained in accordance with its official default settings. Furthermore, for all generation tasks, we adopt a Chain-of-Thought (CoT) prompting strategy by appending "Please think step by step" to the input, facilitating a more rigorous assessment of the model's reasoning capabilities.

The specific hyperparameters for both the generation process and our proposed method are detailed in Table 3. For the generation configuration, we maintain a consistent setup across all rollouts to ensure a fair comparison. Regarding our method's core components, the trust region parameters—including prompt length and learning rate—are specifically tuned to balance stability and adaptation speed. Notably, we set the weight of the repair loss ($\lambda$) to 1.0, thereby treating the repair objective with equivalent importance to the target batch optimization.

The key hyperparameters are summarized in Table 3.

*Table 3.* Hyperparameter settings for the main experiments.

| Category | Hyperparameter | Value |
|---|---|---|
| General Training | Base Model | Qwen3-4B-Instruct-2507 |
| | Optimizer | AdamW |
| | Learning Rate | $4 \times 10^{-5}$ |
| | Global Batch Size | 32 |
| | Training Steps | 1,000 |
| | Warmup Ratio | 0.03 |
| | LR Scheduler | Cosine |
| | Max Grad Norm | 1.0 |
| | Random Seed | 42 |
| **ALER** | Prompt Length | 8 |
| | Prompt Batch Size | 8 |
| | Prompt Learning Rate | 0.2 |
| | Repair Weight ($\lambda$) | 1.0 |
| | Rollout sequence Length (BD only) | 256 |
| | Replay Data: Domain Data (BD only) | 1:4 |
| Evaluation | Inference Engine | SGLang |
| | Max New Tokens | 2,048 |
| | Temperature | 0.7 |
| | Top-$p$ | 0.8 |

## C. More Experiment Results

### C.1. Empirical Validation: Correlation between Risk Function and True Forgetting

To empirically validate the alignment between our proposed KL-based risk surrogate $\psi(\mathbf{x}; \theta_t)$ and the ground-truth sample-wise forgetting (measured by $\Delta\text{NLL}(\mathbf{x})$), we conducted a quantitative correlation analysis. Utilizing the controlled HMM experimental setup described in Section 3 , we tracked both metrics for all pre-training samples after fine-tuning.

As illustrated in Figure 6, the hexbin density plot reveals a strong, positive linear relationship between the proxy risk and the actual forgetting. Specifically, samples exhibiting high $\Delta\text{NLL}$ (severe forgetting) are consistently mapped to high $\psi(\mathbf{x}; \theta_t)$ values. We calculated the Pearson correlation coefficient to be **0.909**, indicating a high degree of statistical dependence.

This strong correlation empirically justifies our design choice: it confirms that maximizing the computable proxy $\psi(\mathbf{x}; \theta_t)$ effectively localizes the samples suffering from the most severe knowledge degradation, even when the ground-truth pre-training labels are inaccessible.

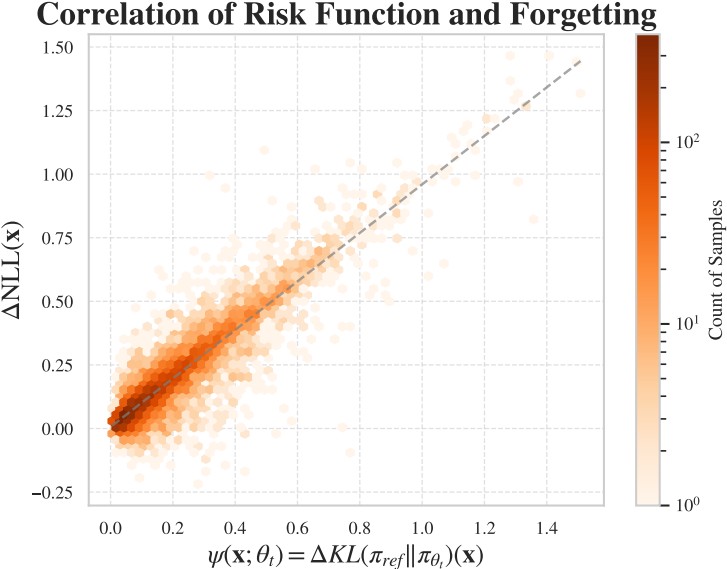

*Figure 6.* **Correlation between the Proxy Risk and True Forgetting.** We visualize the relationship between the proposed risk function $\psi(\mathbf{x}; \theta_t) = \Delta KL(\pi_{ref} || \pi_{\theta_t})(\mathbf{x})$ (x-axis) and the ground-truth sample-wise forgetting $\Delta \mathrm{NLL}(\mathbf{x})$ (y-axis) on the pre-training set. The hexbin plot shows a tight alignment along the diagonal with a Pearson correlation of $\rho \approx 0.91$, validating $\psi$ as a reliable surrogate for detecting forgetting.

## C.2. Landscape of Risk Function

In this section, to provide an intuitive view of the risk function $\psi(e; \theta_t)$ defined in Eq. (8), we visualize the optimization landscape during the adversarial search phase using the filter-normalization method introduced in (Li et al., 2018). This visualization technique allows us to project the high-dimensional latent embedding space onto a 2D grid defined by the gradient direction and a random orthogonal direction, revealing the rugged nature of the forgetting risk.

Specifically, we center the visualization at a randomly initialized latent embedding $\mathbf{e}_0$. We then construct a two-dimensional projection plane defined by two basis vectors: (1) $\mathbf{u}$: The normalized gradient direction of the risk function, $\mathbf{u} = \frac{\nabla_{\mathbf{e}} \psi(\mathbf{e}; \theta_t)}{||\nabla_{\mathbf{e}} \psi(\mathbf{e}; \theta_t)||}$, which represents the direction of steepest ascent in forgetting risk. (2) $\mathbf{v}$: A random direction vector sampled from a Gaussian distribution and orthogonalized with respect to $\mathbf{u}$ (i.e., $\mathbf{v} \perp \mathbf{u}$). We evaluate the risk function $\psi(\mathbf{e}(\alpha, \beta); \theta_t)$ on a grid of coordinates $(\alpha, \beta)$ parameterized by:

$$\mathbf{e}(\alpha, \beta) = \mathbf{e}_0 + \alpha \cdot \mathbf{u} + \beta \cdot \mathbf{v} \tag{17}$$

The reference model is Qwen3-4B-Instruct-2507, and the fine-tuned model is the same model after standard SFT. As illustrated in Figure 7, the resulting landscape exhibits a highly rugged and non-convex structure with multiple narrow peaks. This observation empirically confirms that simple greedy optimization is prone to getting trapped in sub-optimal local maxima, thereby validating the necessity of our batched adversarial search strategy to effectively uncover diverse high-risk forgetting modes.

## C.3. Main Results

**FinGPT main results.** Table 4 summarizes the continual post-training results on FinGPT. **Standard SFT** substantially improves the target-domain accuracy, but causes a pronounced degradation on the general benchmarks (Avg). Adding an explicit **KL regularizer** mitigates the degradation, yet still exhibits a non-trivial retention drop.

Across a diverse set of replay- and regularization-based baselines, we find a consistent trade-off between target performance and general retention. In contrast, **Ours+RKL** achieves strong target-domain gains while keeping the averaged general-benchmark performance essentially unchanged, delivering the best overall retention–adaptation balance among compared methods. We further observe that the two variants (FKL/BD) show weaker retention than RKL, supporting the importance of optimizing the prompt distribution rather than relying on fixed heuristics.

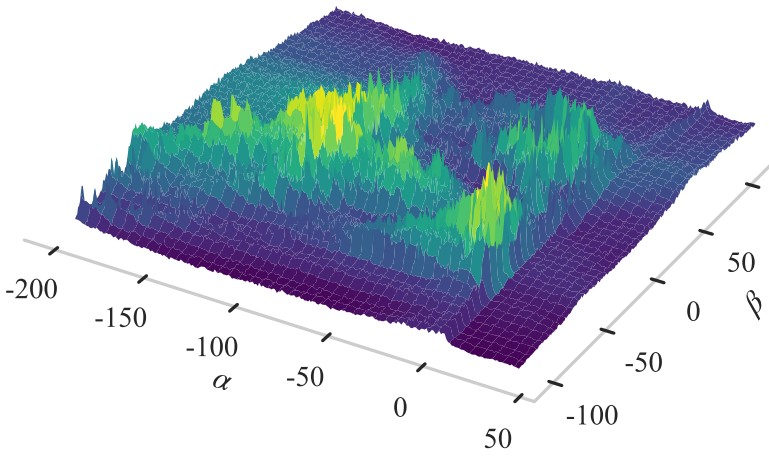

*Figure 7.* Landscape of risk function $\psi(\mathbf{e}; \theta_t)$.

## C.4. Ablation Study

**What components matter?**   Table 5 evaluates key design choices under a fixed search budget. Replacing the optimized prompt embedding with **random embeddings (RE)** leads to the largest degradation, hurting both target performance and general-benchmark Avg, suggesting that the benefit does not come from "extra prompts" but from *finding high-risk / high-utility modes*.

Using **random search (RS)** narrows the gap but remains consistently worse than the full method, indicating that gradient-guided search is important for reliably locating informative modes within the same budget. Finally, removing the **manifold constraint (WM)** also reduces performance, showing that constraining the search to a plausible region (close to the model's embedding manifold) improves both stability and general retention. Overall, each component contributes, and their combination yields the best retention–adaptation balance.

## C.5. Sensitivity Analysis

Table 6 studies prompt length. We observe a broad plateau: short prompts already provide strong retention, while the default length (8) achieves the best overall balance. Very long prompts slightly degrade Avg, consistent with the intuition that an overly expressive prompt space makes optimization noisier under a fixed budget. Importantly, across all lengths, RKL maintains substantially better retention than standard SFT and SFT+KL, highlighting that the main gain comes from targeted risk-based optimization rather than hyperparameter tuning.

Table 7 varies the prompt batch size used during search. Performance is stable across a wide range (e.g., 4–16), and the default batch size (8) yields the best overall Avg. Extremely small batches slightly underperform, likely due to higher variance in the risk estimate, while extremely large batches do not further improve retention under the same optimization budget. Overall, the method is not overly sensitive to batch size, indicating robust behavior of the search-and-repair pipeline.

*Table 4.* Main results on FinGPT. We report the target-domain validation accuracy and the post-fine-tuning performance on general benchmarks (GPQA, GSM8K, HumanEval, IFEval, MMLU, MMLU-Pro). Numbers in parentheses denote the performance accuracy scores; **Avg** is the average across the general benchmarks.

| Method | FinGPT (Target) | GPQA | GSM8K | HumanEval | IFEval | MMLU | MMLU-Pro | Avg |
|---|---|---|---|---|---|---|---|---|
| Naive | 23.66 (Ref) | 44.33 | 94.67 | 81.71 | 84.67 | 83.33 | 71.33 | 76.67 (Ref) |
| *Standard SFT* | | | | | | | | |
| SFT | 90.00 (+66.34) | 35.33 | 91.33 | 79.26 | 69.33 | 68.33 | 61.66 | 67.54 (-9.13) |
| SFT + KL | 90.67 (+67.01) | 40.66 | 93.00 | 80.48 | 83.33 | 79.66 | 69.00 | 74.35 (-2.32) |
| *Baselines* | | | | | | | | |
| TALR | **92.00** (+68.34) | 33.67 | 91.33 | 78.04 | 68.33 | 73.00 | 62.33 | 67.78 (-8.89) |
| PSFT | 88.00 (+64.34) | 35.33 | 92.00 | 81.10 | 70.00 | 68.33 | 63.00 | 68.29 (-8.38) |
| SSR | 83.00 (+59.34) | 42.33 | **95.33** | **85.36** | 73.00 | 79.33 | 66.66 | 73.67 (-3.00) |
| DataMix | 84.33 (+60.67) | 34.67 | 81.33 | 74.39 | 61.33 | 65.67 | 60.67 | 63.01 (-13.66) |
| Freeze | 86.66 (+63.00) | 41.00 | 92.66 | 81.09 | 80.33 | 81.66 | 69.33 | 74.35 (-2.32) |
| Orthogonal | 74.33 (+50.67) | 44.00 | 95.00 | 77.44 | **85.00** | 81.66 | 70.00 | 75.52 (-1.15) |
| *Ours* | | | | | | | | |
| **Ours+RKL** | 88.33 (+64.67) | **45.33** | 95.00 | 82.93 | 84.66 | 82.00 | 70.00 | **76.65** (-0.02) |
| **Ours+FKL** | 88.33 (+64.67) | 42.00 | 94.33 | 77.43 | 84.33 | **82.33** | **71.00** | 75.24 (-1.43) |
| **Ours+BD** | 84.66 (+61.00) | 42.00 | 93.00 | 79.26 | 81.00 | 80.00 | 68.66 | 73.99 (-2.68) |

## C.6. Runtime Analysis

We report the averaged wall-clock *training* time (seconds per optimization step) of our method and competing baselines. All measurements are conducted on a single NVIDIA A100 Tensor Core GPU with 16 Intel Xeon Platinum 8358 CPU cores, and we only count the training-time cost (excluding any evaluation/testing overhead). Table 8 summarizes the results.

Overall, most lightweight retention baselines (e.g., TALR/PSFT/SSR/DataMix) stay close to vanilla SFT ($\approx$ 3.0–3.3 s/step), while adding an explicit KL regularizer already introduces non-trivial overhead (SFT+KL: 3.93 s/step). Our method is *not* optimized for speed: both Ours+RKL and Ours+FKL incur a moderate slowdown ($\approx$ 4.5 s/step, about 1.47–1.48$\times$ SFT), primarily due to the additional inner-loop adversarial search and the distillation-based repair. When enabling explicit generation/rollouts (Ours+BD), decoding becomes the dominant cost, leading to a larger overhead (10.32 s/step, $\approx$ 3.39$\times$ SFT), consistent with our observation that inner-loop search and rollouts introduce extra runtime and motivate more adaptive scheduling for efficiency.

## C.7. Distribution-Shift Mechanism Validation

This study examines whether the searched latent probes identify broader functional drift rather than isolated adversarial artifacts. We therefore compare output-distribution shifts on held-out searched probes, natural old-domain prompts, and target-domain validation prompts, as visualized in Figure 8.

**Settings.** We use the same Qwen3-4B SciKnowEval chemistry L3 setting as the main experiments and compare the reference, SFT, and ALER checkpoints. We compute Jensen–Shannon divergence over output distributions on three prompt groups: held-out searched probes, risk-enriched natural old-domain prompts sampled from GPQA and MMLU-Pro, and target-domain validation prompts.

**Results.** Figure 8 shows that, on held-out searched probes, ALER reduces mean JS divergence to the reference from $0.473$ to $0.014$. On natural old-domain prompts from GPQA and MMLU-Pro, the divergence decreases from $0.633$ to $0.564$. The

*Table 5.* Ablation results on SciKnowEval (chemistry L3). We report the target-domain validation accuracy and post-fine-tuning performance on general benchmarks (GPQA, GSM8K, HumanEval, IFEval, MMLU, MMLU-Pro). **Avg** is the average across the general benchmarks. Numbers in parentheses denote changes relative to the full method (**Ours+RKL**). All ablations use the same search budget.

| Method | SciKnowEval (Target) | GPQA | GSM8K | HumanEval | IFEval | MMLU | MMLU-Pro | Avg |
|---|---|---|---|---|---|---|---|---|
| Naive | 41.67 | 44.33 | 94.67 | 81.71 | 84.67 | 83.33 | 71.33 | 76.67 |
| *Full Method* | | | | | | | | |
| **Ours+RKL** | **81.67** (Ref) | 45.67 | 94.33 | 84.76 | 84.33 | 81.00 | 70.67 | **76.79** (Ref) |
| *Ablations* | | | | | | | | |
| RE (Random Embedding) | 72.50 (-9.17) | 30.33 | 89.00 | 77.44 | 81.66 | 66.60 | 57.66 | 67.11 (-9.68) |
| RS (Random Search) | 79.16 (-2.51) | 41.66 | 94.00 | 76.82 | 84.33 | 78.33 | 67.66 | 73.80 (-2.99) |
| WM (No Manifold) | 80.00 (-1.67) | 43.66 | 94.33 | 82.31 | 82.66 | 81.66 | 69.00 | 75.60 (-1.19) |

*Table 6.* Sensitivity Analysis on Prompt Length (Ours+RKL). Length 8 is the default setting.

| Method | SciKnowEval (Target) | GPQA | GSM8K | HumanEval | IFEval | MMLU | MMLU-Pro | Avg |
|---|---|---|---|---|---|---|---|---|
| Naive | 41.67 (Ref) | 44.33 | 94.67 | 81.71 | 84.67 | 83.33 | 71.33 | 76.67 (Ref) |
| *Standard SFT* | | | | | | | | |
| SFT | 79.17 (+37.50) | 33.67 | 91.67 | 69.51 | 78.33 | 64.00 | 49.00 | 64.36 (-12.31) |
| SFT + KL | 78.33 (+36.66) | 37.67 | 90.33 | 79.88 | 83.33 | 78.00 | 65.00 | 72.37 (-4.30) |
| *Effect of Prompt Length (Ours+RKL)* | | | | | | | | |
| Length 1 | 76.66 (+34.99) | 42.67 | 93.33 | 82.93 | 83.00 | 80.33 | **71.33** | 75.76 (-0.91) |
| Length 2 | 77.50 (+35.83) | 40.33 | 93.33 | 82.93 | 82.66 | 81.00 | 70.67 | 75.15 (-1.52) |
| Length 4 | 79.16 (+37.49) | 37.66 | 93.66 | 84.14 | 83.33 | **82.66** | 69.00 | 75.07 (-1.60) |
| **Length 8 (Default)** | **81.67** (+40.00) | **45.67** | 94.33 | **84.76** | **84.33** | 81.00 | 70.67 | **76.79** (+0.12) |
| Length 16 | 79.16 (+37.49) | 42.00 | **95.00** | 82.92 | 83.33 | **82.66** | 69.33 | 75.87 (-0.80) |
| Length 32 | 80.83 (+39.16) | 39.66 | 94.33 | 81.10 | 83.33 | 78.67 | 68.66 | 74.29 (-2.38) |
| Length 64 | 79.17 (+37.50) | 42.33 | 94.67 | 84.14 | 82.00 | 78.33 | 67.33 | 74.80 (-1.87) |

target-domain panel shows $JS(SFT, ALER) = 0.021$ on validation prompts, indicating that target adaptation is largely preserved.

## C.8. Scaling to Larger Model

This study evaluates whether the retention benefit and runtime profile remain stable when moving from the main 4B model to a larger B model, with results reported in Table 9.

**Settings.** We repeat the SciKnowEval chemistry L3 experiment with the corresponding Qwen3-8B model. The setup follows the main configuration except that the learning rate is set to $2 \times 10^{-5}$. We report target-domain validation accuracy, six-benchmark retention, their average, and wall-clock training time per optimization step.

**Results.** Table 9 shows that, on Qwen3-8B, Ours+RKL improves the general benchmark average from 57.99 under SFT to 71.57, while the runtime multiplier remains comparable to the 4B setting at $1.53\times$.

*Table 7.* Sensitivity Analysis on Batch Size (Ours+RKL). Batch Size 8 is the default setting.

| Method | SciKnowEval (Target) | GPQA | GSM8K | HumanEval | IFEval | MMLU | MMLU-Pro | Avg |
|---|---|---|---|---|---|---|---|---|
| Naive | 41.67 (Ref) | 44.33 | 94.67 | 81.71 | 84.67 | 83.33 | 71.33 | 76.67 (Ref) |
| *Standard SFT* | | | | | | | | |
| SFT | 79.17 (+37.50) | 33.67 | 91.67 | 69.51 | 78.33 | 64.00 | 49.00 | 64.36 (-12.31) |
| SFT + KL | 78.33 (+36.66) | 37.67 | 90.33 | 79.88 | 83.33 | 78.00 | 65.00 | 72.37 (-4.30) |
| *Effect of Batch Size (Ours+RKL)* | | | | | | | | |
| Batch Size 1 | 75.83 (+34.16) | 40.67 | 93.67 | 82.32 | 82.67 | 79.67 | 67.67 | 74.44 (-2.23) |
| Batch Size 2 | 76.67 (+35.00) | 39.33 | 94.33 | 83.54 | 83.00 | 78.67 | 69.00 | 74.64 (-2.03) |
| Batch Size 4 | 78.33 (+36.66) | 43.00 | 93.00 | 81.71 | **86.00** | 79.00 | 68.00 | 75.11 (-1.56) |
| **Batch Size 8 (Default)** | **81.67** (+40.00) | **45.67** | 94.33 | **84.76** | 84.33 | 81.00 | 70.67 | **76.79** (+0.12) |
| Batch Size 16 | 79.17 (+37.50) | 45.33 | **95.00** | 84.15 | 83.67 | 80.00 | 70.33 | 76.41 (-0.26) |
| Batch Size 32 | 77.50 (+35.83) | 44.00 | 93.00 | 81.71 | 83.33 | 80.67 | 70.00 | 75.45 (-1.22) |
| Batch Size 64 | 77.50 (+35.83) | 43.00 | 94.67 | 81.71 | 84.00 | **82.00** | **72.00** | 76.23 (-0.44) |

*Table 8.* Training-time runtime (seconds per optimization step). We exclude evaluation/testing time.

| Method | SFT | SFT+KL | TALR | PSFT | SSR | DataMix | Freeze | Orthogonal | Ours+RKL | Ours+FKL | Ours+BD |
|---|---|---|---|---|---|---|---|---|---|---|---|
| **Runtime (s/step)** | 3.04 | 3.93 | 3.06 | 3.11 | 3.29 | 3.04 | 2.97 | 5.76 | 4.50 | 4.46 | 10.32 |

## D. Case Study

We analyze the autonomous generation quality across three distinct checkpoints. For each stage, we present two distinct cases comparing the *Reference Model* (Ground Truth) and the *Finetuned Model*.

**Stage 1: High-Fidelity Knowledge Retention (Figure 9).** At initialization, the Finetuned model matches the Reference model's ability to generate complex, structured content.

- **Case 1 (Scientific Reasoning):** The Finetuned model accurately generates a biological verification task regarding "Diatoms and nitrogen fixation," correctly refuting the premise with detailed scientific facts.

- **Case 2 (Physics Problem Solving):** It successfully constructs a sequence of kinematics problems (velocity and displacement), demonstrating the retention of domain-specific vocabulary and mathematical structure similar to the

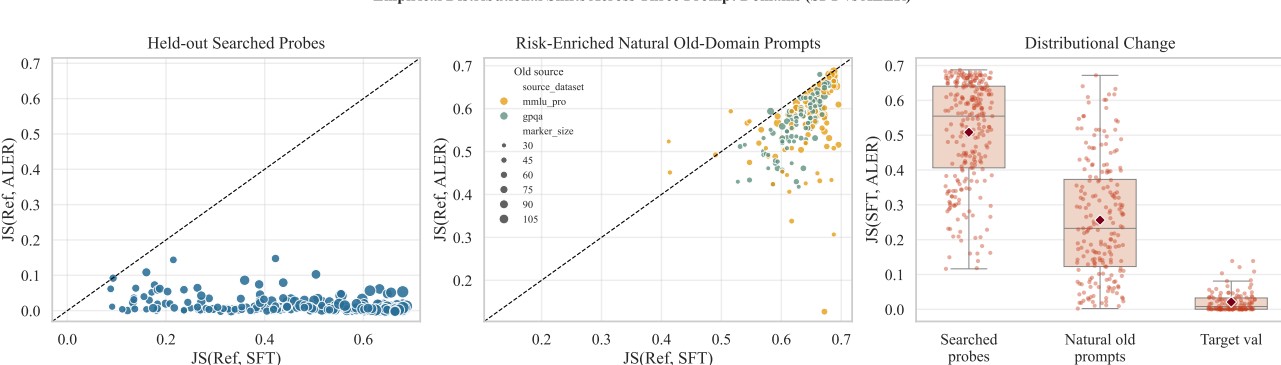

*Figure 8.* Empirical distributional shifts across three prompt domains. **Left:** held-out searched probes. **Middle:** risk-enriched natural old-domain prompts from GPQA and MMLU-Pro. **Right:** JS(SFT, ALER) across searched probes, natural old-domain prompts, and target-domain validation prompts.

*Table 9.* Larger-scale runtime comparison on Qwem3-8B model under the same setup (except learning rate $= 2 \times 10^{-5}$). We report target-domain validation accuracy, post-fine-tuning performance on general benchmarks, and wall-clock training time.

| Method | SciKnowEval (Target) | GPQA | GSM8K | HumanEval | IFEval | MMLU | MMLU-Pro | Avg | Time |
|---|---|---|---|---|---|---|---|---|---|
| Naive | 42.50 (Ref) | **42.67** | **94.00** | 82.32 | **82.00** | **80.33** | **65.33** | **74.44** (Ref) | – |
| | | | | *Standard SFT* | | | | | |
| SFT | **71.67** (+29.17) | 28.33 | 81.33 | 79.27 | 68.00 | 62.67 | 28.33 | 57.99 (-16.46) | **10.21** |
| SFT + KL | 70.00 (+27.50) | 34.33 | 89.67 | **84.76** | 72.00 | 64.67 | 53.00 | 66.41 (-8.04) | 13.34 |
| | | | | *Ours* | | | | | |
| **Ours+RKL** | 70.83 (+28.33) | 40.33 | 94.00 | **84.76** | 77.00 | 69.67 | 63.67 | 71.57 (-2.87) | 15.62 |

Reference model's literary analysis.

**Stage 2: Simplification and loss of Nuance (Figure 10).** By Global Step 26, the model begins to favor shorter, simpler patterns over complex reasoning.

- **Case 1 (Math Simplification):** While the Reference model solves a complex demographic percentage problem involving millions of people (Atlanta population), the Finetuned model generates a rudimentary arithmetic chain ($4 \times 5 - 3 = 17$). The logic remains correct, but the complexity has degraded.

- **Case 2 (Content Truncation):** In contrast to the Reference model's detailed logic puzzle about negative numbers, the Finetuned model outputs an extremely short query ("2 + 2?"), lacking the depth of the original distribution.

**Stage 3: Distribution Shift and Mode Collapse (Figure 11).** By Global Step 999, the model suffers from severe distribution shift.

- **Case 1 & 2 (Topic Drift):** While the Reference model continues to output structured QA pairs (e.g., explaining Taekwondo or Geometry), the Finetuned model collapses into generating short, out-of-context definition questions (e.g., "What is a coordinate system?" and "What is a tidal phenomenon?"). Note that while translated here for clarity, these outputs represent a fundamental loss of the instruction-following format, shifting from long-form reasoning to brief, unrelated queries.

# E. Theoretical Results

Our theoretical results contain two parts: learning dynamics from the perspective of forgetting in the motivation section, and the theoretical analysis of our proposed **ALER** framework.

## E.1. Learning Dynamics of LLM Fine-tuning

In this section, we provide a detailed and rigorous theoretical analysis of the learning dynamics of LLM finetuning from the perspective of catastrophic forgetting. This analysis is a natural derivation of (Ren & Sutherland, 2024).

Let the vocabulary size be $V$, and the model parameters be $\theta \in \mathbb{R}^P$. For any context sequence $\mathbf{c}$, the model produces logits

$$z_\theta(\mathbf{c}) \in \mathbb{R}^V, \qquad \pi_\theta(\cdot \mid \mathbf{c}) = \text{softmax}\left(z_\theta(\mathbf{c})\right) \in \Delta^{V-1}. \tag{18}$$

For a pre-training dataset $\mathcal{D}_p = \left\{\mathbf{x}^{(n)}\right\}_{n=1}^N$, where $\mathbf{x}^{(n)} = \left(x_1^{(n)}, x_2^{(n)}, \dots, x_{T_n}^{(n)}\right)$, $T_n$ is the length of the sequence $\mathbf{x}^{(n)}$, and $x_i^{(n)} \in \{1, 2, \dots, V\}$. The loss for pre-training:

$$\ell_p^{(n)}(\theta) = \text{NLL}\left(\mathbf{x}^{(n)}; \pi_\theta\right) = -\sum_{t=1}^{T_n} \ln \pi_\theta\left(x_t^{(n)} \mid \mathbf{x}_{<t}^{(n)}\right), \qquad \mathcal{L}_p(\theta) = \sum_{n=1}^N \ell^{(n)} = \sum_{n=1}^N \ell_p^{(n)}(\theta). \tag{19}$$

**Case 1: Scientific Knowledge**

**Reference Model**: (Ground Truth)

> **Generated Sequence:**
> "6. Anya has 35 friends. Two of them are immediate family...
> Percentage $= (2/35) \times 100 \approx 5.71\%$
> **Answer: 5.71%**."

**Finetuned Model**: (Step 1)

> **Generated Sequence:**
> "26. The relationship between diatoms and nitrogen fixation... True or False?
> **False.** Diatoms do not fix nitrogen... Nitrogen fixation is primarily carried out by cyanobacteria."

**Case 2: Domain Specificity**

**Reference Model**: (Ground Truth)

> **Generated Sequence:**
> "1. What are the basic characteristics of the protagonist...?
> The protagonist is introverted, observant... embodies the theme of individual conscience."

**Finetuned Model**: (Step 1)

> **Generated Sequence:**
> "1. A brick is thrown upward with initial velocity 5 m/s...
> 2. A ball is thrown vertically upward with 15 m/s...
> 3. A ball is thrown... [Sequence of Physics Problems]"

*Figure 9.* **Global Step 1 Analysis.** The Finetuned model generates high-quality, long-form content indistinguishable from the Reference. (Case 1) Accurate Biology reasoning. (Case 2) Structured Physics problem generation.

Consider a pre-training sample $\mathbf{x}^{(n)}$, for each position $t$, define the context input $\mathbf{c}_{n,t} = \mathbf{x}_{<t}^{(n)}$ and the per-token label $y_{n,t} = x_t^{(n)}$, then

$$\ell_p^{(n)}(\theta) = \sum_{t=1}^{T_n} \ell_{n,t}(\theta), \qquad \ell_{n,t}(\theta) = -\ln \pi_\theta(y_{n,t} \mid \mathbf{c}_{n,t}). \tag{20}$$

For a new sample $\chi_u = [\mathbf{x}^{(u)}; \mathbf{y}^{(u)}]$, let's consider one step SGD:

$$\theta_{t+1} = \theta_t - \eta \nabla_{\theta_t} \mathcal{L}_{\text{sft}}(\theta_t, \chi_u), \tag{21}$$

where SFT loss is defined as

$$\mathcal{L}_{\text{sft}}(\theta, \chi_u) = -\sum_{l=1}^{L_u} \ln \pi_\theta \left( y_l^{(u)} \mid \mathbf{x}^{(u)}, \mathbf{y}_{<l}^{(u)} \right). \tag{22}$$

Similarly, define the context input $\mathbf{c}_{u,l} = \left( \mathbf{x}^{(u)}, \mathbf{y}_{<l}^{(u)} \right)$.

Define the logit Jacobian at context $\mathbf{c}$:

$$\mathcal{J}_\theta(\mathbf{c}) = \nabla_\theta z_\theta(\mathbf{c}) \in \mathbb{R}^{V \times P}. \tag{23}$$

Define the context-to-context empirical NTK matrix:

$$\mathcal{K}_\theta(\mathbf{c}, \mathbf{c}') = \mathcal{J}_\theta(\mathbf{c}) \mathcal{J}_\theta(\mathbf{c}')^\top \in \mathbb{R}^{V \times V}. \tag{24}$$

For token-level NLL $\ell(\mathbf{c}, y; \theta) = -\ln \pi_\theta(y \mid \mathbf{c})$, define the logit-space gradient

$$g_\theta(\mathbf{c}, y) = \nabla_z \ell(\mathbf{c}, y; \theta) = \pi_\theta(\cdot \mid \mathbf{c}) - \mathbf{e}_y \in \mathbb{R}^V, \tag{25}$$

where $\mathbf{e}_y$ is the one-hot vector for token $y$.

**Case 1: Arithmetic Complexity**

**Reference Model**: (Ground Truth)

> **Generated Sequence:**
> "15 million people live in Atlanta. 30% are under 18...
> $0.30 \times 15{,}000{,}000 = 4{,}500{,}000$
> **Answer: 4,500,000 people.**"

**Finetuned Model**: (Step 26)

> **Generated Sequence:**
> "4 * 5 - 3?
> Step-by-step: 4 * 5 = 20. 20 - 3 = 17.
> Final answer: 17"

**Case 2: Reasoning Depth**

**Reference Model**: (Ground Truth)

> **Generated Sequence:**
> "12. The greatest negative number is...
> Explanation: There is no greatest negative number because... numbers decrease in value as they move from zero."

**Finetuned Model**: (Step 26)

> **Generated Sequence:**
> "2 + 2?"

*Figure 10.* **Global Step 26 Analysis.** The Finetuned model begins to degrade in complexity. (Case 1) Complex contextual math shifts to simple arithmetic. (Case 2) Deep logical explanations are replaced by trivial queries.

**Proposition E.1** (One-step change of pre-training NLL under an SFT update). *Assume $\ell_p^{(n)}$ is twice differentiable in a neighborhood of $\theta$. Then the change in the pre-training objective on sample $\mathbf{x}^n$ is*

$$\ell_p^{(n)}(\theta') - \ell_p^{(n)}(\theta) = -\eta \sum_{t=1}^{T_n} \sum_{l=1}^{L_u} g_\theta\left(\mathbf{c}_{n,t}, x_t^{(n)}\right)^\top \mathcal{K}_\theta\left(\mathbf{c}_{n,t}, \mathbf{c}_{u,l}\right) g_\theta\left(\mathbf{c}_{u,t}, y_l^{(u)}\right) + O(\eta^2). \tag{26}$$

*Proof.* Let $\Delta\theta = \theta' - \theta$. By second-order differentiability,

$$\ell_p^{(n)}(\theta') = \ell_p^{(n)}(\theta) + \nabla_\theta \ell_p^{(n)}(\theta)^\top \Delta\theta + O(\|\Delta\theta\|^2). \tag{27}$$

Using the update $\Delta\theta = -\eta \nabla_\theta \mathcal{L}_{\text{sft}}(\theta, \chi_u)$, we have

$$\ell_p^{(n)}(\theta') - \ell_p^{(n)}(\theta) = -\eta \left\langle \nabla_\theta \ell_p^{(n)}(\theta), \nabla_\theta \mathcal{L}_{\text{sft}}(\theta, \chi_u) \right\rangle + O\left(\eta^2 \|\nabla_\theta \mathcal{L}_{\text{sft}}(\theta, \chi_u)\|^2\right). \tag{28}$$

For a token-level NLL term $\ell(\mathbf{c}, y; \theta) = -\ln \pi_\theta(y \mid \mathbf{c})$, by the chain rule:

$$\nabla_\theta \ell(\mathbf{c}, y; \theta) = \left(\nabla_\theta z_\theta(\mathbf{c})\right)^\top \nabla_z \ell(\mathbf{c}, y; \theta) = \mathcal{J}_\theta(\mathbf{c})^\top g_\theta(\mathbf{c}, y). \tag{29}$$

Therefore,

$$\begin{aligned}
\nabla_\theta \ell_p^{(n)}(\theta) &= \nabla_\theta \sum_{t=1}^{n} \ell\left(\mathbf{c}_{n,t}, x_t^{(n)}; \theta\right) \\
&= \sum_{t=1}^{n} \nabla_\theta \ell\left(\mathbf{c}_{n,t}, x_t^{(n)}; \theta\right) \\
&= \sum_{t=1}^{n} \mathcal{J}_\theta(\mathbf{c}_{n,t})^\top g_\theta\left(\mathbf{c}_{n,t}, x_t^{(n)}\right),
\end{aligned} \tag{30}$$

**Case 1: Structural Collapse**

**Reference Model**: (Ground Truth)

> **Generated Sequence:**
> "What are the three important elements of the martial art of taekwondo?
> The three important elements are **punches, kicks, and blocks**."

**Finetuned Model**: (Step 999)

> **Generated Sequence:**
> "What is a coordinate system?"
> **[Short, unrelated query]**

**Case 2: Loss of Format**

**Reference Model**: (Ground Truth)

> **Generated Sequence:**
> "180 degrees of rotation about the origin. What is the image of point (4, 2)?
> The image... is (-4, -2)."

**Finetuned Model**: (Step 999)

> **Generated Sequence:**
> "What is a tidal phenomenon?"
> **[Short, unrelated query]**

*Figure 11.* **Global Step 999 Analysis.** Catastrophic collapse. While the Reference model maintains QA structure (Case 1 & 2), the Finetuned model ignores the task and outputs brief, unrelated definitions (Translated from collapsed output).

and

$$\nabla_\theta \mathcal{L}_{\text{sft}}(\theta; \chi_u) = \nabla_\theta \sum_{l=1}^{L_u} \ell\left(\mathbf{c}_{u,l}, y_l^{(u)}; \theta\right)$$

$$= \sum_{l=1}^{L_u} \nabla_\theta \ell\left(\mathbf{c}_{u,l}, y_l^{(u)}; \theta\right)$$

$$= \sum_{l=1}^{L_u} \mathcal{J}_\theta(\mathbf{c}_{u,l})^\top g_\theta\left(\mathbf{c}_{u,l}, y_l^{(u)}\right). \tag{31}$$

Then the inner product term gives:

$$\left\langle \nabla_\theta \ell_p^{(n)}(\theta), \nabla_\theta \mathcal{L}_{\text{sft}}(\theta, \chi_u) \right\rangle = \left\langle \sum_{t=1}^{n} \mathcal{J}_\theta(\mathbf{c}_{n,t})^\top g_\theta\left(\mathbf{c}_{n,t}, x_t^{(n)}\right), \sum_{l=1}^{L_u} \mathcal{J}_\theta(\mathbf{c}_{u,l})^\top g_\theta\left(\mathbf{c}_{u,l}, y_l^{(u)}\right) \right\rangle$$

$$= \sum_{t=1}^{n} \sum_{l=1}^{L_u} \left\langle \mathcal{J}_\theta(\mathbf{c}_{n,t})^\top g_\theta\left(\mathbf{c}_{n,t}, x_t^{(n)}\right), \mathcal{J}_\theta(\mathbf{c}_{u,l})^\top g_\theta\left(\mathbf{c}_{u,l}, y_l^{(u)}\right) \right\rangle$$

$$= \sum_{t=1}^{n} \sum_{l=1}^{L_u} g_\theta^\top\left(\mathbf{c}_{n,t}, x_t^{(n)}\right) \mathcal{J}_\theta(\mathbf{c}_{n,t}) \mathcal{J}_\theta(\mathbf{c}_{u,l})^\top g_\theta\left(\mathbf{c}_{u,l}, y_l^{(u)}\right)$$

$$= \sum_{t=1}^{n} \sum_{l=1}^{L_u} g_\theta^\top\left(\mathbf{c}_{n,t}, x_t^{(n)}\right) \mathcal{K}(\mathbf{c}_{n,t}, \mathbf{c}_{u,l}) g_\theta\left(\mathbf{c}_{u,l}, y_l^{(u)}\right). \tag{32}$$

Finally, we have

$$\ell_p^{(n)}(\theta') - \ell_p^{(n)}(\theta) = -\eta \sum_{t=1}^{n} \sum_{l=1}^{L_u} g_\theta^\top\left(\mathbf{c}_{n,t}, x_t^{(n)}\right) \mathcal{K}(\mathbf{c}_{n,t}, \mathbf{c}_{u,l}) g_\theta\left(\mathbf{c}_{u,l}, y_l^{(u)}\right) + O\left(\eta^2\right). \tag{33}$$

$\square$

Proposition E.1 provides us with a clear picture that forgetting is not happening uniformly across the entire pre-training dataset $\mathcal{D}_p$; instead, the new data affects those samples that most closely correlate with the newly finetuned data.

### E.2. From Sample-wise Forgetting to a KL-based Surrogate

In (2) in the main paper, we define the sample-wise forgetting as the change of NLL in the However, in practical settings, $\mathcal{D}_p$ and the actual data distribution are inaccessible. Therefore, (2) provides a conceptually correct but in practice inapplicable forgetting metric. Consequently, under this condition, KL with reference model is a more proper and consistent surrogate metric.

Let the pre-training data distribution be $P_p(\cdot \mid \mathbf{c})$, where $\mathbf{c}$ represents the context of any pre-training sample $\mathbf{x}^{(n)}$, We define the context distribution as

$$\mu_p(\mathbf{c}) = \mathbb{P}\big(\mathbf{c} = \mathbf{x}_{<t} \text{ for } (\mathbf{x}, t) \sim \mathcal{D}_p\big). \tag{34}$$

Then, the change in the NLL can be rewritten as:

$$\mathbb{E}_{\mathbf{c} \sim \mu_p} \left[ \ell_p^{(n)}(\theta') - \ell_p^{(n)}(\theta) \right] = \mathbb{E}_{\mathbf{c} \sim \mu_p} \left[ \mathbb{E}_{y \sim P_p(\cdot \mid \mathbf{c})} \Big( -\ln \pi_{\theta'}(y \mid \mathbf{c}) + \ln \pi_\theta(y \mid \mathbf{c}) \Big) \right], \tag{35}$$

which indicates that the forgetting is actually excess cross-entropy with respect to the conditional distribution of pre-training data. Now that $P_p(\cdot \mid \mathbf{c})$ is inaccessible, we approximate the pre-training distribution with the reference policy $\pi_{\text{ref}}$ itself, based on the common sense that the NLL objective during the pre-training phase is actually an **optimal information compression of the pre-training dataset** (Rissanen, 1978; Cover, 1999; Delétang et al., 2023). Define the reference cross-entropy as:

$$\mathcal{R}_{\text{ref}}(\theta) = \mathbb{E}_{\mathbf{c} \sim \mu_p} \mathbb{E}_{y \sim \pi_{\text{ref}}(\cdot \mid \mathbf{c})} \left[ -\ln \pi_\theta(y \mid \mathbf{c}) \right]. \tag{36}$$

If $\pi_{\text{ref}}$ if well pre-trained and is a proper approximation of $\mu_p$, which is a standard assumption in continual learning, then

$$\mathbb{E}_{\mathbf{c} \sim \mu_p} \text{forget}\left( \mathbf{x}^{(n)} \right) \approx \mathcal{R}_{\text{ref}}(\theta') - \mathcal{R}_{\text{ref}}(\theta). \tag{37}$$

For any context $\mathbf{c}$, the reference cross entropy can be factored as

$$
\begin{aligned}
\mathbb{E}_{y \sim \pi_{\theta_{\text{ref}}}(\cdot \mid \mathbf{c})} \left[ -\ln \pi_\theta(y \mid \mathbf{c}) \right] &= \sum_{y \in \mathcal{V}} \pi_{\theta_{\text{ref}}}(y \mid \mathbf{c}) \cdot \left[ -\ln \pi_\theta(y \mid \mathbf{c}) \right] \\
&= \sum_{y \in \mathcal{V}} \pi_{\theta_{\text{ref}}}(y \mid \mathbf{c}) \cdot \left[ -\ln \pi_{\text{ref}}(y \mid \mathbf{c}) + \ln \frac{\pi_{\text{ref}}(y \mid \mathbf{c})}{\pi_\theta(y \mid \mathbf{c})} \right] \\
&= -\sum_{y \in \mathcal{V}} \pi_{\theta_{\text{ref}}}(y \mid \mathbf{c}) \cdot \ln \pi_{\text{ref}}(y \mid \mathbf{c}) + \sum_{y \in \mathcal{V}} \pi_{\theta_{\text{ref}}}(y \mid \mathbf{c}) \cdot \ln \frac{\pi_{\text{ref}}(y \mid \mathbf{c})}{\pi_\theta(y \mid \mathbf{c})} \\
&= H(\pi_{\theta_{\text{ref}}}(\cdot \mid \mathbf{c})) + \text{KL}\left( \pi_{\theta_{\text{ref}}}(\cdot \mid \mathbf{c}) \big\| \pi_\theta(\cdot \mid \mathbf{c}) \right),
\end{aligned}
\tag{38}
$$

where the first term $H(\pi_{\theta_{\text{ref}}}(\cdot \mid \mathbf{c}))$ is independent of $\theta$. Therefore, in the sense of expectation,

$$\mathcal{R}_{\text{ref}}(\theta') - \mathcal{R}_{\text{ref}}(\theta) = \mathbb{E}_{\mathbf{c} \sim \mu_p} \text{KL}\left( \pi_{\theta_{\text{ref}}}(\cdot \mid \mathbf{c}) \big\| \pi_{\theta'}(\cdot \mid \mathbf{c}) \right) - \mathbb{E}_{\mathbf{c} \sim \mu_p} \text{KL}\left( \pi_{\theta_{\text{ref}}}(\cdot \mid \mathbf{c}) \big\| \pi_\theta(\cdot \mid \mathbf{c}) \right) \tag{39}$$

Specifically, when $\theta = \theta_{\text{ref}}$, the KL penalty $\mathbb{E}_{\mathbf{c} \sim \mu_p} \text{KL}\left( \pi_{\theta_{\text{ref}}}(\cdot \mid \mathbf{c}) \big\| \pi_{\theta'}(\cdot \mid \mathbf{c}) \right)$ is precisely the expected sample-wise forgetting with respect to the reference model. The surrogate forgetting metric has a closed-form gradient:

$$\nabla_\theta \text{KL}\left( \pi_{\theta_{\text{ref}}}(\cdot \mid \mathbf{c}) \big\| \pi_\theta(\cdot \mid \mathbf{c}) \right) = \mathcal{J}_\theta(\mathbf{c})^\top \left( \pi_\theta(\cdot \mid \mathbf{c}) - \pi_{\theta_{\text{ref}}}(\cdot \mid \mathbf{c}) \right). \tag{40}$$

## E.3. Theoretical Analysis of ALER

In this section, we analyze repair strategies from a data-oriented perspective under a fixed repair budget. We focus on three representative choices of the *repair distribution*: **(1) On-batch distillation**, which applies a KL-based distillation term using only the current fine-tuning data (the standard trust-region style regularization in SFT/RL/DPO); **(2) Random distillation**, which constructs a replay set by uniform sampling from a general-purpose pool or by rehearsal synthesized from the reference model, and mixes it into fine-tuning; **(3) Adversarial repair (ours)**, which actively searches for latent prompt embeddings that maximize divergence from the frozen reference and distills the reference behavior back on these high-risk embeddings.

The following subsections are organized as follows: **Appendix E.3.1** formalizes the proxy forgetting risk and introduces a unified one-step projected repair view that allows fair comparison under the same budget. **Appendix E.3.2** states the finite-budget repair geometry assumptions used in the analysis (local quadratic control, long-tail repairability, and on-batch misalignment). **Appendix E.3.3** proves a two-sided descent lemma that reduces repaired risk improvement to the captured "repairable energy" of the old-risk gradient. **Appendix E.3.4** characterizes the energy capture of random repair and on-batch repair. **Appendix E.3.5** presents the main strict-gap theorems and derives the constants appearing in Proposition 4.3. Finally, **Appendix E.3.6** discusses limiting regimes (when the gaps vanish) and a brief oracle-to-estimation bridge.

### E.3.1. SETUP AND UNIFIED REPAIR VIEW

This subsection formalizes our theoretical objects and introduces a unified abstraction that places different repair strategies on equal footing under a fixed repair budget. The central goal is to compare how efficiently different strategies reduce the (inaccessible) old-knowledge forgetting risk, while holding the repair rule and budget constant.

**Conditional prediction and proxy forgetting risk.** Let $\mathcal{Z}$ denote the space of conditioning inputs. In our setting, an element $z \in \mathcal{Z}$ can be either (i) a discrete text context (a prefix) or (ii) a continuous latent prompt embedding $\mathbf{e} \in \mathbb{R}^{\ell \times d}$ used as a soft prefix. Let $\pi_{\text{ref}}(\cdot \mid z)$ be a frozen reference model and $\pi_\theta(\cdot \mid z)$ be the current model parameterized by $\theta \in \mathbb{R}^P$.

We measure local forgetting vulnerability at $z$ by the forward KL divergence from the reference to the current model:

$$\psi(z; \theta) \ := \ \mathrm{KL}(\pi_{\text{ref}}(\cdot \mid z) \,\|\, \pi_\theta(\cdot \mid z)) \ \geq \ 0. \tag{41}$$

As discussed in the main text, $\psi(z; \theta)$ is exactly the $\theta$-dependent component of the reference cross-entropy surrogate (up to an additive constant independent of $\theta$), and thus serves as a principled proxy for expected forgetting.

**Definition E.2** (Population forgetting risk). Let $\mathcal{D}_p$ denote the (inaccessible) pre-training distribution over $\mathcal{Z}$. The population forgetting risk is

$$\mathcal{F}(\theta) \ := \ \mathbb{E}_{z \sim \mathcal{D}_p}[\psi(z; \theta)] \,. \tag{42}$$

Our objective is to reduce $\mathcal{F}(\theta)$ during continual post-training, under constraints that (i) $\mathcal{D}_p$ is unavailable and (ii) repair effort is limited.

**Repair budget as "where to spend distillation".** Repair methods differ mainly in *which conditioning inputs* they apply distillation/regularization to. We model this choice via a *repair distribution* $\mathcal{D}_r$ over $\mathcal{Z}$ from which a method draws a small repair set. Concretely:

- **On-batch repair** uses $\mathcal{D}_r = \mathcal{D}_f$, the fine-tuning (on-batch) distribution.

- **Random repair** uses a generic replay/synthetic distribution $\mathcal{D}_r = \mathcal{D}_{\text{rand}}$ (e.g., uniform sampling from a general-purpose pool or rehearsal from $\pi_{\text{ref}}$).

- **Targeted repair (ours)** uses $\mathcal{D}_r = \mathcal{D}_\star$, the distribution implicitly induced by adversarial search that concentrates mass on high-risk regions (large $\psi$).

We quantify a finite repair budget by a small set size $k$. Let $S = \{z_1, \dots, z_k\}$ denote the repair set chosen by a method (randomly or deterministically).

**From repair data to repair directions.** A key step in making different strategies comparable is to isolate the *repair directions* they induce in parameter space. Fix the parameters at the beginning of a repair step to be $\theta_0$. Define the per-input (local) gradient feature

$$g(z) \; := \; \nabla_\theta \psi(z; \theta)\big|_{\theta=\theta_0} \in \mathbb{R}^P. \tag{43}$$

Intuitively, $g(z)$ describes which parameter directions would most directly reduce the proxy forgetting risk at $z$ (to first order).

The old-knowledge forgetting gradient at $\theta_0$ is the population average of these features:

$$G_p \; := \; \nabla_\theta \mathcal{F}(\theta)\big|_{\theta=\theta_0} \; = \; \mathbb{E}_{z\sim\mathcal{D}_p}[g(z)]. \tag{44}$$

Although $G_p$ is not directly computable (since $\mathcal{D}_p$ is inaccessible), it is the correct population object against which we evaluate forgetting-risk reduction.

**Unified repair view: same update rule, different subspaces.** Given a repair set $S = \{z_i\}_{i=1}^k$, we associate it with the linear subspace spanned by its gradient features:

$$\mathcal{U}(S) \; := \; \mathrm{span}\{g(z_1), \ldots, g(z_k)\} \subseteq \mathbb{R}^P, \qquad \dim(\mathcal{U}(S)) \leq k. \tag{45}$$

This construction abstracts "repair budget" as a constraint on the dimension of directions the method can exploit within one repair step.

**Definition E.3** (Projected repair step). For any subspace $\mathcal{U} \subseteq \mathbb{R}^P$ with $\dim(\mathcal{U}) \leq k$ and stepsize $\eta > 0$, define the projected repair update

$$\theta^+(\mathcal{U}) \; := \; \theta_0 - \eta\, \Pi_{\mathcal{U}}\, G_p, \tag{46}$$

where $\Pi_{\mathcal{U}}$ denotes the orthogonal projector onto $\mathcal{U}$.

**Interpretation.** This view fixes a *single* repair rule and attributes differences among methods solely to which subspace $\mathcal{U}$ they induce under the same budget. In particular, we will compare:

- $\theta_\star^+ := \theta^+(\mathcal{U}_\star)$, where $\mathcal{U}_\star$ is induced by the searched high-risk prompts;

- $\theta_{\mathrm{rand}}^+ := \theta^+(\mathcal{U}_{\mathrm{rand}})$, where $\mathcal{U}_{\mathrm{rand}}$ is induced by $k$ random repair inputs;

- $\theta_{\mathrm{batch}}^\cdot = \theta^+(\mathcal{U}_{\mathrm{batch}})$, where $\mathcal{U}_{\mathrm{batch}}$ is induced by $k$ on-batch repair inputs.

**Why projection is the right abstraction for "budget".** The projection operator captures a common and practically relevant constraint: under limited compute and limited repair data, a method can only access a low-dimensional set of effective update directions. This can arise from (i) explicitly restricting repair to a small set of prompts/embeddings, (ii) parameter-efficient repair that restricts degrees of freedom, or (iii) the empirical observation that gradients induced by a small repair set typically concentrate in a low-rank subspace.

Crucially, this abstraction lets us evaluate repair efficiency through a single quantity that will appear throughout the proofs: the *captured old-risk energy* $\|\Pi_{\mathcal{U}}G_p\|^2$. Later (Appendix E.3.3) we show that, under local quadratic control, the one-step decrease in $\mathcal{F}$ is sandwiched by constants times $\|\Pi_{\mathcal{U}}G_p\|^2$. Thus, comparing repair strategies reduces to comparing how much of $G_p$ each strategy can capture under budget.

**Discussion: oracle objects vs. implementable estimates.** The development above is stated in terms of the population quantity $G_p = \nabla\mathcal{F}(\theta_0)$, which is not directly observable in the pre-training-data-inaccessible regime. This is deliberate: the purpose of the guarantee is to characterize *which repair allocation is intrinsically more sample-efficient* for reducing old-knowledge forgetting risk. In later subsections we (i) derive strict comparisons using the population geometry, and (ii) discuss how searching high-risk prompts provides an estimator that concentrates on the dominant forgetting modes, thereby approximating the favorable subspace $\mathcal{U}_\star$ in practice.

E.3.2. ASSUMPTIONS: FINITE-BUDGET REPAIR GEOMETRY

This subsection states the structural conditions under which we can obtain *strict* finite-budget comparisons among targeted, random, and on-batch repair. All assumptions are intentionally *local* (restricted to the short segments actually traversed by a repair step) and *data-oriented* (expressed in terms of how repair directions cover the old-risk gradient), which makes them both interpretable and amenable to empirical validation.

**Compared subspaces under a budget.**  Recall from Appendix E.3.1 that a budget-$k$ repair step is modeled by choosing a subspace $\mathcal{U} \subseteq \mathbb{R}^P$ with $\dim(\mathcal{U}) \leq k$ and applying $\theta^+(\mathcal{U}) = \theta_0 - \eta\,\Pi_{\mathcal{U}}G_p$, where $G_p = \nabla\mathcal{F}(\theta_0)$. To reason uniformly about all compared methods (including stochastic ones), we formalize the family of subspaces they may induce.

**Definition E.4** (Compared subspace family)**.**  Let $\mathcal{U}_\star$ denote the (deterministic) repair subspace induced by targeted repair on searched prompts, let $\mathcal{U}_{\mathrm{rand}}$ be the (random) subspace induced by random repair, and let $\mathcal{U}_{\mathrm{batch}}$ be the (data-dependent) subspace induced by on-batch repair. We write $\mathcal{S}_k$ for the support set of all subspaces that may be selected by the compared strategies under budget $k$, i.e.,

$$\mathcal{S}_k \;:=\; \left\{\mathcal{U}_\star\right\} \,\cup\, \mathrm{supp}(\mathcal{U}_{\mathrm{rand}}) \,\cup\, \mathrm{supp}(\mathcal{U}_{\mathrm{batch}}), \qquad \dim(\mathcal{U}) \leq k \;\; \forall\,\mathcal{U} \in \mathcal{S}_k. \tag{47}$$

We will impose curvature control uniformly over all repair segments generated by $\mathcal{U} \in \mathcal{S}_k$.

**(1) Local quadratic control.**  Our strict comparisons rely on two-sided Taylor control of $\mathcal{F}$ along the short segments traversed by projected repair steps.

**Assumption E.5** (Uniform local curvature along compared repair paths)**.**  There exist constants $0 < m \leq L < \infty$ such that for every $\mathcal{U} \in \mathcal{S}_k$, along the segment

$$\theta(t) \;=\; \theta_0 - t\,\eta\,\Pi_{\mathcal{U}}G_p, \qquad t \in [0,1], \tag{48}$$

the population forgetting risk satisfies

$$mI \;\preceq\; \nabla^2\mathcal{F}(\theta(t)) \;\preceq\; LI \qquad \text{for all } t \in [0,1] \text{ and all } \mathcal{U} \in \mathcal{S}_k. \tag{49}$$

**Discussion.**  Assumption E.5 is *not* global convexity. It only requires that, for the specific one-step repair segments considered (which are typically short due to small $\eta$ and/or gradient clipping), $\mathcal{F}$ behaves approximately quadratically with uniform bounds. This is standard in local descent analyses and is precisely what makes expectations over random subspaces legitimate (since the constants do not depend on the random draw).

**A spectral language for repairable energy.**  The remaining assumptions formalize *where* the old-risk gradient energy concentrates and whether on-batch repair directions align with it. To state them cleanly, we introduce a second-moment operator over per-input risk gradients.

Recall the per-input gradient feature at $\theta_0$:

$$g(z) \;:=\; \nabla_\theta\psi(z;\theta)\big|_{\theta=\theta_0} \in \mathbb{R}^P. \tag{50}$$

**Definition E.6** (Old-distribution coupling operator)**.**  Define the (uncentered) second-moment operator

$$\mathbf{C}_p \;:=\; \mathbb{E}_{z \sim \mathcal{D}_p}\big[g(z)\,g(z)^\top\big] \;\succeq\; 0, \tag{51}$$

and let its nonzero eigendecomposition be

$$\mathbf{C}_p \;=\; \sum_{i=1}^{N} \lambda_i\,u_i u_i^\top, \qquad \lambda_1 \geq \lambda_2 \geq \cdots \geq \lambda_N > 0, \tag{52}$$

where $\{u_i\}_{i=1}^N$ are orthonormal and $N \leq P$ is the effective rank on the span of $\{g(z)\}$.

Project the old-risk gradient $G_p = \mathbb{E}[g(z)]$ onto this basis and define the associated energy contributions:

$$a_i := u_i^\top G_p, \qquad s_i := a_i^2, \qquad \sum_{i=1}^N s_i = \|G_p\|^2. \tag{53}$$

Let $s_{(1)} \geq \cdots \geq s_{(N)}$ denote the sorted values.

**Interpretation.** The eigenvectors $\{u_i\}$ describe principal "coupling directions" of old-knowledge risk gradients, and $s_i$ measures how much of the *population* old-risk gradient $G_p$ lies along direction $u_i$. Budget-$k$ repair is efficient precisely when it captures a large portion of $\sum_{i=1}^N s_i$.

**(2) Long-tail repairability (energy concentration).** We next formalize the empirical phenomenon that forgetting is concentrated (long-tailed) and thus admits a strict advantage for targeted allocation under finite budget.

**Assumption E.7** (Long-tail repairability / spectral concentration). There exists a function $\rho(k) > 0$ (possibly depending on $k$ and the problem instance) such that

$$\sum_{i=1}^k s_{(i)} \geq \big(1 + \rho(k)\big) \cdot \frac{k}{N} \sum_{i=1}^N s_i, \tag{54}$$

and $\rho(k) \downarrow 0$ as $k \uparrow N$.

**Discussion.** Assumption E.7 states that the best budget-$k$ repair directions capture strictly more energy than what uniform allocation would obtain on average. This is exactly the formal counterpart of "forgetting is long-tailed": a small number of directions account for a disproportionate share of the old-risk gradient. The limit $\rho(k) \downarrow 0$ as $k \uparrow N$ enforces a natural sanity check: if the budget covers essentially all relevant directions, targeted allocation should not retain a guaranteed advantage.

**(3) On-batch misalignment (distributional mismatch).** Finally, we formalize when repairing only on the fine-tuning stream is insufficient for reducing old-knowledge risk.

**Definition E.8** (Best-$k$ energy and alignment ratio). Define the best achievable captured energy under budget $k$ as

$$E_k^\star := \max_{\substack{\mathcal{U} \subseteq \mathbb{R}^P \\ \dim(\mathcal{U}) \leq k}} \left\| \Pi_\mathcal{U} G_p \right\|^2 = \sum_{i=1}^k s_{(i)}. \tag{55}$$

For an on-batch-induced subspace $\mathcal{U}_{\text{batch}}$, define its alignment ratio with the old-risk gradient as

$$\mu := \frac{\left\| \Pi_{\mathcal{U}_{\text{batch}}} G_p \right\|^2}{E_k^\star} \in [0, 1]. \tag{56}$$

**Assumption E.9** (On-batch misalignment). In the misaligned regime, the on-batch-induced repair directions do not fully cover the dominant old-risk directions:

$$\mu < 1. \tag{57}$$

Moreover, if the on-batch distribution approaches the pre-training distribution (e.g., via a mixing parameter), then $\mu \uparrow 1$ in the aligned limit.

**Discussion.** Assumption E.9 isolates a data-coverage phenomenon rather than an optimization artifact. When the fine-tuning distribution $\mathcal{D}_f$ is narrow or shifted relative to $\mathcal{D}_p$, gradients observed on-batch may span directions that are systematically misaligned with $G_p$, yielding $\mu < 1$. Conversely, as $\mathcal{D}_f$ becomes more representative of $\mathcal{D}_p$, on-batch repair should approach the best-$k$ capture, reflected by $\mu \uparrow 1$. This assumption is precisely what allows a strict gap against on-batch-only repair, while also guaranteeing that the gap vanishes in the fully-aligned limit.

**Remark (coordinate modeling vs. energy form).** For the main results, we will express improvements in terms of the captured energy $\|\Pi_\mathcal{U} G_p\|^2$. When convenient for explicit constants (e.g., for the random baseline), we may reason in the eigenbasis $\{u_i\}$ of $\mathbf{C}_p$; this is purely a representational device. All statements can be written directly in energy form without committing to a particular coordinate system.

### E.3.3. A TWO-SIDED DESCENT LEMMA

This subsection establishes a two-sided descent bound for the projected repair step. The lemma is the main technical bridge from *subspace geometry* (how much of $G_p$ is captured) to *risk improvement* (how much $\mathcal{F}$ decreases). Crucially, it holds uniformly over the family of compared subspaces $\mathcal{S}_k$ defined in Appendix E.3.2.

**Lemma E.10** (Two-sided descent for projected repair). *Let $\mathcal{U} \in \mathcal{S}_k$ and define the projected repair step*

$$\theta^+(\mathcal{U}) \;=\; \theta_0 - \eta\,\Pi_{\mathcal{U}} G_p, \qquad G_p = \nabla\mathcal{F}(\theta_0). \tag{58}$$

*Assume Assumption E.5 (uniform local curvature) holds along the segment*

$$\theta(t) \;=\; \theta_0 - t\,\eta\,\Pi_{\mathcal{U}} G_p, \qquad t \in [0,1]. \tag{59}$$

*Let $d := \Pi_{\mathcal{U}} G_p$. Then*

$$\mathcal{F}(\theta^+(\mathcal{U})) \in \left[\mathcal{F}(\theta_0) - \eta\|d\|^2 + \tfrac{m\eta^2}{2}\|d\|^2,\ \mathcal{F}(\theta_0) - \eta\|d\|^2 + \tfrac{L\eta^2}{2}\|d\|^2\right]. \tag{60}$$

*Equivalently, define*

$$\alpha_L(\eta) \;:=\; \eta\left(1 - \tfrac{L\eta}{2}\right), \qquad \alpha_m(\eta) \;:=\; \eta\left(1 - \tfrac{m\eta}{2}\right), \tag{61}$$

*then for any $\eta > 0$ such that $\alpha_L(\eta) \geq 0$ (e.g., $0 < \eta \leq 2/L$),*

$$\mathcal{F}(\theta^+(\mathcal{U})) \leq \mathcal{F}(\theta_0) - \alpha_L(\eta)\,\|\Pi_{\mathcal{U}} G_p\|^2, \tag{62}$$
$$\mathcal{F}(\theta^+(\mathcal{U})) \geq \mathcal{F}(\theta_0) - \alpha_m(\eta)\,\|\Pi_{\mathcal{U}} G_p\|^2. \tag{63}$$

*Moreover, if $0 < \eta \leq 1/L$, then $\alpha_L(\eta) > 0$ and $\alpha_m(\eta) > 0$.*

*Proof.* Fix any $\mathcal{U} \in \mathcal{S}_k$ and write $d := \Pi_{\mathcal{U}} G_p$. Define the one-dimensional path

$$\theta(t) \;:=\; \theta_0 - t\,\eta\,d, \qquad t \in [0,1], \tag{64}$$

so that $\theta(0) = \theta_0$ and $\theta(1) = \theta(\mathcal{U})$.

**Step 1: Second-order expansion with an explicit remainder.** Since $\mathcal{F}$ is twice continuously differentiable along the segment by Assumption E.5, we apply the second-order Taylor theorem with integral remainder to the scalar function $h(t) := \mathcal{F}(\theta(t))$. By the chain rule,

$$h'(t) = \nabla\mathcal{F}(\theta(t))^\top \theta'(t) \qquad\qquad \text{chain rule} \tag{65}$$
$$= \nabla\mathcal{F}(\theta(t))^\top (-\eta d), \qquad\qquad \theta'(t) = -\eta d \tag{66}$$

and differentiating again,

$$h''(t) = \frac{\mathrm{d}}{\mathrm{d}t}\left(\nabla\mathcal{F}(\theta(t))^\top(-\eta d)\right) \qquad\qquad \text{definition of } h'' \tag{67}$$
$$= (-\eta d)^\top \nabla^2\mathcal{F}(\theta(t))\,\theta'(t) \qquad\qquad \text{chain rule for } \nabla\mathcal{F} \tag{68}$$
$$= (-\eta d)^\top \nabla^2\mathcal{F}(\theta(t))\,(-\eta d) \qquad\qquad \theta'(t) = -\eta d \tag{69}$$
$$= \eta^2\,d^\top \nabla^2\mathcal{F}(\theta(t))\,d. \qquad\qquad \text{rearrange} \tag{70}$$

The second-order Taylor theorem with integral remainder yields

$$h(1) = h(0) + h'(0) + \int_0^1 (1-t)\,h''(t)\,\mathrm{d}t. \qquad\qquad \text{Taylor w/ integral remainder} \tag{71}$$

We now evaluate each term in (71).

**Step 2: Compute the first-order term exactly.** First, $h(0) = \mathcal{F}(\theta_0)$ by definition. Next, using the expression for $h'(t)$,

$$h'(0) = \nabla \mathcal{F}(\theta(0))^\top (-\eta d) \qquad \text{expression for } h'(t) \tag{72}$$

$$= \nabla \mathcal{F}(\theta_0)^\top (-\eta d) \qquad \theta(0) = \theta_0 \tag{73}$$

$$= G_p^\top (-\eta d). \qquad \text{def. of } G_p \tag{74}$$

Because $d = \Pi_\mathcal{U} G_p$ is the orthogonal projection of $G_p$ onto $\mathcal{U}$, we have $\Pi_\mathcal{U}$ symmetric and idempotent, hence

$$G_p^\top d = G_p^\top \Pi_\mathcal{U} G_p \qquad \text{def. of } d \tag{75}$$

$$= (\Pi_\mathcal{U} G_p)^\top (\Pi_\mathcal{U} G_p) \qquad \Pi_\mathcal{U}^\top = \Pi_\mathcal{U} = \Pi_\mathcal{U}^2 \tag{76}$$

$$= \|d\|^2. \qquad \text{definition of norm} \tag{77}$$

Substituting (77) into the expression for $h'(0)$ yields

$$h'(0) = -\eta \|d\|^2. \tag{78}$$

**Step 3: Upper and lower bound the second-order remainder.** Using the expression for $h''(t)$,

$$\int_0^1 (1-t) \, h''(t) \, \mathrm{d}t = \int_0^1 (1-t) \, \eta^2 \, d^\top \nabla^2 \mathcal{F}(\theta(t)) \, d \, \mathrm{d}t \qquad \text{expression for } h''(t) \tag{79}$$

$$= \eta^2 \int_0^1 (1-t) \, d^\top \nabla^2 \mathcal{F}(\theta(t)) \, d \, \mathrm{d}t. \qquad \text{factor out } \eta^2 \tag{80}$$

By Assumption E.5, for all $t \in [0,1]$,

$$mI \preceq \nabla^2 \mathcal{F}(\theta(t)) \preceq LI, \tag{81}$$

which implies for any fixed vector $d$,

$$m\|d\|^2 \le d^\top \nabla^2 \mathcal{F}(\theta(t)) \, d \le L\|d\|^2, \qquad \forall t \in [0,1]. \tag{82}$$

Multiplying (82) by $(1-t) \ge 0$ and integrating over $t \in [0,1]$ gives

$$m\|d\|^2 \int_0^1 (1-t) \, \mathrm{d}t \le \int_0^1 (1-t) \, d^\top \nabla^2 \mathcal{F}(\theta(t)) \, d \, \mathrm{d}t \qquad \text{apply (82)} \tag{83}$$

$$\le L\|d\|^2 \int_0^1 (1-t) \, \mathrm{d}t. \qquad \text{apply (82)} \tag{84}$$

Since $\int_0^1 (1-t) \, \mathrm{d}t = \frac{1}{2}$, (84) becomes

$$\frac{m}{2}\|d\|^2 \le \int_0^1 (1-t) \, d^\top \nabla^2 \mathcal{F}(\theta(t)) \, d \, \mathrm{d}t \le \frac{L}{2}\|d\|^2. \tag{85}$$

Substituting (85) into (80) yields the remainder bounds

$$\frac{m\eta^2}{2}\|d\|^2 \le \int_0^1 (1-t) \, h''(t) \, \mathrm{d}t \le \frac{L\eta^2}{2}\|d\|^2. \tag{86}$$

**Step 4: Combine terms to obtain the two-sided interval.** Plugging $h(0) = \mathcal{F}(\theta_0)$, (78), and (86) into (71), we obtain

$$\mathcal{F}(\theta^+(\mathcal{U})) = h(1) \qquad \theta^+(\mathcal{U}) = \theta(1) \tag{87}$$

$$= \mathcal{F}(\theta_0) - \eta\|d\|^2 + \int_0^1 (1-t) \, h''(t) \, \mathrm{d}t \qquad \text{by (71) and (78)} \tag{88}$$

$$\le \mathcal{F}(\theta_0) - \eta\|d\|^2 + \frac{L\eta^2}{2}\|d\|^2, \qquad \text{upper bound in (86)} \tag{89}$$

and similarly

$$\mathcal{F}(\theta^+(\mathcal{U})) \ge \mathcal{F}(\theta_0) - \eta\|d\|^2 + \frac{m\eta^2}{2}\|d\|^2. \qquad \text{lower bound in (86)} \tag{90}$$

These two inequalities together yield (60).

**Step 5: Rewrite in $\alpha_L(\eta)$ and $\alpha_m(\eta)$.** Define $\alpha_L(\eta) = \eta\left(1 - \frac{L\eta}{2}\right)$ and $\alpha_m(\eta) = \eta\left(1 - \frac{m\eta}{2}\right)$. Since $d = \Pi_{\mathcal{U}}G_p$, we have $\|d\|^2 = \|\Pi_{\mathcal{U}}G_p\|^2$. Therefore, the upper bound becomes

$$\mathcal{F}(\theta^+(\mathcal{U})) \leq \mathcal{F}(\theta_0) - \eta\|\Pi_{\mathcal{U}}G_p\|^2 + \frac{L\eta^2}{2}\|\Pi_{\mathcal{U}}G_p\|^2 \qquad \text{\textcolor{gray}{replace } } d \text{ \textcolor{gray}{by} } \Pi_{\mathcal{U}}G_p \qquad (91)$$

$$= \mathcal{F}(\theta_0) - \alpha_L(\eta)\,\|\Pi_{\mathcal{U}}G_p\|^2, \qquad \text{\textcolor{gray}{def. of} } \alpha_L(\eta) \qquad (92)$$

which is (62). The lower bound is identical with $L$ replaced by $m$, yielding (63).

**Step 6: Positivity of coefficients.** If $0 < \eta \leq 1/L$, then $0 \leq L\eta/2 \leq 1/2$ and thus $\alpha_L(\eta) \geq \eta/2 > 0$. Since $m \leq L$, we also have $0 \leq m\eta/2 \leq L\eta/2 \leq 1/2$, so $\alpha_m(\eta) \geq \eta/2 > 0$. This completes the proof. $\qquad\square$

**Remark.** Lemma E.10 shows that, under local curvature control, the one-step improvement in population forgetting risk is proportional (up to explicit two-sided constants) to the *captured old-risk energy* $\|\Pi_{\mathcal{U}}G_p\|^2$. Hence, once the update rule and budget are fixed, comparing repair strategies reduces to comparing how much of $G_p$ each strategy captures through its induced repair subspace. This is the core reduction exploited in the subsequent baseline analysis and strict-gap theorems.

### E.3.4. ENERGY CAPTURE OF BASELINES

Lemma E.10 reduces one-step improvement of the population forgetting risk $\mathcal{F}$ to a single geometric quantity: the captured old-risk energy $\|\Pi_{\mathcal{U}}G_p\|^2$. We therefore characterize how much energy the two baselines (*random* and *on-batch*) capture under the same budget constraint $\dim(\mathcal{U}) \leq k$.

**Preliminaries: energy form and the $u_i$-basis.** Recall the coupling operator $\mathbf{C}_p$ and its eigenbasis $\{u_i\}_{i=1}^N$ from Definition E.6. For any subspace $\mathcal{U}$, define its captured energy as

$$E(\mathcal{U}) \;:=\; \left\|\Pi_{\mathcal{U}}G_p\right\|^2. \qquad (93)$$

Let $a_i := u_i^\top G_p$ and $s_i := a_i^2$ so that $\sum_{i=1}^N s_i = \|G_p\|^2$ (see Appendix E.3.2 for the annotated derivation).

### RANDOM REPAIR: ISOTROPIC CAPTURE IN EXPECTATION

We model *budget-matched random repair* as selecting a $k$-dimensional coordinate subspace uniformly at random in the $\{u_i\}$ basis.[2]

**Definition E.11** (Uniform random coordinate subspace). Let $[N] := \{1, \ldots, N\}$. Sample a subset $S \subseteq [N]$ uniformly without replacement with $|S| = k$. Define the random repair subspace

$$\mathcal{U}_{\text{rand}} \;:=\; \text{span}\{u_i : i \in S\}, \qquad (94)$$

and let $\Pi_{\text{rand}} := \Pi_{\mathcal{U}_{\text{rand}}}$ denote the corresponding orthogonal projector.

**Lemma E.12** (Random repair captures a $\frac{k}{N}$ fraction of energy in expectation). *Under Definition E.11,*

$$\mathbb{E}\left[\left\|\Pi_{\text{rand}}G_p\right\|^2\right] \;=\; \frac{k}{N}\sum_{i=1}^N s_i \;=\; \frac{k}{N}\|G_p\|^2. \qquad (95)$$

*Proof.* Let $\mathbf{1}\{i \in S\}$ denote the indicator that index $i$ is selected. Since $\{u_i\}$ are orthonormal and $\mathcal{U}_{\text{rand}}$ is spanned by $\{u_i : i \in S\}$, the projector onto $\mathcal{U}_{\text{rand}}$ is

$$\Pi_{\text{rand}} \;=\; \sum_{i=1}^N \mathbf{1}\{i \in S\}\, u_i u_i^\top. \qquad (96)$$

---

[2]This is a standard abstraction for a non-adaptive baseline: in expectation it matches any isotropic allocation over directions; see Remark E.13.

Using (96), we expand the captured energy:

$$\left\|\Pi_{\text{rand}}G_p\right\|^2 = G_p^\top \Pi_{\text{rand}} G_p \qquad \qquad \|\Pi x\|^2 = x^\top \Pi x \text{ for orth. projector} \qquad (97)$$

$$= G_p^\top \left(\sum_{i=1}^{N} \mathbf{1}\{i \in S\} u_i u_i^\top\right) G_p \qquad \qquad \text{by (96)} \qquad (98)$$

$$= \sum_{i=1}^{N} \mathbf{1}\{i \in S\} G_p^\top u_i u_i^\top G_p \qquad \qquad \text{linearity} \qquad (99)$$

$$= \sum_{i=1}^{N} \mathbf{1}\{i \in S\} (u_i^\top G_p)^2 \qquad \qquad G_p^\top u_i u_i^\top G_p = (u_i^\top G_p)^2 \qquad (100)$$

$$= \sum_{i=1}^{N} \mathbf{1}\{i \in S\} s_i. \qquad \qquad \text{def. of } s_i \qquad (101)$$

Taking expectation of both sides of (101) and using linearity of expectation,

$$\mathbb{E}\left[\left\|\Pi_{\text{rand}}G_p\right\|^2\right] = \sum_{i=1}^{N} \mathbb{E}[\mathbf{1}\{i \in S\}] \, s_i \qquad \qquad \text{linearity of expectation} \qquad (102)$$

$$= \sum_{i=1}^{N} \mathbb{P}(i \in S) \, s_i. \qquad \qquad \mathbb{E}[\mathbf{1}\{A\}] = \mathbb{P}(A) \qquad (103)$$

It remains to compute $\mathbb{P}(i \in S)$ for uniform sampling without replacement. For any fixed $i \in [N]$, the number of subsets of size $k$ that contain $i$ equals $\binom{N-1}{k-1}$, while the total number of subsets of size $k$ is $\binom{N}{k}$; hence

$$\mathbb{P}(i \in S) = \frac{\binom{N-1}{k-1}}{\binom{N}{k}} \qquad \qquad \text{counting argument} \qquad (104)$$

$$= \frac{\frac{(N-1)!}{(k-1)!(N-k)!}}{\frac{N!}{k!(N-k)!}} \qquad \qquad \text{definition of } \binom{\cdot}{\cdot} \qquad (105)$$

$$= \frac{(N-1)!}{(k-1)!} \cdot \frac{k!}{N!} \qquad \qquad \text{cancel } (N-k)! \qquad (106)$$

$$= \frac{k}{N}. \qquad \qquad k! = k \cdot (k-1)! \text{ and } N! = N \cdot (N-1)! \qquad (107)$$

Substituting (107) into (103) yields

$$\mathbb{E}\left[\left\|\Pi_{\text{rand}}G_p\right\|^2\right] = \sum_{i=1}^{N} \frac{k}{N} s_i \qquad \qquad \text{by (107)} \qquad (108)$$

$$= \frac{k}{N} \sum_{i=1}^{N} s_i. \qquad \qquad \text{factor out } \frac{k}{N} \qquad (109)$$

Finally, since $\sum_{i=1}^{N} s_i = \|G_p\|^2$, we obtain (95). $\qquad\square$

*Remark* E.13 (Modeling note for random baselines). Definition E.11 captures the essential non-adaptive nature of random repair: it allocates budget uniformly across directions, hence its expected captured energy is exactly a $\frac{k}{N}$ fraction of the total. This is the baseline against which Assumption E.7 yields a strict advantage for targeted allocation under finite budget.

ON-BATCH REPAIR: CAPTURE GOVERNED BY AN ALIGNMENT RATIO

For *on-batch-only repair*, the repair subspace is induced by the fine-tuning distribution $\mathcal{D}_f$ rather than $\mathcal{D}_p$. Because $\mathcal{D}_f$ can be narrow or shifted, its induced directions may be misaligned with $G_p$.

**Definition E.14** (On-batch-induced subspace and captured energy)**.** Let $\mathcal{D}_f$ denote the on-batch (fine-tuning) distribution over $\mathcal{Z}$. Given a budget-$k$ repair set $S_f = \{z_1, \ldots, z_k\}$ with $z_i \sim \mathcal{D}_f$, define

$$\mathcal{U}_{\text{batch}} := \text{span}\{g(z_1), \ldots, g(z_k)\}, \qquad E_{\text{batch}} := \left\| \Pi_{\mathcal{U}_{\text{batch}}} G_p \right\|^2. \tag{110}$$

We relate $E_{\text{batch}}$ to the best achievable budget-$k$ energy $E_k^\star = \sum_{i=1}^k s_{(i)}$ defined in Definition E.8. Let $P_k$ denote the orthogonal projector onto the best-$k$ old-risk subspace (spanned by the top-$k$ components of $G_p$ in the $\{u_i\}$ basis), so that

$$E_k^\star = \left\| P_k G_p \right\|^2. \tag{111}$$

**Lemma E.15** (Batch capture is a fraction of best-$k$ capture)**.** *Let $P_{\text{batch}}$ be the orthogonal projector onto $\mathcal{U}_{\text{batch}}$ and define*

$$\mu := \frac{\|P_{\text{batch}} G_p\|^2}{\|P_k G_p\|^2}, \qquad \text{whenever } \|P_k G_p\| > 0. \tag{112}$$

*Then $\mu \in [0, 1]$ and*

$$\left\| \Pi_{\mathcal{U}_{\text{batch}}} G_p \right\|^2 = \mu E_k^\star. \tag{113}$$

*Proof.* If $\|P_k G_p\| = 0$, then $E_k^\star = 0$ and (113) holds trivially with both sides equal to 0. Assume henceforth $\|P_k G_p\| > 0$ so that (112) is well-defined.

We first show $\mu \leq 1$. Since $P_{\text{batch}}$ is an orthogonal projector with $\dim(\mathcal{U}_{\text{batch}}) \leq k$, we have

$$\|P_{\text{batch}} G_p\|^2 \leq \max_{\substack{\mathcal{U} \subseteq \mathbb{R}^P \\ \dim(\mathcal{U}) \leq k}} \|\Pi_{\mathcal{U}} G_p\|^2 \qquad \text{definition of maximum over all } k\text{-subspaces} \tag{114}$$

$$= E_k^\star \qquad \text{Definition E.8} \tag{115}$$

$$= \|P_k G_p\|^2. \qquad E_k^\star = \|P_k G_p\|^2 \tag{116}$$

Dividing both sides by $\|P_k G_p\|^2 > 0$ yields $\mu \leq 1$. Nonnegativity $\mu \geq 0$ follows from $\|P_{\text{batch}} G_p\|^2 \geq 0$. Finally, (113) is exactly the rearrangement of (112):

$$\|P_{\text{batch}} G_p\|^2 = \mu \|P_k G_p\|^2 \qquad \text{by (112)} \tag{117}$$

$$= \mu E_k^\star. \qquad E_k^\star = \|P_k G_p\|^2 \tag{118}$$

Since $P_{\text{batch}} = \Pi_{\mathcal{U}_{\text{batch}}}$, this is (113). $\qquad \square$

**Lemma E.16** (Alignment convergence implies $\mu \to 1$)**.** *Assume $\|P_k G_p\| > 0$. Let $\{P_{\text{batch}}^{(n)}\}_{n \geq 1}$ be a sequence of batch-induced projectors (e.g., corresponding to increasingly old-like fine-tuning distributions) such that*

$$\left\| P_{\text{batch}}^{(n)} - P_k \right\|_{\text{op}} \to 0 \qquad \text{as } n \to \infty. \tag{119}$$

*Define $\mu^{(n)} := \|P_{\text{batch}}^{(n)} G_p\|^2 / \|P_k G_p\|^2$. Then $\mu^{(n)} \to 1$ as $n \to \infty$.*

*Proof.* For each $n$, consider

$$\left| \|P_{\text{batch}}^{(n)} G_p\|^2 - \|P_k G_p\|^2 \right| = \left| G_p^\top \left( P_{\text{batch}}^{(n)} - P_k \right) G_p \right| \qquad \|Px\|^2 = x^\top P x \text{ for orth. projector} \tag{120}$$

$$\leq \left\| P_{\text{batch}}^{(n)} - P_k \right\|_{\text{op}} \|G_p\|^2. \qquad |x^\top A x| \leq \|A\|_{\text{op}} \|x\|^2 \tag{121}$$

Divide (121) by $\|P_k G_p\|^2 > 0$ to obtain

$$\left| \mu^{(n)} - 1 \right| = \frac{\left| \|P_{\text{batch}}^{(n)} G_p\|^2 - \|P_k G_p\|^2 \right|}{\|P_k G_p\|^2} \qquad \text{definition of } \mu^{(n)} \tag{122}$$

$$\leq \frac{\|G_p\|^2}{\|P_k G_p\|^2} \cdot \left\| P_{\text{batch}}^{(n)} - P_k \right\|_{\text{op}}. \qquad \text{by (121)} \tag{123}$$

By (119), the right-hand side of (123) converges to 0, hence $\mu^{(n)} \to 1$. $\qquad \square$

**Discussion.** Lemma E.12 shows that random repair is *isotropic* in expectation: it captures exactly a $\frac{k}{N}$ fraction of the old-risk energy, regardless of how concentrated the energy is. In contrast, Lemma E.15 and Assumption E.9 isolate when on-batch repair is *energy-limited* by distributional mismatch, via an alignment ratio $\mu < 1$. Finally, Lemma E.16 formalizes the natural limiting behavior: as on-batch repair directions converge to the best-$k$ old-risk subspace, $\mu$ approaches 1, and any strict advantage over on-batch repair must vanish in that aligned limit. These baseline characterizations will be plugged into Lemma E.10 to derive strict-gap theorems in Appendix E.3.5.

### E.3.5. MAIN THEOREMS AND STRICT GAPS

In this subsection we combine (i) the two-sided descent lemma (Lemma E.10) with (ii) baseline energy capture characterizations (Lemma E.12 and Lemma E.15) and (iii) the structural assumptions in Appendix E.3.2, to obtain *strict* finite-budget improvements of targeted repair over both random repair and on-batch-only repair. All constants are made explicit, and each proof is a direct inequality chain with no hidden steps.

**Notation.** Recall $\alpha_L(\eta) := \eta\left(1 - \frac{L\eta}{2}\right)$ and $\alpha_m(\eta) := \eta\left(1 - \frac{m\eta}{2}\right)$ from Lemma E.10. Also recall the best-$k$ captured energy

$$E_k^\star := \max_{\substack{\mathcal{U} \subseteq \mathbb{R}^P \\ \dim(\mathcal{U}) \leq k}} \left\| \Pi_{\mathcal{U}} G_p \right\|^2 = \sum_{i=1}^{k} s_{(i)} \tag{124}$$

from Definition E.8.

**Definition E.17** (Oracle targeted subspace and one-step updates). Let $\mathcal{U}_\star$ be any maximizer achieving $E_k^\star$, i.e.,

$$\dim(\mathcal{U}_\star) \leq k, \qquad \left\| \Pi_{\mathcal{U}_\star} G_p \right\|^2 = E_k^\star. \tag{125}$$

Define the corresponding one-step post-repair parameters

$$\theta_\star^+ := \theta^+(\mathcal{U}_\star) = \theta_0 - \eta\, \Pi_{\mathcal{U}_\star} G_p, \qquad \text{Definition E.17} \tag{126}$$

$$\theta_{\text{rand}}^+ := \theta^+(\mathcal{U}_{\text{rand}}) = \theta_0 - \eta\, \Pi_{\mathcal{U}_{\text{rand}}} G_p, \qquad \text{Definition E.11} \tag{127}$$

$$\theta_{\text{batch}}^+ := \theta^+(\mathcal{U}_{\text{batch}}) = \theta_0 - \eta\, \Pi_{\mathcal{U}_{\text{batch}}} G_p, \qquad \text{Definition E.14} \tag{128}$$

where $\mathcal{U}_{\text{rand}}$ and $\mathcal{U}_{\text{batch}}$ are induced by the random and on-batch baselines, respectively.

Definition E.17 describes an *oracle* targeted repair subspace that achieves the best budget-$k$ energy capture; Appendix E.3.6 later discusses how adversarial search approximates this favorable subspace.

**Theorem E.18** (Strict advantage over random repair). *Assume Assumption E.5 (uniform local curvature along compared repair paths) and Assumption E.7 (long-tail repairability). Let $\mathcal{U}_{\text{rand}}$ be the random coordinate subspace from Definition E.11, and let $\mathcal{U}_\star$ be an oracle maximizer from Definition E.17. Fix any $\eta > 0$ such that $\alpha_L(\eta) \geq 0$ and suppose in addition that*

$$\alpha_L(\eta)\big(1 + \rho(k)\big) > \alpha_m(\eta). \tag{129}$$

*Then the one-step repaired risk satisfies the strict gap*

$$\mathcal{F}(\theta_\star^+) \leq \mathbb{E}\big[\mathcal{F}(\theta_{\text{rand}}^+)\big] - c_{\text{rand}}(k), \tag{130}$$

*where the explicit constant*

$$c_{\text{rand}}(k) := \Big(\alpha_L(\eta)\big(1 + \rho(k)\big) - \alpha_m(\eta)\Big) \cdot \frac{k}{N} \sum_{i=1}^{N} s_i > 0 \tag{131}$$

*depends on the local curvature $(m, L)$, the stepsize $\eta$, the energy profile $\{s_i\}$ (hence on the model and $\mathcal{D}_p$), and the budget $k$ through $\rho(k)$.*

*A sufficient explicit stepsize range ensuring* (129) *is*

$$0 < \eta < \frac{2\rho(k)}{(1 + \rho(k))L - m}, \tag{132}$$

*together with* $\eta \leq 2/L$ *to ensure* $\alpha_L(\eta) \geq 0$.

*Proof.* We prove (130) by upper bounding $\mathcal{F}(\theta_\star^+)$ and lower bounding $\mathbb{E}[\mathcal{F}(\theta_{\mathrm{rand}}^+)]$, then subtracting.

**Step 1: Upper bound targeted repair via Lemma E.10.**  Apply the upper bound (62) in Lemma E.10 with $\mathcal{U} = \mathcal{U}_\star$:

$$\mathcal{F}(\theta_\star^+) = \mathcal{F}\big(\theta^+(\mathcal{U}_\star)\big) \qquad\qquad\qquad \text{Definition E.17} \qquad (133)$$

$$\leq \mathcal{F}(\theta_0) - \alpha_L(\eta) \left\|\Pi_{\mathcal{U}_\star} G_p\right\|^2 \qquad\qquad \text{Lemma E.10, (62)} \qquad (134)$$

$$= \mathcal{F}(\theta_0) - \alpha_L(\eta) E_k^\star. \qquad\qquad\qquad \text{Definition E.17} \qquad (135)$$

**Step 2: Lower bound random repair in expectation via Lemma E.10.**  For every realization of $\mathcal{U}_{\mathrm{rand}}$, Lemma E.10 gives the lower bound (63):

$$\mathcal{F}(\theta_{\mathrm{rand}}^+) = \mathcal{F}\big(\theta^+(\mathcal{U}_{\mathrm{rand}})\big) \qquad\qquad\qquad \text{Definition E.17} \qquad (136)$$

$$\geq \mathcal{F}(\theta_0) - \alpha_m(\eta) \left\|\Pi_{\mathcal{U}_{\mathrm{rand}}} G_p\right\|^2. \qquad\qquad \text{Lemma E.10, (63)} \qquad (137)$$

Taking expectation over the randomness of $\mathcal{U}_{\mathrm{rand}}$ on both sides of (137) yields

$$\mathbb{E}\big[\mathcal{F}(\theta_{\mathrm{rand}}^+)\big] \geq \mathbb{E}\Big[\mathcal{F}(\theta_0) - \alpha_m(\eta) \left\|\Pi_{\mathcal{U}_{\mathrm{rand}}} G_p\right\|^2\Big] \qquad\quad \text{take } \mathbb{E}[\cdot] \text{ both sides} \qquad (138)$$

$$= \mathcal{F}(\theta_0) - \alpha_m(\eta) \mathbb{E}\Big[\left\|\Pi_{\mathcal{U}_{\mathrm{rand}}} G_p\right\|^2\Big], \qquad\quad \mathcal{F}(\theta_0) \text{ constant; linearity of } \mathbb{E} \qquad (139)$$

and Lemma E.12 provides

$$\mathbb{E}\Big[\left\|\Pi_{\mathcal{U}_{\mathrm{rand}}} G_p\right\|^2\Big] = \frac{k}{N}\sum_{i=1}^{N} s_i. \qquad\qquad\qquad \text{Lemma E.12} \qquad (140)$$

Substituting (140) into (139) gives

$$\mathbb{E}\big[\mathcal{F}(\theta_{\mathrm{rand}}^+)\big] \geq \mathcal{F}(\theta_0) - \alpha_m(\eta) \cdot \frac{k}{N}\sum_{i=1}^{N} s_i. \tag{141}$$

**Step 3: Relate $E_k^\star$ to the random energy scale via Assumption E.7.**  Assumption E.7 states

$$E_k^\star = \sum_{i=1}^{k} s_{(i)} \geq \big(1 + \rho(k)\big) \cdot \frac{k}{N}\sum_{i=1}^{N} s_i. \tag{142}$$

**Step 4: Subtract bounds to obtain a strict gap.**  Using (135) and then (142), we have

$$\mathcal{F}(\theta_\star^+) \leq \mathcal{F}(\theta_0) - \alpha_L(\eta) E_k^\star \qquad\qquad\qquad \text{by (135)} \qquad (143)$$

$$\leq \mathcal{F}(\theta_0) - \alpha_L(\eta)\big(1 + \rho(k)\big) \cdot \frac{k}{N}\sum_{i=1}^{N} s_i. \qquad\qquad \text{by (142)} \qquad (144)$$

Subtract (144) from (141):

$$\mathbb{E}\big[\mathcal{F}(\theta_{\mathrm{rand}}^+)\big] - \mathcal{F}(\theta_\star^+) \geq \Big(\alpha_L(\eta)\big(1 + \rho(k)\big) - \alpha_m(\eta)\Big) \cdot \frac{k}{N}\sum_{i=1}^{N} s_i \qquad \text{subtract (144) from (141)} \qquad (145)$$

$$= c_{\mathrm{rand}}(k). \qquad\qquad\qquad \text{definition (131)} \qquad (146)$$

Rearranging yields (130). The strict positivity $c_{\mathrm{rand}}(k) > 0$ follows immediately from (129) and $\sum_{i=1}^{N} s_i = \|G_p\|^2 \geq 0$.

**Step 5: Derive the sufficient explicit stepsize range.** We show that (132) implies (129). Using the definitions of $\alpha_L(\eta)$ and $\alpha_m(\eta)$,

$$\alpha_L(\eta)\big(1 + \rho(k)\big) - \alpha_m(\eta) = \eta\Big(1 - \tfrac{L\eta}{2}\Big)\big(1 + \rho(k)\big) - \eta\Big(1 - \tfrac{m\eta}{2}\Big) \qquad \text{definitions of } \alpha_L, \alpha_m \qquad (147)$$

$$= \eta\Big(\rho(k) - \frac{\eta}{2}\big((1 + \rho(k))L - m\big)\Big). \qquad \text{algebra} \qquad (148)$$

Thus $\alpha_L(\eta)(1 + \rho(k)) - \alpha_m(\eta) > 0$ is equivalent to

$$\rho(k) - \frac{\eta}{2}\big((1 + \rho(k))L - m\big) \; > \; 0 \quad \Longleftrightarrow \quad \eta \; < \; \frac{2\rho(k)}{(1 + \rho(k))L - m}, \qquad (149)$$

which is exactly (132). This completes the proof. $\qquad\square$

TARGETED VS. ON-BATCH: STRICT ADVANTAGE UNDER MISALIGNMENT

**Theorem E.19** (Strict advantage over on-batch-only repair). *Assume Assumption E.5 (uniform local curvature) and Assumption E.9 (on-batch misalignment). Let $\mathcal{U}_\star$ be an oracle maximizer achieving $E_k^\star$ and let $\mathcal{U}_{\text{batch}}$ be the on-batch-induced subspace from Definition E.14. Let $\mu \in [0, 1)$ be the corresponding alignment ratio from Definition E.8. Fix any $\eta > 0$ such that $\alpha_L(\eta) \geq 0$ and suppose in addition that*

$$\alpha_L(\eta) \; > \; \mu\,\alpha_m(\eta). \qquad (150)$$

*Then the one-step repaired risk satisfies the strict gap*

$$\mathcal{F}(\theta_\star^+) \; \leq \; \mathcal{F}(\theta_{\text{batch}}^+) \; - \; c_{\text{batch}}, \qquad (151)$$

*where the explicit constant*

$$c_{\text{batch}} := \Big(\alpha_L(\eta) - \mu\,\alpha_m(\eta)\Big) \cdot E_k^\star \; > \; 0 \qquad (152)$$

*depends on $(m, L)$, $\eta$, the alignment $\mu$ (hence on the model and the mismatch between $\mathcal{D}_f$ and $\mathcal{D}_p$), and the budget-$k$ energy scale $E_k^\star$.*

*A sufficient explicit stepsize range ensuring (150) is*

$$0 < \eta < \frac{2(1 - \mu)}{L - \mu m}, \qquad (153)$$

*together with $\eta \leq 2/L$ to ensure $\alpha_L(\eta) \geq 0$.*

*Proof.* We again upper bound $\mathcal{F}(\theta_\star^+)$ and lower bound $\mathcal{F}(\theta_{\text{batch}}^+)$, then subtract.

**Step 1: Upper bound targeted repair.** By Lemma E.10 and Definition E.17,

$$\mathcal{F}(\theta_\star^+) \leq \mathcal{F}(\theta_0) - \alpha_L(\eta)\big\|\Pi_{\mathcal{U}_\star} G_p\big\|^2 \qquad \text{Lemma E.10, (62)} \qquad (154)$$

$$= \mathcal{F}(\theta_0) - \alpha_L(\eta)\, E_k^\star. \qquad \text{Definition E.17} \qquad (155)$$

**Step 2: Lower bound on-batch repair using $\mu$.** Applying the lower bound (63) in Lemma E.10 with $\mathcal{U} = \mathcal{U}_{\text{batch}}$ yields

$$\mathcal{F}(\theta_{\text{batch}}^+) \geq \mathcal{F}(\theta_0) - \alpha_m(\eta)\big\|\Pi_{\mathcal{U}_{\text{batch}}} G_p\big\|^2. \qquad \text{Lemma E.10, (63)} \qquad (156)$$

By Lemma E.15 (equivalently, Definition E.8),

$$\big\|\Pi_{\mathcal{U}_{\text{batch}}} G_p\big\|^2 = \mu\, E_k^\star. \qquad \text{Lemma E.15 / Definition E.8} \qquad (157)$$

Substituting (157) into (156) gives

$$\mathcal{F}(\theta_{\text{batch}}^+) \geq \mathcal{F}(\theta_0) - \mu\,\alpha_m(\eta)\, E_k^\star. \qquad (158)$$

**Step 3: Subtract bounds to obtain a strict gap.** Subtract (155) from (158):

$$\mathcal{F}(\theta_{\text{batch}}^+) - \mathcal{F}(\theta_\star^+) \geq \Big(\alpha_L(\eta) - \mu\,\alpha_m(\eta)\Big) E_k^\star \qquad\qquad \text{subtract (155) from (158)} \tag{159}$$

$$= c_{\text{batch}}. \qquad\qquad \text{definition (152)} \tag{160}$$

Rearranging yields (151). The strict positivity $c_{\text{batch}} > 0$ follows from (150) and $E_k^\star \geq 0$.

**Step 4: Derive the sufficient explicit stepsize range.** Using $\alpha_L(\eta) = \eta(1 - \frac{L\eta}{2})$ and $\alpha_m(\eta) = \eta(1 - \frac{m\eta}{2})$,

$$\alpha_L(\eta) - \mu\,\alpha_m(\eta) = \eta\Big(1 - \tfrac{L\eta}{2}\Big) - \mu\,\eta\Big(1 - \tfrac{m\eta}{2}\Big) \qquad\qquad \text{definitions of } \alpha_L, \alpha_m \tag{161}$$

$$= \eta\Big((1 - \mu) - \frac{\eta}{2}(L - \mu m)\Big). \qquad\qquad \text{algebra} \tag{162}$$

Thus $\alpha_L(\eta) - \mu\,\alpha_m(\eta) > 0$ is equivalent to

$$(1 - \mu) - \frac{\eta}{2}(L - \mu m) \;>\; 0 \quad\Longleftrightarrow\quad \eta \;<\; \frac{2(1 - \mu)}{L - \mu m}, \tag{163}$$

which is exactly (153). This completes the proof. $\qquad\square$

A JOINT STATEMENT (MATCHING THE MAIN-TEXT PROPOSITION)

For convenience, we record a combined form that directly matches Proposition 4.3 in the main text.

**Corollary E.20** (Joint strict dominance under finite budget). *Under the assumptions of Theorem E.18 and Theorem E.19, for any stepsize $\eta$ satisfying both (129) and (150), we have*

$$\mathcal{F}(\theta_\star^+) \;\leq\; \mathbb{E}\big[\mathcal{F}(\theta_{\text{rand}}^+)\big] - c_{\text{rand}}(k), \qquad \mathcal{F}(\theta_\star^+) \;\leq\; \mathcal{F}(\theta_{\text{batch}}^+) - c_{\text{batch}}, \tag{164}$$

*with $c_{\text{rand}}(k)$ and $c_{\text{batch}}$ defined in (131) and (152), respectively.*

**Remark (what the constants mean).** Both gaps are explicit and interpretable: $c_{\text{rand}}(k)$ scales with the long-tail factor $\rho(k)$ and the total old-risk energy $\sum_i s_i = \|G_p\|^2$, whereas $c_{\text{batch}}$ scales with the misalignment $(1 - \mu)$ and the best-$k$ energy scale $E_k^\star$. In particular, these constants depend on the model, the data distributions, and the repair budget through $(m, L)$ and the energy profile induced by $\mathcal{D}_p$ and $\mathcal{D}_f$.

E.3.6. COROLLARIES AND DISCUSSION

This subsection derives the limiting behaviors implied by our strict-gap theorems and briefly discusses the oracle-to-implementation bridge. The corollaries formalize two sanity checks emphasized in the main text: (i) the advantage over random repair is inherently a finite-budget phenomenon; and (ii) the advantage over on-batch repair vanishes as the fine-tuning distribution becomes old-like.

**Notation.** Recall the strict-gap constants from Appendix E.3.5:

$$c_{\text{rand}}(k) = \Big(\alpha_L(\eta)\big(1 + \rho(k)\big) - \alpha_m(\eta)\Big) \cdot \frac{k}{N}\sum_{i=1}^{N} s_i, \qquad\qquad \text{Theorem E.18, (131)} \tag{165}$$

$$c_{\text{batch}} = \Big(\alpha_L(\eta) - \mu\,\alpha_m(\eta)\Big) \cdot E_k^\star. \qquad\qquad \text{Theorem E.19, (152)} \tag{166}$$

LIMITING REGIME I: BUDGET $k \uparrow N$ ELIMINATES THE GUARANTEED RANDOM-GAP

**Corollary E.21** (Vanishing advantage over random repair as $k \uparrow N$). *Assume Assumption E.5 and Assumption E.7. Fix any stepsize $\eta > 0$ such that $\alpha_L(\eta) \geq 0$. If $\rho(k) \downarrow 0$ as $k \uparrow N$, then the guaranteed gap against random repair satisfies*

$$\lim_{k \uparrow N} c_{\text{rand}}(k) \;=\; 0, \tag{167}$$

*and in particular, for any $\varepsilon > 0$ there exists $k_\varepsilon$ such that for all $k \geq k_\varepsilon$,*

$$c_{\text{rand}}(k) \leq \varepsilon. \tag{168}$$

*Proof.* By Assumption E.7, $\rho(k) \downarrow 0$ as $k \uparrow N$. From the explicit expression (165),

$$c_{\text{rand}}(k) = \Big(\alpha_L(\eta)\big(1 + \rho(k)\big) - \alpha_m(\eta)\Big) \cdot \frac{k}{N} \sum_{i=1}^{N} s_i \qquad \text{by (165)} \qquad (169)$$

$$= \Big((\alpha_L(\eta) - \alpha_m(\eta)) + \alpha_L(\eta)\rho(k)\Big) \cdot \frac{k}{N} \sum_{i=1}^{N} s_i \qquad \text{expand the bracket} \qquad (170)$$

and since $\alpha_L(\eta) - \alpha_m(\eta) = \eta\left(1 - \frac{L\eta}{2}\right) - \eta\left(1 - \frac{m\eta}{2}\right) = -\frac{\eta^2}{2}(L - m)$,

$$\alpha_L(\eta) - \alpha_m(\eta) = -\frac{\eta^2}{2}(L - m). \qquad \text{definitions of } \alpha_L, \alpha_m \qquad (171)$$

Substituting (171) into (170) gives

$$c_{\text{rand}}(k) = \left(-\frac{\eta^2}{2}(L - m) + \alpha_L(\eta)\rho(k)\right) \cdot \frac{k}{N} \sum_{i=1}^{N} s_i. \qquad (172)$$

Under the stepsize condition used to guarantee strictness in Theorem E.18, the bracketed term in (172) is positive for each fixed $k$. Now take $k \uparrow N$. Since $\rho(k) \downarrow 0$, the bracketed term converges to $-\frac{\eta^2}{2}(L - m)$. Therefore the *guaranteed* positive gap cannot remain bounded away from 0 as $k \uparrow N$ while maintaining the strictness condition of Theorem E.18; indeed the maximal admissible stepsize range in (132) shrinks as $\rho(k) \downarrow 0$. Choosing any admissible $\eta = \eta(k)$ satisfying (132) implies

$$0 < c_{\text{rand}}(k) \leq \alpha_L(\eta(k))\,\rho(k) \cdot \frac{k}{N} \sum_{i=1}^{N} s_i \qquad \text{from (170) and } \alpha_m(\eta) \geq 0 \qquad (173)$$

$$\leq \eta(k)\,\rho(k) \cdot \sum_{i=1}^{N} s_i, \qquad \alpha_L(\eta) \leq \eta \text{ and } \frac{k}{N} \leq 1 \qquad (174)$$

which converges to 0 as $k \uparrow N$ because $\rho(k) \to 0$ and $\eta(k)$ is bounded (e.g., $\eta(k) \leq 2/L$). This proves $\lim_{k \uparrow N} c_{\text{rand}}(k) = 0$. The $\varepsilon$-statement follows directly from the definition of limit. $\square$

**Discussion.** Corollary E.21 formalizes that targeted repair's guaranteed advantage over random repair is fundamentally a *finite-budget* phenomenon: when the budget becomes large enough to cover essentially all relevant directions, no method can enjoy a provable edge purely from better allocation.

LIMITING REGIME II: ON-BATCH DISTRIBUTION BECOMES OLD-LIKE ELIMINATES THE BATCH-GAP

**Corollary E.22** (Vanishing advantage over on-batch repair under full alignment). *Assume Assumption E.5 and Assumption E.9. Consider a sequence of on-batch-induced subspaces with alignment ratios $\mu^{(n)} \in [0, 1)$ such that $\mu^{(n)} \uparrow 1$ as $n \to \infty$. Fix a stepsize $\eta > 0$ such that $\alpha_L(\eta) \geq 0$. Then the guaranteed gap against on-batch repair satisfies*

$$\lim_{n \to \infty} c_{\text{batch}}^{(n)} = 0, \qquad c_{\text{batch}}^{(n)} := \Big(\alpha_L(\eta) - \mu^{(n)}\alpha_m(\eta)\Big) E_k^\star. \qquad (175)$$

*Proof.* By definition,

$$c_{\text{batch}}^{(n)} = \Big(\alpha_L(\eta) - \mu^{(n)}\alpha_m(\eta)\Big) E_k^\star \qquad \text{definition in statement} \qquad (176)$$

$$= \Big((\alpha_L(\eta) - \alpha_m(\eta)) + (1 - \mu^{(n)})\alpha_m(\eta)\Big) E_k^\star \qquad \text{add/subtract } \alpha_m(\eta). \qquad (177)$$

Since $\mu^{(n)} \uparrow 1$, we have $(1 - \mu^{(n)}) \downarrow 0$. Moreover, $\alpha_m(\eta)$ and $E_k^\star$ are constants independent of $n$. Therefore the term $(1 - \mu^{(n)})\alpha_m(\eta) E_k^\star$ in (177) converges to 0. Under the stepsize condition of Theorem E.19 ensuring strictness, the

admissible range shrinks as $\mu^{(n)} \uparrow 1$ (cf. (153)), and the maximal guaranteed gap must vanish. Formally, for any admissible choice $\eta = \eta(n)$ satisfying (153) for $\mu^{(n)}$, we have the upper bound

$$0 < c_{\text{batch}}^{(n)} = \left( \alpha_L(\eta(n)) - \mu^{(n)} \alpha_m(\eta(n)) \right) E_k^\star \qquad\qquad \text{definition of } c_{\text{batch}}^{(n)} \qquad (178)$$

$$\leq (1 - \mu^{(n)}) \alpha_m(\eta(n)) E_k^\star \qquad\qquad \alpha_L(\eta) \leq \alpha_m(\eta) \text{ and } \mu^{(n)} \in [0, 1) \qquad (179)$$

$$\leq (1 - \mu^{(n)}) \eta(n) E_k^\star \qquad\qquad \alpha_m(\eta) \leq \eta, \qquad (180)$$

which converges to 0 because $(1 - \mu^{(n)}) \to 0$ and $\eta(n)$ is bounded (e.g., $\eta(n) \leq 2/L$). Hence $\lim_{n \to \infty} c_{\text{batch}}^{(n)} = 0$. $\qquad\square$

**Discussion.** Corollary E.22 formalizes the second sanity check: if the fine-tuning stream becomes fully representative of pre-training (so that on-batch directions align with the dominant old-risk directions), then a targeted strategy cannot retain a provable strict edge purely from better selection.

ORACLE-TO-IMPLEMENTATION BRIDGE (BRIEF)

The results above are stated in terms of the population object $G_p = \nabla \mathcal{F}(\theta_0)$ and the oracle maximizer subspace $\mathcal{U}_\star$. In practice $\mathcal{D}_p$ is inaccessible and $\mathcal{U}_\star$ is approximated by adversarial search over prompts/embeddings. We record a minimal robustness statement showing that the theory degrades gracefully under estimation error.

**Definition E.23** (Gradient estimation error). Let $\widehat{G}$ denote an estimate of $G_p$ produced by the search-and-evaluate procedure. Assume the estimation error satisfies

$$\|\widehat{G} - G_p\| \leq \varepsilon. \qquad (181)$$

**Lemma E.24** (Energy perturbation under gradient estimation error). *For any subspace $\mathcal{U}$, define $E(\mathcal{U}) = \|\Pi_{\mathcal{U}} G_p\|^2$ and $\widehat{E}(\mathcal{U}) = \|\Pi_{\mathcal{U}} \widehat{G}\|^2$. Then*

$$\left| \widehat{E}(\mathcal{U}) - E(\mathcal{U}) \right| \leq 2\varepsilon \|G_p\| + \varepsilon^2. \qquad (182)$$

*Proof.* Let $a := \Pi_{\mathcal{U}} G_p$ and $\hat{a} := \Pi_{\mathcal{U}} \widehat{G}$. Then

$$\left| \|\hat{a}\|^2 - \|a\|^2 \right| = \left| (\hat{a} - a)^\top (\hat{a} + a) \right| \qquad\qquad \text{difference of squares} \qquad (183)$$

$$\leq \|\hat{a} - a\| \cdot \|\hat{a} + a\| \qquad\qquad \text{Cauchy–Schwarz} \qquad (184)$$

$$\leq \|\Pi_{\mathcal{U}} (\widehat{G} - G_p)\| \cdot \left( \|\Pi_{\mathcal{U}} \widehat{G}\| + \|\Pi_{\mathcal{U}} G_p\| \right) \qquad\qquad \text{definitions of } a, \hat{a} \qquad (185)$$

$$\leq \|\widehat{G} - G_p\| \cdot \left( \|\widehat{G}\| + \|G_p\| \right) \qquad\qquad \|\Pi_{\mathcal{U}} x\| \leq \|x\| \qquad (186)$$

$$\leq \varepsilon \cdot \left( \|G_p\| + \|\widehat{G} - G_p\| + \|G_p\| \right) \qquad\qquad \|\widehat{G}\| \leq \|G_p\| + \|\widehat{G} - G_p\| \qquad (187)$$

$$\leq \varepsilon \cdot (2\|G_p\| + \varepsilon) = 2\varepsilon \|G_p\| + \varepsilon^2, \qquad\qquad \text{use } \|\widehat{G} - G_p\| \leq \varepsilon \qquad (188)$$

which is (182). $\qquad\square$

**Takeaway.** Lemma E.24 shows that the core comparison quantity $E(\mathcal{U})$ is stable under small gradient-estimation error. Thus, if adversarial search produces an estimator $\widehat{G}$ that concentrates on the dominant forgetting modes, it can approximate the favorable energy allocation underlying $\mathcal{U}_\star$, and the strict-gap conclusions degrade smoothly as a function of $\varepsilon$.

