# OpenReview forum: "Adversarial Latent Embedding Repair for LLM Continual Learning"
_ICML.cc/2026/Conference — ICML 2026 regular_

### Official Review · Reviewer_j1MV · 2026-02-28

**Soundness:** 3
**Presentation:** 3
**Significance:** 3
**Originality:** 3
**Overall Recommendation:** 4
**Confidence:** 4

**Summary:**

This paper tackles catastrophic forgetting in large language models during continual learning, specifically when pre-training data is unavailable. The authors propose ALER, a framework that follows a find-before-fix protocol. It adversarially searches for latent prompt embeddings that expose high-risk forgetting modes and then performs online distillation from a frozen reference model to repair these vulnerabilities. This work intends to assess the central question of how to effectively identify and fix knowledge loss without data replay. Experiments on domain-specific tasks and general benchmarks show it helps retain general capabilities while adapting to new domains.

**Compliance With Llm Reviewing Policy:**

Affirmed.

**Final Justification:**

I keep my original rating as weak accept.

**Key Questions For Authors:**

1.	How is the search overhead expected to scale with much larger models (e.g., 70B+)? Are there plans to optimize the inner loop?

2.	How sensitive is the method to the quality of the frozen reference model? Would an outdated reference model hinder adaptation to new knowledge?

3.	Given the noted non-smoothness of the objective, were there instances where the search failed to identify meaningful forgetting modes?

**Limitations:**

yes

**Strengths And Weaknesses:**

Strengths:

1.	Unlike most methods that react to forgetting after it happens, ALER tries to locate vulnerable regions before repairing them. This makes sense given that forgetting is often concentrated rather than uniform.

2.	The paper provides proofs showing why targeted repair on high-risk prompts is more efficient than random or on-batch repair under a finite budget.

3.	Solid Empirical Results: The evaluation covers multiple benchmarks (MMLU, GSM8K, etc.), and the method consistently keeps general performance stable while improving target tasks.


Weaknesses:

1.	The inner-loop adversarial search adds noticeable runtime overhead (about 1.5x compared to standard SFT). For very large models, this could become a bottleneck.

2.	There are new parameters like latent embedding length and batch size. While sensitivity analysis is provided, it's unclear how well these transfer across different model scales without re-tuning.

3.	The objective landscape is non-smooth, which might lead to unstable gradients during the search phase. The paper acknowledges this, but it remains a practical risk.

---

> ### Author Rebuttal · Authors · 2026-03-31
>
> # Response to Reviewer j1MV
>
> > Detailed experimental results are provided in the supplementary material: https://anonymous.4open.science/r/alerdistill-supplementary/supplementary.pdf
>
> We sincerely appreciate your constructive comments. Please find below our detailed responses to the concerns and suggestions you raised.
>
> ## **Weakness 1 & Question 1: Runtime Overhead and Scaling**
>
> **We agree that the inner-loop search introduces noticeable overhead.** To quantify the precision–efficiency trade-off, we performed an additional **search-frequency ablation** on the 4B setting, where adversarial search and repair are activated once every `k` optimization steps.
>
> > Table R1. Search-frequency ablation on the 4B setting.
>
> | Method | Target (SciKnowEval) | Target Impr. | Avg. on Six Benchmarks | Avg. Impr. | Time per Step | Time Multiplier |
> | - | :-: | :-: | :-: | :-: | :-: | :-: |
> | SFT | 79.17 |  | 64.36 |  | 3.04 |  |
> | Ours+OPD (k=1) | 81.67 | +3.16% | 76.79 | +19.31% | 4.50 | 1.48× |
> | k=2 | 80.83 | +2.10% | 74.46 | +15.69% | 4.05 | 1.33× |
> | k=4 | 77.50 | -2.11% | 74.53 | +15.80% | 3.75 | 1.23× |
> | k=8 | 80.00 | +1.05% | 72.70 | +12.96% | 3.50 | 1.15× |
> | k=16 | 79.17 | +0.00% | 68.77 | +6.85% | 3.29 | 1.08× |
>
> **Key finding**: `k=2–4` preserves most retention gain while reducing the multiplier to `1.23–1.33×`; overly sparse search (`k=16`) markedly degrades retention.
>
> We further validated on `Qwen3-8B` (same hyperparameters except `lr=2e-5`):
>
> > Table R2. Runtime and performance on the 8B setting.
>
> | Method | Target (SciKnowEval) | Target Impr. | Avg. on Six Benchmarks | Avg. Impr. | Time per Step | Time Multiplier |
> | - | :-: | :-: | :-: | :-: | :-: | :-: |
> | SFT | 71.67 |  | 57.99 |  | 10.21 |  |
> | SFT+KL | 70.00 | -2.33% | 66.41 | +14.51% | 13.34 | 1.31× |
> | Ours+OPD | 70.83 | -1.17% | 71.57 | +23.42% | 15.62 | 1.53× |
>
> These results on 4B and 8B models provide **encouraging evidence that the overhead ratio remains stable across scales**. Extending to 70B+ models is a natural next step. **Plans to optimize the inner loop** includes:
> - **Sparser or adaptive search schedules**, since `k=2–4` already retains most of the benefit;
> - **Asynchronous search**, so search and main training need not be fully serialized;
> - **Top-layer / hidden-state search** and **normalized or approximate single-step ascent**, to reduce both cost and instability in the inner loop.
>
> ## **Weakness 2: Hyperparameter Transfer Across Scales**
>
> Notably, for the 8B experiment above, **`NO` ALER-specific hyperparameters were re-tuned** (only the base learning rate was adjusted). The method still achieved strong retention and a comparable overhead ratio, suggesting it is not overly brittle to moderate scale changes. **We will add broader cross-scale transfer analysis as a future direction**.
>
> ## **Question 2: Sensitivity to the Frozen Reference Model**
>
> In our current setup, the frozen reference is the **pre-fine-tuning checkpoint itself**, i.e., exactly the prior behavior we aim to preserve. In that sense, it is not “outdated” relative to the preservation target. Two design choices further mitigate the risk of an overly constraining reference:
> - **repair is targeted rather than global**, i.e., applied only to searched high-risk modes;
> - the **retention–adaptation balance is explicitly controlled** by the repair weight `λ`.
>
> Empirically, we consistently observe **strong target adaptation together with improved retention**, suggesting that the current reference does not prevent learning new domain knowledge. We agree, however, that a systematic study of **stale or mismatched references** would be valuable, and we will add this discussion to the revision.
>
> ## **Weakness 3 & Question 3: Non-smooth Objective and Search Stability**
>
> We agree that the search landscape is rugged and non-smooth. In practice, however, we did **not** observe systematic search collapse or obvious instability. Three pieces of evidence are relevant.
>
> - **Stabilizing design choices**: batched search, manifold grounding, and entropy regularization collectively smooth the inner-loop landscape.
> - **Ablation evidence**: replacing gradient-based search with random embeddings (`RE`) or random search (`RS`), or removing the manifold constraint (`WM`), all degrade performance—confirming that the search identifies functionally meaningful forgetting modes.
> - **Generalization beyond synthetic probes**: supplementary Fig. 1 shows that ALER moves substantially closer to the frozen reference on **held-out searched probes**, also improves on **risk-enriched natural old-domain prompts**, and changes **target-domain validation prompts** only minimally—evidence that the search is not merely exploiting unstable synthetic artifacts.
>
> In future work, we plan to explore **more stable inner-loop updates**, including normalized or approximate single-step ascent schemes, together with the efficiency optimizations above.

---

> > ### Author Rebuttal · Reviewer_j1MV · 2026-04-02
> >
> > My concern have been well addressed. I recommend a weak accept.

---

> > > ### Author Response · Authors · 2026-04-02
> > >
> > > Dear Reviewer j1MV,
> > >
> > > Thank you for reviewing our rebuttal and your positive feedback. We are glad that our additional experiments on scaling and overhead, along with our further clarifications, have fully addressed your concerns.
> > >
> > > We sincerely appreciate the time and effort you dedicated to reviewing our work. Your constructive feedback has been invaluable in strengthening the overall quality of our manuscript.
> > >
> > > Best regards,
> > > The Authors

---

### Official Review · Reviewer_C9QV · 2026-03-10

**Soundness:** 3
**Presentation:** 2
**Significance:** 3
**Originality:** 2
**Overall Recommendation:** 4
**Confidence:** 4

**Summary:**

Standard fine-tuning of Large Language Models (LLMs) often causes catastrophic forgetting of prior knowledge, especially when original pre-training data is unavailable.

The proposed method, ALER is a data-free framework that proactively identifies "high-risk" knowledge areas by adversarially searching for latent prompt embeddings that maximize divergence from the original model.

The system performs online distillation using these discovered embeddings to mirror the original model's behavior while simultaneously adapting to the new target domain.

Supported by theoretical efficiency guarantees, ALER consistently outperformed baselines across two domain-specific datasets and six general benchmarks.

This approach shifts continual learning from reactive mitigation to a proactive repair strategy, effectively balancing the frontier between knowledge retention and new skill acquisition

**Compliance With Llm Reviewing Policy:**

Affirmed.

**Final Justification:**

My concerns have been addressed. I raise my score to weak accept.

**Key Questions For Authors:**

See the above comments - observations, comparisons, and Performances.

**Limitations:**

yes.

**Strengths And Weaknesses:**

(+) By analyzing learning dynamics, the authors proved that forgetting during domain-specific fine-tuning is not a uniform decline but rather a non-uniform, long-tailed process targeting specific fragile behaviors.

(+) The proposed ALER framework employs an adversarial search to identify latent prompt embeddings that expose these high-risk "forgetting modes," allowing for targeted online distillation to repair the model's prior knowledge without needing pre-training data.

(+) Results across multiple benchmarks demonstrate that ALER effectively preserves general-purpose performance while adapting to new domains, successfully pushing the retention - adaptation frontier beyond existing baselines.

(-) **Empirical Evidence**: The framework lacks concrete observational data to support the conceptual distribution shifts (repaired vs. target) illustrated in its theoretical Figure 2.

(-) **Baseline Comparison**: The study requires more rigorous comparisons against prior distillation-based methods to demonstrate its unique effectiveness clearly.

(-) **Performances**: The observed performance improvements are relatively minor, raising questions about the practical significance of the gain in real-world applications.

---

> ### Author Rebuttal · Authors · 2026-03-31
>
> # Response to Reviewer C9QV
>
> > Detailed experimental results are provided in the supplementary material: https://anonymous.4open.science/r/alerdistill-supplementary/supplementary.pdf
>
> We sincerely appreciate your constructive comments. Please find below our detailed responses to the concerns and suggestions you raised.
>
> ## **Weakness 1: Empirical Evidence for Distributional Shifts**
>
> We agree that Fig. 2 is a conceptual schematic, and we therefore added **a direct empirical counterpart**.
>
> **Setup.** We fine-tune `Qwen3-4B-Instruct-2507` on `SciKnowEval`. `Ref` = frozen pre-trained model; `SFT` = standard fine-tuning; `ALER` = our method. We evaluate three prompt domains: 256 **held-out searched probes** excluded from repair (HSP), 200 **natural old-domain prompts** from `GPQA` and `MMLU-Pro` (NOP), and 120 **target-domain validation prompts** from `SciKnowEval` (TVP). We report Jensen–Shannon divergence (JS) over output distributions. All other settings follow the main paper.
>
> **Results** (please see [Supplementary Fig. 1](https://anonymous.4open.science/r/alerdistill-supplementary/supplementary.pdf)):
>
> 1. **Repair concentrates where forgetting occurs.** On HSP, ALER reduces mean JS divergence to the reference from `0.473` to `0.014` (prompt-wise win rate: **100%**).
> 2. **Repair generalizes to natural old-domain prompts.** On NOP, mean divergence decreases from `0.633` to `0.564` (win rate: **95%**), confirming that the benefit is not confined to synthetic probes.
> 3. **Target adaptation is preserved.** On TVP, `JS(SFT, ALER) = 0.021`; `JS(Ref, ·)` is virtually unchanged (`0.450` for SFT vs. `0.450` for ALER).
>
> We retain Fig. 2 in the main submission as an **intuitive high-level summary of the proposed mechanism**, and we now complement it with **direct quantitative evidence** matching the schematic. We will include this analysis in the revision as a direct empirical counterpart to the conceptual schematic.
>
> ## **Weakness 2: Distillation-Based Baseline Comparisons**
>
> We appreciate this suggestion and we have added three representative distillation baselines:
> - **LwF (= our SFT+KL)** [1]: applies a KL penalty between the current model and the frozen reference on every training batch, which is the most widely adopted distillation-based continual learning strategy.
> - **GKD** [2]: on-policy generalized knowledge distillation that trains on student-generated outputs to reduce the train–inference distribution mismatch inherent in standard KD.
> - **L2KD** [3]: lifelong language knowledge distillation with task-aware soft-label transfer designed for sequential task learning.
>
> | Method | Target (SciKnowEval) | Target Impr. | Avg. on Six Benchmarks | Avg. Impr. |
> | - | :-: | :-: | :-: | :-: |
> | Naive | 41.67 | — | 76.67 | — |
> | SFT | 79.17 | +37.50 | 64.36 | −12.31 |
> | LwF (= our SFT+KL) | 78.33 | +36.66 | 72.37 | −4.30 |
> | GKD | 74.17 | +32.50 | 71.17 | −5.50 |
> | L2KD | 72.50 | +30.83 | 74.40 | −2.27 |
> | **Ours+OPD** | **81.67** | **+40.00** | **76.79** | **+0.12** |
>
> **Key observations.**
> - ALER achieves both the **best target performance** and the **best overall retention** among the compared distillation-based methods. We will add these baselines and their citations explicitly in the revision.
> - This comparison clarifies that ALER’s advantage is not merely due to using distillation, but due to **combining targeted search with repair**.
>
> ## **Weakness 3: Practical Significance of the Gains**
>
> We agree that some absolute gains on individual metrics appear moderate. However, ALER's primary goal is **not** to maximize target performance in isolation but to **improve the retention–adaptation frontier** under a strict no-replay, no-pretraining-data setting.
>
> 1. **Evaluation objective**: We intentionally evaluate **both** target adaptation and retention on six general-purpose benchmarks, because the core challenge in our setting is not maximizing target accuracy alone, but adapting to a new domain **without sacrificing prior capabilities**.
> 2. **Why this is meaningful**: Under this criterion, the gain is **not minor**. On SciKnowEval, standard SFT improves the target score to **79.17** but drops the general average from **76.67** to **64.36**; in contrast, **Ours+OPD reaches 81.67 while restoring the general average to 76.79**, and **Ours+WD reaches 76.92** on the general average.
> 3. **Revision**: We will revise the paper to make this framing clearer and avoid presenting ALER as a method whose primary goal is simply higher target-task performance.
>
> ## Reference
> [1] Li, et al. Learning without forgetting. IEEE transactions on pattern analysis and machine intelligence, 40(12), 2935-2947.
>
> [2] Agarwal, et al. On-policy distillation of language models: Learning from self-generated mistakes. The twelfth international conference on learning representations. 2024.
>
> [3] Chuang, et al. Lifelong language knowledge distillation. Proceedings of the 2020 Conference on Empirical Methods in Natural Language Processing (EMNLP). 2020.

---

> > ### Author Rebuttal · Reviewer_C9QV · 2026-04-03
> >
> > My concerns have been addressed. I raise my score to weak accept.

---

> > > ### Author Response · Authors · 2026-04-03
> > >
> > > Dear Reviewer C9QV,
> > >
> > > Thank you for reviewing our rebuttal and for your positive feedback. We are very glad that our additional empirical evidence, baseline comparisons, and further clarifications have fully addressed your concerns.
> > >
> > > We sincerely appreciate the time and effort you dedicated to reviewing our work. Your constructive feedback has been invaluable in helping us improve the clarity, evaluation, and overall quality of our manuscript.
> > >
> > > Best regards,
> > > The Authors

---

### Official Review · Reviewer_ywtB · 2026-03-11

**Soundness:** 3
**Presentation:** 1
**Significance:** 3
**Originality:** 3
**Overall Recommendation:** 4
**Confidence:** 4

**Summary:**

This paper focuses on the continual domain-specific fine-tuning of LLMs. They observe that forgetting during fine-tuning is not uniform but highly dependent on specific samples. Hence, they propose an adversarial prompt-search paradigm as a regularizer to optimize a worst-case forgetting surrogate in order to regularize the model training during continual fine-tuning. They evaluate their method on multiple benchmarks and show the relative effectiveness of their method.

**Compliance With Llm Reviewing Policy:**

Affirmed.

**Final Justification:**

I believe the rebuttal adequately addresses my concerns, mostly having to do with notation and discussion of other works.I remain supportive of the paper and retain my positive score.

**Key Questions For Authors:**

In addition to addressing the points in the weaknesses section, could the authors answer the following questions:

- For Fig 1., is $\Delta \text{NLL}(x)$ measured after each update? Or measured for all samples after training is complete?

- The naming convention and choice between $ \mathcal{L}\_{\text{OPD}} $ , $\mathcal{L}_{\text{WD}}$ is not well motivated, since in both cases only $\theta_t$ is optimized. Can the authors elaborate on the reasoning? Have the authors considered the Jensen-Shannon divergence instead?

- What is $K$ in Algorithm 1 set to in practice? Can the authors justify the additional computational budget required as compared to the empirical performance gained?

- Will the authors investigate and explain the connection/differences with the relevant works mentioned in the weaknesses section, and discuss these connections if necessary?

- $\mu_p$ is not defined anywhere as far as I can tell. Can the authors elaborate?

- Is $\pi_{\text{ref}}$ based on a frozen model or previously computed frozen embeddings? in other words what are the additional memory constraints?

Finally, below are some minor presentation points:

- Line 83, Column 2: the constants c should be introduced either in the beginning before eq.3 or after eq. 4, where they are used.
- The definition of c is ambiguous, as it changes based on the index.

**Limitations:**

Yes

**Strengths And Weaknesses:**

## Strengths

- **Soundness**:  The authors show both theoretical and empirical insights into the dynamics of forgetting during continual learning, and provide an intuitively sound algorithm to reduce forgetting.
- **Significance**: Continual Learning remains a practically significant and largely unsolved problem. This paper steps in the direction of a better understanding of forgetting dynamics and lower forgetting during continual learning.
- **Originality**: Regularizing the distribution of outputs with respect to an adversarially generated prompt is largely novel, with some exceptions as mentioned in the weaknesses section below.

## Weaknesses

- **Presentation**: Symbols and concepts are scattered across the paper and introduced ad hoc. There are many important concepts that are only elaborated in the appendix, and many mathematical symbols are defined through long, unnecessary chains. Overall, the composition of the paper is hard to follow and takes much time and effort to understand a relatively simple method. Related works are only mentioned in the appendix. Concepts such as NTKs should ideally be introduced on a high level in the main text.

- **Originality**:  There are two existing published works within the continual learning literature that are highly similar but not cited by the paper: [1] is a seminal work in continual learning and introduces the concept of "Dark experience replay," which is connected to regularizing the KL divergence of the latent space between the present and the past. [2] Also explores adversarial "worst-case" KL-divergence regularization for reducing forgetting, in the parameter space instead of the prompt space.

[1] Buzzega, Pietro, et al. "Dark experience for general continual learning: a strong, simple baseline." Advances in neural information processing systems 33 (2020): 15920-15930.

[2] Eskandar, Masih, et al. "Star: Stability-inducing weight perturbation for continual learning." arXiv preprint arXiv:2503.01595 (2025).

---

> ### Author Rebuttal · Authors · 2026-03-31
>
> # Response to Reviewer ywtB
>
> > Detailed experimental results are provided in the supplementary material: https://anonymous.4open.science/r/alerdistill-supplementary/supplementary.pdf
>
> We sincerely appreciate your careful reading and constructive comments. Please find below our detailed responses to the concerns and suggestions you raised.
>
> ## **Weakness 1: Presentation**
>
> We agree that the current presentation is denser and more fragmented than necessary. In the revision, we will:
>
> - move the most relevant related work from the appendix into the main text;
> - introduce the high-level intuition of the method (“find before fix”) earlier, before the formal derivation;
> - add a brief, intuitive explanation of the NTK perspective in the main text rather than deferring it almost entirely to the appendix;
> - simplify notation and define symbols closer to first use;
> - reduce overloaded notation and shorten long symbol-definition chains.
>
> We thank the reviewer for highlighting this issue.
>
> ## **Weakness 2: Relation to DER and STAR**
>
> We thank the reviewer for pointing out these highly relevant works. We will add explicit citations and a clearer comparison in the revision. ALER is related to both, but **differs in setting and mechanism**:
>
> - **DER** [1] stores past logits in a replay buffer and regularizes the current model's latent representations to match them. It therefore **requires a rehearsal memory of past data**, which is unavailable in our pre-training-data-inaccessible setting.
> - **STAR** [2] searches for worst-case **parameter perturbations** that reduce KL divergence, encouraging flat loss basins for stability. ALER instead searches in the **prompt/latent embedding space** to locate high-risk forgetting modes and then repairs them via targeted distillation from a frozen reference. The search space, the objective, and the repair mechanism are therefore different.
>
> ## **Questions**
>
> - **Q1: Measurement of $\Delta \text{NLL}$ in Fig. 1** $\Delta \text{NLL}$ is measured **after training is complete**: we compute NLL under the fine-tuned model minus NLL under the pre-trained model for each sample. We will clarify this in the figure caption.
>
> - **Q2: Comparison between different distillation backends**: Our intention was to distinguish the distillation behavior and objective, not the optimized variable. `OPD` uses reverse KL and is more **mode-seeking**, which we found **better suited for preserving newly acquired target-domain modes** while suppressing deviations from the reference. `WD` uses forward KL and is more directly aligned with the proxy forgetting-risk formulation. We agree the motivation should be clearer. We will also briefly discuss Jensen–Shannon divergence as a possible symmetric alternative.
>
> - **Q3: $K$ in Algorithm 1 and additional compute**
>     - $K$ refers to the search steps and in all experiments $K=10$.
>     - **Memory** ALER requires **a frozen reference model**, so its memory footprint is higher than plain SFT but comparable to other reference-based baselines.
>     - **Runtime** ALER does incur additional time overhead. To quantify the compute–performance tradeoff, we performed an additional search-frequency ablation on the 4B setting, where adversarial search and repair are activated once every `k` optimization steps.
>
> > Please refer to `Table R1` at Reviewer gR2q due to limited space.
>
> These results show a clear tradeoff: the full method performs best, while **less frequent search retains much of the benefit at lower runtime**.
>
> - **Q4: Discussion of related works.** Yes. We will add both citations and a dedicated discussion comparing ALER with replay-based dark-knowledge methods and worst-case stability regularization methods.
>
> - **Q5: $\mu_p$ is not defined**: Thank you for catching this omission. $\mu_p$ denotes the distribution over pre-training context and is introduced in Appendix E.2. We agree it should be defined explicitly near Eq. (5), and we will fix this in the revision.
>
> - **Q6: $\pi_\text{ref}$**: $\pi_\text{ref}$ is a frozen reference model initialized from the same pre-trained checkpoint. As noted above, this requires an additional frozen model, but no replay buffer of pre-training data.
>
> - **Minor presentation points**. We agree that the constants $c$ should be introduced closer to where they are used, and that the current notation is ambiguous. We will revise these definitions and remove symbol overloading to improve clarity.
>
> We thank the reviewer again for the thoughtful feedback. We believe these revisions will substantially improve the paper’s clarity and positioning.
>
> ## Reference
>
> [1] Buzzega, et al. Dark experience for general continual learning: a strong, simple baseline. Advances in neural information processing systems 33 (2020): 15920-15930.
>
> [2] Eskandar, et al. Star: Stability-inducing weight perturbation for continual learning. arXiv preprint arXiv:2503.01595 (2025).

---

> > ### Author Rebuttal · Reviewer_ywtB · 2026-04-02
> >
> > Thank you for the detailed response. I have remaining concerns regarding the runtime. You said K = 10 in all experiments, and in the paper, it's noted that it involves gradient ascent steps. That means 10 additional forward and backward passes. However, in Table R1, even when k=1, meaning 10 extra forward-backward steps per optimization step, the slowdown is only around 48%. Can the authors elaborate on why, despite the relatively high number of passes (11 compared to 1), the slowdown is relatively small?

---

> > > ### Author Response · Authors · 2026-04-03
> > >
> > > # Response to Reviewer ywtB
> > >
> > > We thank the reviewer for the follow-up question. In our implementation, the search phase and the main update phase carry **not directly comparable computational loads**. The moderate overhead is due to the following structural differences:
> > >
> > > - **Frozen Base Model**: During the adversarial search phase, the current model parameters are frozen. We only perform gradient ascent steps on the latent prompt embedding rather than updating the entire network.
> > > - **Smaller Search Configuration**: The search operates on very small latent embeddings, specifically utilizing a prompt length of `8` and a batch size of `8`. This is substantially smaller than the main target-batch fine-tuning phase, which uses a global batch size of `32`  over full-length sequences.
> > >
> > > In summary, due to the fundamental differences in both the optimized variables and the sequence dimensions, the computational burden of the 10 lightweight search steps is not linearly comparable to 10 standard training updates.

---

### Official Review · Reviewer_gR2q · 2026-03-12

**Soundness:** 3
**Presentation:** 3
**Significance:** 3
**Originality:** 3
**Overall Recommendation:** 4
**Confidence:** 4

**Summary:**

This paper focuses on the forgetting issue of LLMs during domain-specific fine-tuning and proposes the ALER framework. At its core is a data-free mechanism: it proactively detects the model’s high-risk forgetting modes via adversarial search in the latent space, and performs online distillation repair using the discovered latent embeddings. Experiments demonstrate that this approach can effectively preserve the model’s general capabilities while improving new skills.

**Compliance With Llm Reviewing Policy:**

Affirmed.

**Final Justification:**

Thanks the authors for their rebuttal. It has addressed some of my concerns; therefore, I am maintaining weak accept.

**Key Questions For Authors:**

1. Searching for adversarial embeddings at each step requires multiple forward and backward passes. How does ALER balance the trade-off between the precision of risk discovery and the overall training wall-clock time, especially when scaling to very large models?

2. How does ALER interact with PEFT methods like LoRA? Since LoRA only updates a small subset of parameters, does the adversarial search become more efficient, or does it require different search hyperparameters compared to full-parameter fine-tuning?

**Limitations:**

Please see Weakness

**Strengths And Weaknesses:**

Strengths
1. The paper shifts the CL paradigm from passive constraints to proactive discovery by introducing an innovative adversarial search mechanism in the latent space.

2. The research targets a problem of significant practical importance and immediate relevance in real-world deployment.

3. The findings are backed by rigorous validation, including extensive performance benchmarking and systematic ablation studies.

Weaknesses:
1. Adversarial search is performed in every fine-tuning step, typically requiring multiple gradient backward passes. This can significantly increase training time compared to standard fine-tuning. While the authors may mitigate this by reducing search steps, such a trade-off could reduce the accuracy of identifying high-risk forgetting regions.

2. The embeddings found via adversarial search may be mathematically “high-risk” but do not correspond to any meaningful natural language (i.e., these vectors lie off the manifold of valid linguistic representations). Although the authors empirically validate the effectiveness of ALER, it remains unclear whether such “logic-disconnected” repair could harm the reasoning coherence of the model.

3. Experiments are mostly conducted on relatively short sequences or specialized prompt embeddings. For continual learning with extremely long contexts, there is still insufficient empirical evidence to confirm whether such point-wise embedding-based repair can adequately mitigate forgetting in long-range dependencies.

---

> ### Author Rebuttal · Authors · 2026-03-31
>
> ## Response to Reviewer gR2q
>
> > Detailed experimental results are provided in https://anonymous.4open.science/r/alerdistill-supplementary/supplementary.pdf
>
> We sincerely appreciate your constructive comments. Please find below our detailed responses to the concerns and suggestions you raised.
>
> ## **Weakness 1 & Question 1: Training Efficiency**
>
> **We agree that runtime is an important practical concern**. To directly quantify the precision–efficiency trade-off, we performed an additional search-frequency ablation on the 4B setting, where adversarial search and repair are activated once every `k` optimization steps.
>
> > Table R1. Search-frequency ablation on the 4B setting.
>
> | Method | Target (SciKnowEval) | Target Impr. | Avg. on Six Benchmarks | Avg. Impr. | Time per Step | Time Multiplier
> | - | :-: | :-: | :-: | :-: | :-: | :-:
> | SFT | 79.17 |  | 64.36 |  | 3.04 |
> | Ours+OPD (k=1) | 81.67 | +3.16% | 76.79 | +19.31% | 4.50 | 1.48×
> | k=2 | 80.83 | +2.10% | 74.46 | +15.69% | 4.05 | 1.33×
> | k=4 | 77.50 | -2.11% | 74.53 | +15.80% | 3.75 | 1.23×
> | k=8 | 80.00 | +1.05% | 72.70 | +12.96% | 3.50 | 1.15×
> | k=16 | 79.17 | +0.00% | 68.77 | +6.85% | 3.29 | 1.08×
>
>
> Two conclusions are clear:
> - **ALER does not require search at every step**: `k=2-4` retains most of the benefit while reducing runtime substantially.
> - **Excessively sparse search degrades retention**, confirming that less frequent risk discovery weakens localization of high-risk forgetting modes.
>
> We also tested a larger `Qwen3-8B` model (same setup, except `lr=2e-5`):
>
> > Table R2. Runtime and performance on the 8B setting.
>
> | Method | Target (SciKnowEval) | Target Impr. | Avg. on Six Benchmarks | Avg. Impr. | Time per Step | Time Multiplier |
> | - | :-: | :-: | :-: | :-: | :-: | :-:
> | SFT | 71.67 |  | 57.99 |  | 10.21
> | SFT+KL | 70.00 | -2.33% | 66.41 | +14.51% | 13.34 | 1.31×
> | Ours+OPD | 70.83 | -1.17% | 71.57 | +23.42% | 15.62 | 1.53×
>
>
> - **The runtime multiplier remains similar** from `4B` to `8B` (`1.48×` to `1.53×`), rather than exploding with model size;
> - **The retention benefit persists at larger scale**, with Avg. improving from 57.99 to 71.57 over SFT.
>
> In the revision, we will add both the search-frequency ablation and the 8B runtime results.
>
> ## **Weakness 2: Semantic Validity and Reasoning Coherence**
>
> The searched prompts are **NOT** guaranteed to correspond to natural-language inputs; our goal is therefore **not to claim linguistic interpretability**, but to **reduce off-manifold drift** and test whether repair harms natural-language reasoning in practice.
>
> - **Method design**. ALER uses manifold grounding and entropy regularization to bias the search toward stable, distillable prompts.
> - **Ablation evidence**. Removing the manifold constraint (WM) degrades both target accuracy and general retention, indicating that this design improves search quality.
> - **Behavioral outcome**. The searched prompts are used only as training-time probes, and the repair is a conservative alignment toward the frozen reference. On standard natural benchmarks, we do not observe systematic degradation in reasoning-related performance.
>
> ## **Weakness 3: Long Context Effectiveness**
>
> **The current submission does not evaluate ALER in extremely long-context continual learning. We will state this explicitly as a limitation.** Our present scope is domain-specific post-training with point-wise latent prompt search and targeted online distillation, not a dedicated long-context repair framework. Long-context extensions (e.g., chunk-/segment-level repair) are **an important direction for future work**.
>
> ## **Question 2: Interact with PEFT Methods**
>
> **ALER is compatible with gradient-based PEFT methods such as LoRA**, because its search-and-repair mechanism is defined at the **functional level** rather than being tied to a specific full-parameter update rule. We verified this with an **additional LoRA experiment** under the same SciKnowEval setting, using a standard LoRA configuration (`rank = 16`, `alpha = 32`).
>
> > Table R3. Results under LoRA
>
> | Method | Target (SciKnowEval) | Target Impr. | Avg. on Six Benchmarks | Avg. Impr. | Time per Step | Time Multiplier |
> | - | :-: | :-: | :-: | :-: | :-: | :-: |
> | SFT+LoRA | 69.17 |  | 72.75 |  | 2.94 | 1.00× |
> | SFT+KL+LoRA | 66.67 | -3.61% | 74.88 | +2.93% | 3.80 | 1.29× |
> | Ours+LoRA | 68.33 | -1.21% | 76.42 | +5.05% | 4.43 | 1.51× |
>
> Two points are worth noting:
> - **ALER remains effective under LoRA: `Ours+LoRA` achieves the best general retention among the LoRA baselines**.
> - **The target-side trade-off is more constrained** under LoRA than under full fine-tuning, which is expected given the limited trainable capacity of PEFT. Even so, `Ours+LoRA` remains competitive on the target task while providing the strongest retention.
>
> Overall, these results indicate that ALER is not tied to full-parameter fine-tuning: its retention benefit persists under LoRA/PEFT as well. We will add this result and discussion to the revision.

---

### Decision · Program_Chairs · 2026-04-30

**Decision:**

Accept (regular)

**Comment:**

This paper proposes the ALER, a data-free approach that proactively mitigates forgetting in large language models during domain-specific fine-tuning through adversarial search in the latent space. Reviewers appreciate the following points: (1) shifting the continual learning paradigm from passive constraints to proactive discovery, (2) backing their claims with sound theoretical insights and solid empirical results. However, reviewers raised valid concerns: (1) the computational overhead of the search mechanism, (2) the clarity of the presentation, (3) and the robustness of baseline comparisons. The authors successfully addressed these specific issues in their rebuttal by clarifying the mathematical notations and discussing the relevant prior literature in more detail. Because the framework provides a technically solid advancement with unified backing from the review committee, the AC recommends to accept the paper to ICML 2026.